# SHARPER ANALYSIS OF SPARSELY ACTIVATED WIDE NEURAL NETWORKS WITH TRAINABLE BIASES

## ABSTRACT

This work studies training one-hidden-layer overparameterized ReLU networks via gradient descent in the neural tangent kernel (NTK) regime, where, differently from the previous works, the networks' biases are trainable and are initialized to some constant rather than zero. The tantalizing benefit of such initialization is that the neural network will provably have sparse activation pattern before, during and after training, which can enable fast training procedures and, therefore, reduce the training cost. The first set of results of this work characterize the convergence of the network's gradient descent dynamics. The required width is provided to ensure gradient descent can drive the training error towards zero at a linear rate. The contribution over previous work is that not only the bias is allowed to be updated by gradient descent under our setting but also a finer analysis is given such that the required width to ensure the network's closeness to its NTK is improved. Secondly, the networks' generalization bound after training is provided. A width-sparsity dependence is presented which yields sparsity-dependent localized Rademacher complexity and a generalization bound matching previous analysis (up to logarithmic factors). To our knowledge, this is the first sparsity-dependent generalization result via localized Rademacher complexity. As a by-product, if the bias initialization is chosen to be zero, the width requirement improves the previous bound for the shallow networks' generalization. Lastly, since the generalization bound has dependence on the smallest eigenvalue of the limiting NTK and the bounds from previous works yield vacuous generalization, this work further studies the least eigenvalue of the limiting NTK. Surprisingly, while it is not shown that trainable biases are necessary, trainable bias helps to identify a nice data-dependent region where a much finer analysis of the NTK's smallest eigenvalue can be conducted, which leads to a much sharper lower bound than the previously known worst-case bound and, consequently, a non-vacuous generalization bound. Experimental evaluation is provided to evaluate our results.

## 1 INTRODUCTION

The literature of sparse neural networks can be dated back to the early work of LeCun et al. (1989) where they showed that a fully-trained neural network can be pruned to preserve generalization. Recently, training sparse neural networks has been receiving increasing attention since the discovery of the lottery ticket hypothesis (Frankle & Carbin, 2018). In their work, they showed that if we repeatedly train and prune a neural network and then rewind the weights to the initialization, we are able to find a sparse neural network that can be trained to match the performance of its dense counterpart. However, this method is more of a proof of concept and is computationally expensive for any practical purposes. Nonetheless, this inspires further interest in the machine learning community to develop efficient methods to find the sparse pattern at the initialization such that the performance of the sparse network matches the dense network after training (Lee et al., 2018; Wang et al., 2019; Tanaka et al., 2020; Liu & Zenke, 2020; Chen et al., 2021; He et al., 2017; Liu et al., 2021b).

On the other hand, instead of trying to find some desired sparsity patterns at the initialization, another line of research has been focusing on inducing the sparsity pattern naturally and then creatively utilizing such sparse structure via high-dimensional geometric data structures as well as sketching or even quantum algorithms to speedup per-step gradient descent training (Song et al., 2021a;b; Hu et al., 2022; Gao et al., 2022). In this line of theoretical studies, the sparsity is induced by shifted

ReLU which is the same as initializing the bias of the network's linear layer to some large constant instead of zero and holding the bias fixed throughout the entire training. By the concentration of Gaussian, at the initialization, the total number of activated neurons (i.e., ReLU will output some non-zero value) will be *sublinear* in the total number $m$ of neurons, as long as the bias is initialized to be $C\sqrt{\log m}$ for some appropriate constant $C$. We call this ***sparsity-inducing initialization***. If the network is in the NTK regime, each neuron weight will exhibit microscopic change after training, and thus the sparsity can be preserved throughout the entire training process. Therefore, during the entire training process, only a sublinear number of the neuron weights need to be updated, which can significantly speedup the training process.

The focus of this work is along the above line of theoretical studies of sparsely trained overparameterized neural networks and address the two main research limitations in the aforementioned studies. (1) The bias parameters used in the previous works are not trainable, contrary to what people are doing in practice. (2) The previous works only provided the convergence guarantee, while lacking the **generalization performance** which is of the central interest in deep learning theory. Thus, our study will fill the above important gaps, by providing a comprehensive study of training one-hidden-layer sparsely activated neural networks in the NTK regime with (a) trainable biases incorporated in the analysis; (b) finer analysis of the convergence; and (c) first generalization bound for such sparsely activated neural networks after training with sharp bound on the restricted smallest eigenvalue of the limiting NTK. We further elaborate our technical contributions are follows:

1. **Convergence.** Theorem 3.1 provides the required width to ensure that gradient descent can drive the training error towards zero at a linear rate. Our convergence result contains two novel ingredients compared to the existing study. (1) Our analysis handles trainable bias, and shows that even though the biases are allowed to be updated from its initialization, the network's activation remains sparse during the entire training. This relies on our development of a new result showing that the change of bias is also diminishing with a $O(1/\sqrt{m})$ dependence on the network width $m$. (2) A finer analysis is provided such that the required network width to ensure the convergence can be much smaller, with an improvement upon the previous result by a factor of $\widetilde{\Theta}(n^{8/3})$ under appropriate bias initialization, where $n$ is the sample size. This relies on our novel development of (1) a better characterization of the activation flipping probability via an analysis of the Gaussian anti-concentration based on the location of the strip and (2) a finer analysis of the initial training error.

2. **Generalization.** Theorem 3.8 studies the generalization of the network after gradient descent training where we characterize how the network width should depend on activation sparsity, which lead to a sparsity-dependent localized Rademacher complexity and a generalization bound matching previous analysis (up to logarithmic factors). To our knowledge, this is the first sparsity-dependent generalization result via localized Rademacher complexity. In addition, compared with previous works, our result yields a better width's dependence by a factor of $n^{10}$. This relies on (1) the usage of symmetric initialization and (2) a finer analysis of the weight matrix change in Frobenius norm in Lemma 3.13.

3. **Restricted Smallest Eigenvalue.** Theorem 3.8 shows that the generalization bound heavily depends on the smallest eigenvalue $\lambda_{\min}$ of the limiting NTK. However, the previously known worst-case lower bounds on $\lambda_{\min}$ under data separation have a $1/n^2$ explicit dependence in (Oymak & Soltanolkotabi, 2020; Song et al., 2021a), making the generalization bound vacuous. Instead, our Theorem 3.11 establishes a much sharper lower bound restricted to a data-dependent region, which is sample-size-independent. This hence yields a desirable generalization bound that vanishes as fast as $O(1/\sqrt{n})$, given that the label vector is in this region, which can be done with simple label-shifting.

## 1.1 FURTHER RELATED WORKS

Besides the works mentioned in the introduction, another work related to ours is (Liao & Kyrillidis, 2022) where they also considered training a one-hidden-layer neural network with sparse activation and studied its convergence. However, different from our work, their sparsity is induced by sampling a random mask at each step of gradient descent whereas our sparsity is induced by non-zero initialization of the bias terms. Also, their network has no bias term, and they only focus on studying the training convergence but not generalization. We discuss additional related works here.

**Training Overparameterized Neural Networks.** Over the past few years, a tremendous amount of efforts have been made to study training overparameterized neural networks. A series of works have shown that if the neural network is wide enough (polynomial in depth, number of samples, etc), gradient descent can drive the training error towards zero in a fast rate either explicitly (Du et al., 2018; 2019; Ji & Telgarsky, 2019) or implicitly (Allen-Zhu et al., 2019; Zou & Gu, 2019; Zou et al., 2020) using the neural tangent kernel (NTK) (Jacot et al., 2018). Further, under some conditions, the networks can generalize (Cao & Gu, 2019). Under the NTK regime, the trained neural network can be well-approximated by its first order Taylor approximation from the initialization and Liu et al. (2020) showed that this transition to linearity phenomenon is a result from a diminishing Hessian 2-norm with respect to width. Later on, Frei & Gu (2021) and Liu et al. (2022) showed that closeness to initialization is sufficient but not necessary for gradient descent to achieve fast convergence as long as the non-linear system satisfies some variants of the Polyak-Łojasiewicz condition. On the other hand, although NTK offers good convergence explanation, it contradicts the practice since (1) the neural networks need to be unrealistically wide and (2) the neuron weights merely change from the initialization. As Chizat et al. (2019) pointed out, this "lazy training" regime can be explained by a mere effect of scaling. Other works have considered the mean-field limit (Chizat & Bach, 2018; Mei et al., 2019; Chen et al., 2020), feature learning (Allen-Zhu & Li, 2020; 2022; Shi et al., 2021; Telgarsky, 2022) which allow the weights to travel far away from the initialization.

**Sparse Neural Networks in Practice.** Besides finding a fixed sparse mask at the initialization as we mentioned in introduction, on the other hand, dynamic sparse training allows the sparse mask to be updated during training, e.g., (Mocanu et al., 2018; Mostafa & Wang, 2019; Evci et al., 2020; Jayakumar et al., 2020; Liu et al., 2021a;c;d).

## 2 PRELIMINARIES

**Notations.** We use $\|\cdot\|_2$ to denote vector or matrix 2-norm and $\|\cdot\|_F$ to denote the Frobenius norm of a matrix. When the subscript of $\|\cdot\|$ is unspecified, it is default to be the 2-norm. For matrices $A \in \mathbb{R}^{m \times n_1}$ and $B \in \mathbb{R}^{m \times n_2}$, we use $[A, B]$ to denote the row concatenation of $A, B$ and thus $[A, B]$ is a $m \times (n_1 + n_2)$ matrix. For matrix $X \in \mathbb{R}^{m \times n}$, the row-wise vectorization of $X$ is denoted by $\text{vec}(X) = [x_1, x_2, \ldots, x_m]^\top$ where $x_i$ is the $i$-th row of $X$. For a given integer $n \in \mathbb{N}$, we use $[n]$ to denote the set $\{0, \ldots, n\}$, i.e., the set of integers from $0$ to $n$. For a set $S$, we use $\overline{S}$ to denote the complement of $S$. We use $\mathcal{N}(\mu, \sigma^2)$ to denote the Gaussian distribution with mean $\mu$ and standard deviation $\sigma$. In addition, we use $\widetilde{O}, \widetilde{\Theta}, \widetilde{\Omega}$ to suppress (poly-)logarithmic factors in $O, \Theta, \Omega$.

### 2.1 PROBLEM FORMULATION

Let the training set to be $(X, y)$ where $X = (x_1, x_2, \ldots, x_n) \in \mathbb{R}^{d \times n}$ denotes the feature matrix consisting of $n$ $d$-dimensional vectors, and $y = (y_1, y_2, \ldots, y_n) \in \mathbb{R}^n$ consists of the corresponding $n$ response variables. We assume $\|x_i\|_2 \leq 1$ and $y_i = O(1)$ for all $i \in [n]$. We use one-hidden-layer neural network and consider the regression problem with the square loss function:

$$f(x; W, b) := \frac{1}{\sqrt{m}} \sum_{r=1}^{m} a_r \sigma(\langle w_r, x \rangle - b_r), \qquad L(W, b) := \frac{1}{2} \sum_{i=1}^{n} (f(x_i; W, b) - y_i)^2,$$

where $W \in \mathbb{R}^{m \times d}$ with its $r$-th row being $w_r$, $b \in \mathbb{R}^m$ is a vector with $b_r$ being the bias of $r$-th neuron, $a_r$ is the second layer weight, and $\sigma(\cdot)$ denotes the ReLU activation function. We initialize the neural network by $W_{r,i} \sim \mathcal{N}(0, 1)$ and $a_r \sim \text{Uniform}(\{\pm 1\})$ and $b_r = B$ for some value $B \geq 0$ of choice, for all $r \in [m]$, $i \in [d]$. We train only the parameters $W$ and $b$ (i.e., the linear layer $a_r$ for $r \in [m]$ is not trained) via gradient descent, the update of which are given by

$$w_r(t + 1) = w_r(t) - \eta \frac{\partial L(W(t), b(t))}{\partial w_r}, \qquad b_r(t + 1) = b_r(t) - \eta \frac{\partial L(W(t), b(t))}{\partial b_r}.$$

By the chain rule, we have $\frac{\partial L}{\partial w_r} = \frac{\partial L}{\partial f} \frac{\partial f}{\partial w_r}$. The gradient of the loss with respect to the network is $\frac{\partial L}{\partial f} = \sum_{i=1}^{n} (f(x_i; W, b) - y_i)$ and the network gradients with respect to weights and bias are

$$\frac{\partial f(x; W, b)}{\partial w_r} = \frac{1}{\sqrt{m}} a_r x \mathbb{I}(w_r^\top x \geq b_r), \qquad \frac{\partial f(x; W, b)}{\partial b_r} = -\frac{1}{\sqrt{m}} a_r \mathbb{I}(w_r^\top x \geq b_r),$$

where $\mathbb{I}(\cdot)$ is the indicator function. We further define $H$ as the **NTK** matrix of this network with

$$H_{i,j}(W,b) := \left\langle \frac{\partial f(x_i; W, b)}{\partial W}, \frac{\partial f(x_j; W, b)}{\partial W} \right\rangle + \left\langle \frac{\partial f(x_i; W, b)}{\partial b}, \frac{\partial f(x_j; W, b)}{\partial b} \right\rangle$$

$$= \frac{1}{m} \sum_{r=1}^{m} (\langle x_i, x_j \rangle + 1) \mathbb{I}(w_r^\top x_i \geq b_r, w_r^\top x_j \geq b_r) \tag{1}$$

and the **infinite-width version** $H^\infty(B)$ of the NTK matrix $H$ is given by

$$H_{ij}^\infty(B) := \mathbb{E}_{w \sim \mathcal{N}(0,I)} \left[ (\langle x_i, x_j \rangle + 1) \mathbb{I}(w^\top x_i \geq B, w^\top x_j \geq B) \right].$$

Let $\lambda(B) := \lambda_{\min}(H^\infty(B))$. We define $\mathbb{I}_{r,i}(W,b) := \mathbb{I}(w_r^\top x_i \geq b_r)$ and the matrix $Z(W,b)$ as

$$Z(W,b) := \frac{1}{\sqrt{m}} \begin{bmatrix} \mathbb{I}_{1,1}(W,b) a_1 [x_1^\top, -1]^\top & \dots & \mathbb{I}_{1,n}(W,b) a_1 [x_n^\top, -1]^\top \\ \vdots & \ddots & \vdots \\ \mathbb{I}_{m,1}(W,b) a_m [x_1^\top, -1]^\top & \dots & \mathbb{I}_{m,n}(W,b) a_m [x_n^\top, -1]^\top \end{bmatrix} \in \mathbb{R}^{m(d+1) \times n}.$$

Note that $H(W,b) = Z(W,b)^\top Z(W,b)$. Hence, the gradient descent step can be written as

$$\text{vec}([W,b](t+1)) = \text{vec}([W,b](t)) - \eta Z(t)(f(t) - y)$$

where $[W,b](t) \in \mathbb{R}^{m \times (d+1)}$ denotes the row-wise concatenation of $W(t)$ and $b(t)$ at the $t$-th step of gradient descent, and $Z(t) := Z(W(t), b(t))$.

## 3 MAIN THEORY

### 3.1 CONVERGENCE AND SPARSITY

We present the convergence of gradient descent for the sparsely activated neural networks. Compared to the existing convergence result in (Song et al., 2021a), our study handles the trainable bias with constant initialization in the convergence analysis (which is the first of such a type). Also, our bound is sharper and yields a much smaller bound on the width of neural networks to guarantee the convergence.

**Theorem 3.1** (Convergence). *Let the learning rate $\eta \leq O(\frac{\lambda \exp(B^2)}{n^2})$, and the bias initialization $B \in [0, \sqrt{0.5 \log m}]$. Assume $\lambda(B) = \lambda_0 \exp(-B^2/2)$ for some $\lambda_0 > 0$ independent of $B$. Then, if the network width satisfies $m \geq \widetilde{\Omega}\left(\lambda_0^{-4} n^4 \exp(B^2)\right)$, over the randomness in the initialization,*

$$\mathbb{P}\left[ \forall t : L(W(t), b(t)) \leq (1 - \eta \lambda(B)/4)^t L(W(0), b(0)) \right] \geq 1 - \delta - e^{-\Omega(n)}.$$

This theorem show that the training loss decreases linearly, and its rate depends on the smallest eigenvalue of the NTK. The assumption on $\lambda(B)$ in Theorem 3.1 can be justified by (Song et al., 2021a, Theorem F.1) which shows that under some mild conditions, the NTK's least eigenvalue $\lambda(B)$ is positive and has an $\exp(-B^2/2)$ dependence.

**Remark 3.2.** Theorem 3.1 establishes a much sharper bound on the width of the neural network than previous work to guarantee the linear convergence. To elaborate, our bound only requires $m \geq \widetilde{\Omega}\left(\lambda_0^{-4} n^4 \exp(B^2)\right)$, as opposed to the bound $m \geq \widetilde{\Omega}(\lambda_0^{-4} n^4 B^2 \exp(2B^2))$ in (Song et al., 2021a, Lemma D.9). If we take $B = \sqrt{0.25 \log m}$ (as allowed by the theorem), then our lower bound yields a polynomial improvement by a factor of $\widetilde{\Theta}(n/\lambda_0)^{8/3}$, which implies that the neural network width can be much smaller to achieve the same linear convergence.

**Key ideas in the proof of Theorem 3.1.** The proof mainly consists of developing a novel bound on activation flipping probability and a novel upper bound on initial error, as we elaborate below.

Like previous works, in order to prove convergence, we need to show that the NTK during training is close to its initialization. Inspecting the expression of NTK in Equation (1), observe that the training will affect the NTK by changing the output of each indicator function. We say that the $r$-th neuron flips its activation with respect to input $x_i$ at the $k$-th step of gradient descent if $\mathbb{I}(w_r(k)^\top x_i - b_r(k) > 0) \neq \mathbb{I}(w_r(k-1)^\top x_i - b_r(k-1) > 0)$ for all $r \in [m]$. The central idea is that for each neuron, as long as the weight and bias movement $R_w, R_b$ from its initialization is small, then the probability of activation flipping (with respect to random initialization) should not be large. We first present the bound on the probability that a given neuron flips its activation.

**Lemma 3.3** (Bound on Activation flipping probability). *Let $B \geq 0$ and $R_w, R_b \leq \min\{1/B, 1\}$. Let $\tilde{W} = (\tilde{w}_1, \ldots, \tilde{w}_m)$ be vectors generated i.i.d. from $\mathcal{N}(0, I)$ and $\tilde{b} = (\tilde{b}_1, \ldots, \tilde{b}_m) = (B, \ldots, B)$, and weights $W = (w_1, \ldots, w_m)$ and biases $b = (b_1, \ldots, b_m)$ that satisfy for any $r \in [m]$, $\|\tilde{w}_r - w_r\|_2 \leq R_w$ and $|\tilde{b}_r - b_r| \leq R_b$. Define the event*

$$A_{i,r} = \{\exists w_r, b_r : \|\tilde{w}_r - w_r\|_2 \leq R_w, \ |b_r - \tilde{b}_r| \leq R_b, \ \mathbb{I}(x_i^\top \tilde{w}_r \geq \tilde{b}_r) \neq \mathbb{I}(x_i^\top w_r \geq b_r)\}.$$

*Then, for some constant $c$,*

$$\mathbb{P}[A_{i,r}] \leq c(R_w + R_b)\exp(-B^2/2).$$

(Song et al., 2021a, Claim C.11) presents a $O(\min\{R, \exp(-B^2/2)\})$ bound on $\mathbb{P}[A_{i,r}]$. The reason that their bound involving the min operation is because $\mathbb{P}[A_{i,r}]$ can be bounded by the standard Gaussian tail bound and Gaussian anti-concentration bound separately and then, take the one that is smaller. On the other hand, our bound replaces the min operation by the product which creates a more convenient (and tighter) interpolation between the two bounds. Later, we will show that the maximum movement of neuron weights and biases, $R_w$ and $R_b$, both have a $O(1/\sqrt{m})$ dependence on the network width, and thus our bound offers a $\exp(-B^2/2)$ improvement where $\exp(-B^2/2)$ can be as small as $1/m^{1/4}$ when we take $B = \sqrt{0.5 \log m}$.

**Proof idea of Lemma 3.3.** First notice that $\mathbb{P}[A_{i,r}] = \mathbb{P}_{x \sim \mathcal{N}(0,1)}[|x - B| \leq R_w + R_b]$. Thus, here we are trying to solve a fine-grained Gaussian anti-concentration problem with the strip centered at $B$. The problem with the standard Gaussian anti-concentration bound is that it only provides a worst case bound and, thus, is location-oblivious. Centered in our proof is a nice Gaussian anti-concentration bound based on the location of the strip, which we describe as follows: Let's first assume $B > R_w + R_b$. A simple probability argument yields a bound of $2(R_w + R_b)\frac{1}{\sqrt{2\pi}}\exp(-(B - R_w - R_b)^2)$. Since later in the Appendix we can show that $R_w$ and $R_b$ have a $O(1/\sqrt{m})$ dependence (Lemma A.9 bounds the movement for gradient descent and Lemma A.10 for gradient flow) and we only take $B = O(\sqrt{\log m})$, by making $m$ sufficiently large, we can safely assume that $R_w$ and $R_b$ is sufficiently small. Thus, the probability can be bounded by $O((R_w + R_b)\exp(-B^2/2))$. However, when $B < R_w + R_b$ the above bound no longer holds. But a closer look tells us that in this case $B$ is close to zero, and thus $(R_w + R_b)\frac{1}{\sqrt{2\pi}}\exp(-B^2/2) \approx \frac{R_w + R_b}{\sqrt{2\pi}}$ which yields roughly the same bound as the standard Gaussian anti-concentration.

Next, our proof of Theorem 3.1 develops the following initial error bound.

**Lemma 3.4** (Initial error upper bound). *Let $B > 0$ be the initialization value of the biases and all the weights be initialized from standard Gaussian. Let $\delta \in (0, 1)$ be the failure probability. Then, with probability at least $1 - \delta$ over the randomness in the initialization, we have*

$$L(W(0), b(0)) = O\left(n + n\left(\exp(-B^2/2) + 1/m\right)\log^3(2mn/\delta)\right).$$

(Song et al., 2021a, Claim D.1) gives a rough estimate of the initial error with $O(n(1 + B^2)\log^2(n/\delta)\log(m/\delta))$ bound. When we set $B = C\sqrt{\log m}$ for some constant $C$, our bound improves the previous result by a polylogarithmic factor. The previous bound is not tight in the following two senses: (1) the bias will only decrease the magnitude of the neuron activation instead of increasing and (2) when the bias is initialized as $B$, only roughly $O(\exp(-B^2/2)) \cdot m$ neurons will activate. Thus, we can improve the $B^2$ dependence to $\exp(-B^2/2)$.

By combining the above two improved results, we can prove our convergence result with improved lower bound of $m$ as in Remark 3.2. We provide the complete proof in Appendix A.

Lastly, since the total movement of each neuron's bias has a $O(1/\sqrt{m})$ dependence (shown in Lemma A.9), combining with the number of activated neurons at the initialization, we can show that during the entire training, the number of activated neurons is small.

**Lemma 3.5** (Number of Activated Neurons per Iteration). *Assume the parameter settings in Theorem 3.1. With probability at least $1 - e^{-\Omega(n)}$ over the random initialization, we have*

$$|\mathcal{S}_{\text{on}}(i, t)| = O(m \cdot \exp(-B^2/2))$$

*for all $0 \leq t \leq T$ and $i \in [n]$, where $\mathcal{S}_{\text{on}}(i, t) = \{r \in [m] : w_r(t)^\top x_i \geq b_r(t)\}$.*

## 3.2 GENERALIZATION AND RESTRICTED LEAST EIGENVALUE

In this section, we present the sparsity-dependent generalization of our neural networks after gradient descent training. However, for technical reasons stated in Section 3.3, we use symmetric initialization defined below. Further, we adopt the setting in (Arora et al., 2019) and use a non-degenerate data distribution to make sure the infinite-width NTK is positive definite.

**Definition 3.6** (Symmetric Initialization). *For a one-hidden layer neural network with $2m$ neurons, the network is initialized as the following:*

  *1. For $r \in [m]$, independently initialize $w_r \sim \mathcal{N}(0, I)$ and $a_r \sim \mathrm{Uniform}(\{-1, 1\})$.*

  *2. For $r \in \{m + 1, \ldots, 2m\}$, let $w_r = w_{r-m}$ and $a_r = -a_{r-m}$.*

**Definition 3.7** (($\lambda_0, \delta, n$)-non-degenerate distribution, (Arora et al., 2019)). *A distribution $\mathcal{D}$ over $\mathbb{R}^d \times \mathbb{R}$ is $(\lambda_0, \delta, n)$-non-degenerate, if for $n$ i.i.d. samples $\{(x_i, y_i)\}_{i=1}^n$ from $\mathcal{D}$, with probability $1 - \delta$ we have $\lambda_{\min}(H^\infty(B)) \geq \lambda_0 > 0$.*

**Theorem 3.8.** *Fix a failure probability $\delta \in (0, 1)$ and an accuracy parameter $\epsilon \in (0, 1)$. Suppose the training data $S = \{(x_i, y_i)\}_{i=1}^n$ are i.i.d. samples from a $(\lambda, \delta, n)$-non-degenerate distribution $\mathcal{D}$ defined in Definition 3.7. Assume the one-hidden layer neural network is initialized by symmetric initialization in Definition 3.6. Further, assume the parameter settings in Theorem 3.1 except we let $m \geq \widetilde{\Omega}\left(\lambda(B)^{-6} n^6 \exp(-B^2)\right)$. Consider any loss function $\ell : \mathbb{R} \times \mathbb{R} \to [0, 1]$ that is 1-Lipschitz in its first argument. Then with probability at least $1 - 2\delta - e^{-\Omega(n)}$ over the randomness in symmetric initialization of $W(0) \in \mathbb{R}^{m \times d}$ and $a \in \mathbb{R}^m$ and the training samples, the two layer neural network $f(W(t), b(t), a)$ trained by gradient descent for $t \geq \Omega(\frac{1}{\eta \lambda(B)} \log \frac{n \log(1/\delta)}{\epsilon})$ iterations has empirical Rademacher complexity (see its formal definition in Definition C.1 in Appendix) bounded as*

$$\mathcal{R}_S(\mathcal{F}) \leq \sqrt{\frac{y^\top (H^\infty(B))^{-1} y \cdot 8 \exp(-B^2/2)}{n}} + \tilde{O}\left(\frac{\exp(-B^2/4)}{n^{1/2}}\right)$$

*and the population loss $L_\mathcal{D}(f) = \mathbb{E}_{(x,y) \sim \mathcal{D}}[\ell(f(x), y)]$ can be upper bounded as*

$$L_\mathcal{D}(f(W(t), b(t), a)) \leq \sqrt{\frac{y^\top (H^\infty(B))^{-1} y \cdot 32 \exp(-B^2/2)}{n}} + \tilde{O}\left(\frac{1}{n^{1/2}}\right). \tag{2}$$

To show good generalization, we need a larger width: the second term in the Rademacher complexity bound is diminishing with $m$ and to make this term $O(1/\sqrt{n})$, the width needs to have $(n/\lambda(B))^6$ dependence as opposed to $(n/\lambda(B))^4$ for convergence. Now, at the first glance of our generalization result, it seems we can make the Rademacher complexity arbitrarily small by increasing $B$. Recall from the discussion of Theorem 3.1 that the smallest eigenvalue of $H^\infty(B)$ also has an $\exp(-B^2/2)$ dependence. Thus, in the worst case, the $\exp(-B^2/2)$ factor gets canceled and sparsity will not hurt the network's generalization.

Before we present the proof, we make a corollary of Theorem 3.8 for the zero-initialized bias case.

**Corollary 3.9.** *Take the same setting as in Theorem 3.8 except now the biases are initialized as zero, i.e., $B = 0$. Then, if we let $m \geq \widetilde{\Omega}(\lambda(0)^{-6} n^6)$, the empirical Rademacher complexity and population loss are both bounded by*

$$\mathcal{R}_S(\mathcal{F}), \ L_\mathcal{D}(f(W(t), b(t), a)) \leq \sqrt{\frac{y^\top (H^\infty(0))^{-1} y \cdot 32}{n}} + \tilde{O}\left(\frac{1}{n^{1/2}}\right).$$

Corollary 3.9 requires the network width $m \geq \widetilde{\Omega}((n/\lambda(0))^6)$ which significantly improves upon the previous result in (Song & Yang, 2019, Theorem G.7) $m \geq \widetilde{\Omega}(n^{16} \mathrm{poly}(1/\lambda(0)))$ (including the dependence on the rescaling factor $\kappa$) which is a much wider network.

**Generalization Bound via Least Eigenvalue.** Note that in Theorem 3.8, the worst case of the first term in the generalization bound in Equation (2) is given by $\widetilde{O}(\sqrt{1/(\lambda(B) \cdot n)})$. Hence, the least eigenvalue $\lambda(B)$ of the NTK matrix can significantly affect the generalization bound. Previous works (Oymak & Soltanolkotabi, 2020; Song et al., 2021a) established lower bounds on $\lambda(B)$ with an explicit $1/n^2$ dependence on $n$ under the $\delta$ data separation assumption (see Theorem 3.11), which

clearly makes a vacuous generalization bound of $\widetilde{O}(\sqrt{n})$. This thus motivates us to provide a tighter bound (desirably independent on $n$) on the least eigenvalue of the infinite-width NTK in order to make the generalization bound in Theorem 3.8 valid and useful. However, it turns out that there are major difficulties in proving a better lower bound in the general case and thus, we are only able to present a better lower bound when we restrict the domain to some (data-dependent) regions.

**Definition 3.10** (Data-dependent Region). *Let $p_{ij} = \mathbb{P}_{w \sim \mathcal{N}(0,I)}[w^\top x_i \geq B, \ w^\top x_j \geq B]$ for $i \neq j$. Define the (data-dependent) region $\mathcal{R} = \{a \in \mathbb{R}^n : \sum_{i \neq j} a_i a_j p_{ij} \geq \min_{i' \neq j'} p_{i'j'} \sum_{i \neq j} a_i a_j\}$.*

Notice that $\mathcal{R}$ is non-empty for any input data-set since $\mathbb{R}_+^n \subset \mathcal{R}$ where $\mathbb{R}_+^n$ denotes the set of vectors with non-negative entries, and $\mathcal{R} = \mathbb{R}^n$ if $p_{ij} = p_{i'j'}$ for all $i \neq i', j \neq j'$.

**Theorem 3.11** (Restricted Least Eigenvalue). *Let $X = (x_1, \ldots, x_n)$ be points in $\mathbb{R}^d$ with $\|x_i\|_2 = 1$ for all $i \in [n]$ and $w \sim \mathcal{N}(0, I_d)$. Suppose that there exists $\delta \in [0, \sqrt{2}]$ such that*

$$\min_{i \neq j \in [n]} (\|x_i - x_j\|_2, \|x_i + x_j\|_2) \geq \delta.$$

*Let $B \geq 0$. Consider the minimal eigenvalue of $H^\infty$ over the data-dependent region $\mathcal{R}$ defined above, i.e., let $\lambda := \min_{\|a\|_2 = 1, \ a \in \mathcal{R}} a^\top H^\infty a$. Then, $\lambda \geq \max(0, \lambda')$ where*

$$\lambda' \geq \max \left( \frac{1}{2} - \frac{B}{\sqrt{2\pi}}, \ \left( \frac{1}{B} - \frac{1}{B^3} \right) \frac{e^{-B^2/2}}{\sqrt{2\pi}} \right) - e^{-B^2/(2-\delta^2/2)} \frac{\pi - \arctan\left( \frac{\delta\sqrt{1-\delta^2/4}}{1-\delta^2/2} \right)}{2\pi}. \quad (3)$$

To demonstrate the usefulness of our result, if we take the bias initialization $B = 0$ in Equation (3), this bound yields $1/(2\pi) \cdot \arctan((\delta\sqrt{1 - \delta^2/4})/(1 - \delta^2/2)) \approx \delta/(2\pi)$, when $\delta$ is close to 0 whereas (Song et al., 2021a) yields a bound of $\delta/n^2$. On the other hand, if the data has maximal separation, i.e., $\delta = \sqrt{2}$, we get a $\max\left( \frac{1}{2} - \frac{B}{\sqrt{2\pi}}, \ \left( \frac{1}{B} - \frac{1}{B^3} \right) \frac{e^{-B^2/2}}{\sqrt{2\pi}} \right)$ lower bound, whereas (Song et al., 2021a) yields a bound of $\exp(-B^2/2)\sqrt{2}/n^2$. Connecting to our convergence result in Theorem 3.1, if $f(t) - y \in \mathcal{R}$, then the error can be reduced at a much faster rate than the (pessimistic) rate with $1/n^2$ dependence in the previous studies as long as the error vector lies in the region.

**Remark 3.12.** The lower bound on the restricted smallest eigenvalue $\lambda$ in Theorem 3.11 is **independent on** $n$, which makes that the generalization bound in Theorem 3.8 vanishes as fast as $O(1/\sqrt{n})$. Such a lower bound is much sharper than the previous results with a $1/n^2$ explicit dependence which yields vacuous generalization. This improvement relies on a fact that the label vector should lie in the region $\mathcal{R}$, which can be justified by a simple label-shifting strategy as follows. Since $\mathbb{R}_+^n \subset \mathcal{R}$, the condition can be easily achieved by training the neural network on the shifted labels $y + C$ (with appropriate broadcast) where $C$ is a constant such that $\min_i y_i + C \geq 0$.

Careful readers may notice that in the proof of Theorem 3.11 in Appendix B, the restricted least eigenvalue on $\mathbb{R}_+^n$ is always positive even if the data separation is zero. However, we would like to point out that the generalization bound in Theorem 3.8 is meaningful only when the training is successful: when the data separation is zero, the limiting NTK is no longer positive definite and the training loss cannot be minimized toward zero.

### 3.3 Key Ideas in the Proof of Theorem 3.8

Since each neuron weight and bias move little from their initialization, a natural approach is to bound the generalization via localized Rademacher complexity. After that, we can apply appropriate concentration bounds to derive generalization. The main effort of our proof is devoted to bounding the weight movement to bound the localized Rademacher complexity. If we directly take the setting in Theorem 3.1 and compute the network's localized Rademacher complexity, we will encounter a non-diminishing (with the number of samples $n$) term which can be as large as $O(\sqrt{n})$ since the network outputs non-zero values at the initialization. Arora et al. (2019) and Song & Yang (2019) resolved this issue by initializing the neural network weights instead by $\mathcal{N}(0, \kappa^2 I)$ to force the neural network output something close to zero at the initialization. The magnitude of $\kappa$ is chosen to balance different terms in the Rademacher complexity bound in the end. Similar approach can also be adapted to our case by initializing the weights by $\mathcal{N}(0, \kappa^2 I)$ and the biases by $\kappa B$. However, the drawback of such an approach is that the effect of $\kappa$ to all the previously established results

for convergence need to be carefully tracked or derived. In particular, in order to guarantee convergence, the neural network's width needs to have a polynomial dependence on $1/\kappa$ where $1/\kappa$ has a polynomial dependence on $n$ and $1/\lambda$, which means their network width needs to be larger to compensate for the initialization scaling. We resolve this issue by symmetric initialization Definition 3.6 which yields no effect (up to constant factors) on previously established convergence results, see (Munteanu et al., 2022). Symmetric initialization allows us to organically combine the results derived for convergence to be reused for generalization, which leads to a more succinct analysis. Further, we replace the $\ell_1$-$\ell_2$ norm upper bound by finer inequalities in various places in the original analysis. All these improvements lead to the following upper bound of the weight matrix change in Frobenius norm. Further, combining our sparsity-inducing initialization, we present our sparsity-dependent Frobenius norm bound on the weight matrix change.

**Lemma 3.13.** *Assume the one-hidden layer neural network is initialized by symmetric initialization in Definition 3.6. Further, assume the parameter settings in Theorem 3.1. Then with probability at least $1 - \delta - e^{-\Omega(n)}$ over the random initialization, we have for all $t \geq 0$,*

$$\|[W,b](t) - [W,b](0)\|_F \leq \sqrt{y^\top (H^\infty)^{-1} y} + O\left(\frac{n}{\lambda}\left(\frac{\exp(-B^2/2)\log(n/\delta)}{m}\right)^{1/4}\right)$$

$$+ O\left(\frac{n\sqrt{R\exp(-B^2/2)}}{\lambda}\right) + \frac{n}{\lambda^2} \cdot O\left(\exp(-B^2/4)\sqrt{\frac{\log(n^2/\delta)}{m}} + R\exp(-B^2/2)\right)$$

*where $R = R_w + R_b$ denote the maximum magnitude of neuron weight and bias change.*

By Lemma A.9 and Lemma A.11 in the Appendix, we have $R = \widetilde{O}(\frac{n}{\lambda\sqrt{m}})$. Plugging in and setting $B = 0$, we get $\|[W,b](t) - [W,b](0)\|_F \leq \sqrt{y^\top (H^\infty)^{-1} y} + \widetilde{O}(\frac{n}{\lambda m^{1/4}} + \frac{n^{3/2}}{\lambda^{3/2} m^{1/4}} + \frac{n}{\lambda^2 \sqrt{m}} + \frac{n^2}{\lambda^3 \sqrt{m}})$. On the other hand, taking $\kappa = 1$, (Song & Yang, 2019, Lemma G.6) yields a bound of $\|W(t) - W(0)\|_F \leq \sqrt{y^\top (H^\infty)^{-1} y} + \widetilde{O}(\frac{n}{\lambda} + \frac{n^{7/2}\text{poly}(1/\lambda)}{m^{1/4}})$. Notice that the $\widetilde{O}(\frac{n}{\lambda})$ term has no dependence on $1/m$ and is removed by symmetric initialization in our analysis and we improve the upper bound's dependence on $n$ by a factor of $n^2$.

We defer the full proof of Theorem 3.8 and Lemma 3.13 to Appendix C.

### 3.4 Key Ideas in the Proof of Theorem 3.11

In this section, we analyze the smallest eigenvalue $\lambda := \lambda_{\min}(H^\infty)$ of the limiting NTK $H^\infty$ with $\delta$ data separation. We first note that $H^\infty \succeq \mathbb{E}_{w \sim \mathcal{N}(0,I)}\left[\mathbb{I}(Xw \geq B)\mathbb{I}(Xw \geq B)^\top\right]$ and for a fixed vector $a$, we are interested in the lower bound of $\mathbb{E}_{w \sim \mathcal{N}(0,I)}[|a^\top \mathbb{I}(Xw \geq B)|^2]$. In previous works, Oymak & Soltanolkotabi (2020) showed a lower bound $\Omega(\delta/n^2)$ for zero-initialized bias, and later Song et al. (2021a) generalized this result to a lower bound $\Omega(e^{-B^2/2}\delta/n^2)$ for non-zero initialized bias. Both lower bounds have a dependence of $1/n^2$. Their approach is by using an intricate Markov's inequality argument and then proving an lower bound of $\mathbb{P}[|a^\top \mathbb{I}(Xw \geq B)| \geq c\|a\|_\infty]$. The lower bound is proved by only considering the contribution from the largest coordinate of $a$ and treating all other values as noise. It is non-surprising that the lower bound has a factor of $1/n$ since $a$ can have identical entries. On the other hand, the diagonal entries can give a $\exp(-B^2/2)$ upper bound and thus there is a $1/n^2$ gap between the two. Now, we give some evidence suggesting the $1/n^2$ dependence may not be tight in some cases. Consider the following scenario: Assume $n \ll d$ and the data set is orthonormal. For a fixed $a$, we have

$$a^\top \mathbb{E}_{w \sim \mathcal{N}(0,I)}\left[\mathbb{I}(Xw \geq B)\mathbb{I}(Xw \geq B)^\top\right] a$$
$$= \sum_{i,j \in [n]} a_i a_j \, \mathbb{P}[w^\top x_i \geq B, \, w^\top x_j \geq B] = p_0 \|a\|_2^2 + p_1 \sum_{i \neq j} a_i a_j$$
$$= p_0 - p_1 + p_1 \left(\sum_i a_i\right)^2 > p_0 - p_1$$

where $p_0, p_1 \in [0,1]$ are defined such that due to the spherical symmetry of the standard Gaussian we are able to let $p_0 = \mathbb{P}[w^\top x_i \geq B], \, \forall i \in [n]$ and $p_1 = \mathbb{P}[w^\top x_i \geq B, w^\top x_j \geq B], \, \forall i,j \in [n], \, i \neq j$. Notice that $p_0 > p_1$. Since this is true for all $a \in \mathbb{R}^n$, we get a lower bound of $p_0 - p_1$ with no explicit dependence on $n$ and this holds for all $n \leq d$. When $d$ is large and $n = d/2$, this bound is better than previous bound by a factor of $\Theta(1/d^2)$. However, it turns out that the product

terms with $i \neq j$ above creates major difficulties in analyzing the general case. Due to such technical difficulties, we are only able to prove a better lower bound by utilizing the extra constant factor in the NTK thanks to the trainable bias, when we restrict the domain to some data-dependent region. We defer the proof of Theorem 3.11 to Appendix B.

## 4 EXPERIMENTS

In this section, we study how the activation sparsity patterns of multi-layer neural networks change during training when the bias parameters are initialized as non-zero.

**Settings.** We train a 6-layer multi-layer perceptron (MLP) of width 1024 with trainable bias terms on MNIST image classification (LeCun et al., 2010). The biases of the fully-connected layers are initialized as $0, -0.5$ and $-1$. For the weights in the linear layer, we use Kaiming Initialization (He et al., 2015) which is sampled from an appropriately scaled Gaussian distribution. The traditional MLP architecture only has linear layers with ReLU activation. However, we found out that using the sparsity-inducing initialization, the magnitude of the activation will decrease geometrically layer-by-layer, which leads to vanishing gradients and that the network cannot be trained. Thus, we made a slight modification to the MLP architecture to include an extra Batch Normalization after ReLU to normalize the activation. Our MLP implementation is based on (Zhu et al., 2021). We train the neural network by stochastic gradient descent with a small learning rate 5e-3 to make sure the training is in the NTK regime. The sparsity is measured as the total number of activated neurons (i.e., ReLU outputs some positive values) divided by total number of neurons, averaged over every SGD batch. We plot how the sparsity patterns changes for different layers during training.

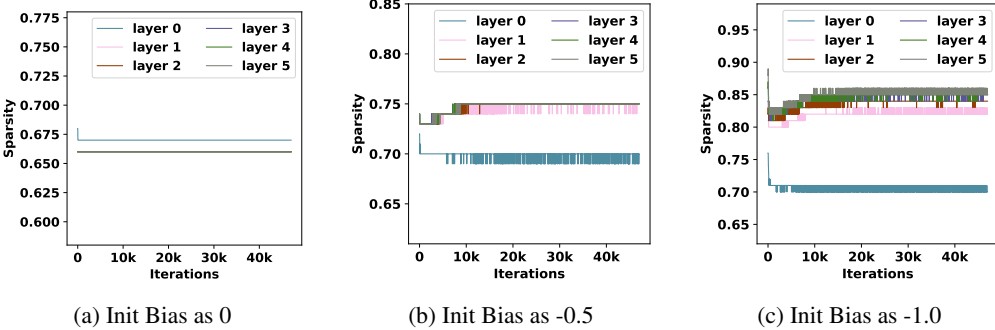

(a) Init Bias as 0          (b) Init Bias as -0.5          (c) Init Bias as -1.0

Figure 1: Sparsity pattern on different layers across different training iterations for three different bias initialization. The $x$ and $y$ axis denote the iteration number and sparsity level, respectively. The models can achieve $97.9\%, 97.7\%$ and $97.3\%$ accuracy after training, respectively. Note that, in Figure (a), the lines of layers 1-5 overlap together except layer 0.

**Observation and Implication.** As demonstrated at Figure 1, when we initialize the bias with three different values, the sparsity patterns are stable across all layers during training: when the bias is initialized as $0$ and $-0.5$, the sparsity change is within $2.5\%$; and when the bias is initialized as $-1.0$, the sparsity change is within $10\%$. Meanwhile, by increasing the initialization magnitude for bias, the sparsity level increases with only marginal accuracy dropping. This implies that our theory can be extended to the multi-layer setting (with some extra care for coping with vanishing gradient) and multi-layer neural networks can also benefit from the sparsity-inducing initialization and enjoy reduction of computational cost. Another interesting observation is that the input layer (layer 0) has a different sparsity pattern from other layers while all the rest layers behave similarly.

## 5 DISCUSSION

In this work, we study training one-hidden-layer overparameterized ReLU networks in the NTK regime with its biases being trainable and initialized as some constants rather than zero. We showed sparsity-dependent results on convergence, restricted least eigenvalue and generalization. A future direction is to generalize our analysis to multi-layer neural networks. In practice, label shifting is unnecessary for achieving good generalization. An open problem is whether it is possible to improve the dependence on the sample size of the lower bound of the infinite-width NTK's least eigenvalue, or even whether a lower bound purely dependent on the data separation is possible so that the generalization bound is no longer vacuous for all labels.

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

# A CONVERGENCE

**Notation simplification.** Since the smallest eigenvalue of the limiting NTK appeared in this proof all has dependence on the bias initialization parameter $B$, for the ease of notation of our proof, we suppress its dependence on $B$ and use $\lambda$ to denote $\lambda := \lambda(B) = \lambda_{\min}(H^\infty(B))$.

## A.1 DIFFERENCE BETWEEN LIMIT NTK AND SAMPLED NTK

**Lemma A.1.** *For a given bias vector $b \in \mathbb{R}^m$ with $b_r \geq 0$, $\forall r \in [m]$, the limit NTK $H^\infty$ and the sampled NTK $H$ are given as*

$$H_{ij}^\infty := \mathbb{E}_{w \sim \mathcal{N}(0,I)} \left[ (\langle x_i, x_j \rangle + 1) \mathbb{I}(w_r^\top x_i \geq b_r, w_r^\top x_j \geq b_r) \right],$$

$$H_{ij} := \frac{1}{m} \sum_{r=1}^m (\langle x_i, x_j \rangle + 1) \mathbb{I}(w_r^\top x_i \geq b_r, w_r^\top x_j \geq b_r).$$

*Let's define $\lambda := \lambda_{\min}(H^\infty)$ and assume $\lambda > 0$. If the network width $m = \Omega(\lambda^{-1} n \cdot \log(n/\delta))$, then*

$$\mathbb{P}\left[ \lambda_{\min}(H) \geq \frac{3}{4}\lambda \right] \geq 1 - \delta.$$

*Proof.* Let $H_r := \frac{1}{m} \widetilde{X}(w_r)^\top \widetilde{X}(w_r)$, where $\widetilde{X}(w_r) \in \mathbb{R}^{(d+1) \times n}$ is defined as

$$\widetilde{X}(w_r) := [\mathbb{I}(w_r^\top x_1 \geq b) \cdot (x_1, 1), \ldots, \mathbb{I}(w_r^\top x_n \geq b) \cdot (x_n, 1)],$$

where $(x_i, 1)$ denotes appending the vector $x_i$ by 1. Hence $H_r \succeq 0$. Since for each entry $H_{ij}$ we have

$$(H_r)_{ij} = \frac{1}{m}(\langle x_i, x_j \rangle + 1)\mathbb{I}(w_r^\top x_i \geq b_r, w_r^\top x_j \geq b_r) \leq \frac{1}{m}(\langle x_i, x_j \rangle + 1) \leq \frac{2}{m},$$

and naively, we can upper bound $\|H_r\|_2$ by:

$$\|H_r\|_2 \leq \|H_r\|_F \leq \sqrt{n^2 \frac{4}{m^2}} = \frac{2n}{m}.$$

Then $H = \sum_{r=1}^m H_r$ and $\mathbb{E}[H] = H^\infty$. Hence, by the Matrix Chernoff Bound in Lemma D.2 and choosing $m = \Omega(\lambda^{-1} n \cdot \log(n/\delta))$, we can show that

$$\mathbb{P}\left[ \lambda_{\min}(H) \leq \frac{3}{4}\lambda \right] \leq n \cdot \exp\left( -\frac{1}{16}\lambda/(4n/m) \right)$$

$$= n \cdot \exp\left( -\frac{\lambda m}{64n} \right)$$

$$\leq \delta.$$

$\square$

**Lemma A.2.** *Assume $m = n^{O(1)}$ and $\exp(B^2/2) = O(\sqrt{m})$ where we recall that $B$ is the initialization value of the biases. With probability at least $1 - \delta$, we have $\|H(0) - H^\infty\|_F \leq 4n \exp(-B^2/4) \sqrt{\frac{\log(n^2/\delta)}{m}}$.*

*Proof.* First, we have $\mathbb{E}[((\langle x_i, x_j \rangle + 1)\mathbb{I}_{r,i}(0)\mathbb{I}_{r,j}(0))^2] \leq 4 \exp(-B^2/2)$. Then, by Bernstein's inequality in Lemma D.1, with probability at least $1 - \delta/n^2$,

$$|H_{ij}(0) - H_{ij}^\infty| \leq 2\exp(-B^2/4)\sqrt{2\frac{\log(n^2/\delta)}{m}} + 2\frac{2}{m}\log(n^2/\delta) \leq 4\exp(-B^2/4)\sqrt{\frac{\log(n^2/\delta)}{m}}.$$

By a union bound, the above holds for all $i, j \in [n]$ with probability at least $1 - \delta$, which implies

$$\|H(0) - H^\infty\|_F \leq 4n \exp(-B^2/4)\sqrt{\frac{\log(n^2/\delta)}{m}}.$$

$\square$

A.2 BOUNDING THE NUMBER OF FLIPPED NEURONS

**Definition A.3** (No-flipping set). *For each $i \in [n]$, let $S_i \subset [m]$ denote the set of neurons that are never flipped during the entire training process,*

$$S_i := \{r \in [m] : \forall t \in [T] \; \text{sign}(\langle w_r(t), x_i \rangle - b_r(t)) = \text{sign}(\langle w_r(0), x_i \rangle - b_r(0))\}.$$

*Thus, the flipping set is $\overline{S}_i$ for $i \in [n]$.*

**Lemma A.4** (Bound on flipping probability). *Let $B \geq 0$ and $R_w, R_b \leq \min\{1/B, 1\}$. Let $\tilde{W} = (\tilde{w}_1, \ldots, \tilde{w}_m)$ be vectors generated i.i.d. from $\mathcal{N}(0, I)$ and $\tilde{b} = (\tilde{b}_1, \ldots, \tilde{b}_m) = (B, \ldots, B)$, and weights $W = (w_1, \ldots, w_m)$ and biases $b = (b_1, \ldots, b_m)$ that satisfy for any $r \in [m]$, $\|\tilde{w}_r - w_r\|_2 \leq R_w$ and $|\tilde{b}_r - b_r| \leq R_b$. Define the event*

$$A_{i,r} = \{\exists w_r, b_r : \|\tilde{w}_r - w_r\|_2 \leq R_w, \; |b_r - \tilde{b}_r| \leq R_b, \; \mathbb{I}(x_i^\top \tilde{w}_r \geq \tilde{b}_r) \neq \mathbb{I}(x_i^\top w_r \geq b_r)\}.$$

*Then,*

$$\mathbb{P}[A_{i,r}] \leq c(R_w + R_b) \exp(-B^2/2)$$

*for some constant c.*

*Proof.* Notice that the event $A_{i,r}$ happens if and only if $|\tilde{w}_r^\top x_i - \tilde{b}_r| < R_w + R_b$. First, if $B > 1$, then by Lemma D.3, we have

$$\mathbb{P}[A_{i,r}] \leq (R_w + R_b)\frac{1}{\sqrt{2\pi}} \exp(-(B - R_w - R_b)^2/2) \leq c_1(R_w + R_b)\exp(-B^2/2)$$

for some constant $c_1$. If $0 \leq B < 1$, then the above analysis doesn't hold since it is possible that $B - R_w - R_b \leq 0$. In this case, the probability is at most $\mathbb{P}[A_{i,r}] \leq 2(R_w + R_b)\frac{1}{\sqrt{2\pi}} \exp(-0^2/2) = \frac{2(R_w+R_b)}{\sqrt{2\pi}}$. However, since $0 \leq B < 1$ in this case, we have $\exp(-1^2/2) \leq \exp(-B^2/2) \leq \exp(-0^2/2)$. Therefore, $\mathbb{P}[A_{i,r}] \leq c_2(R_w + R_b)\exp(-B^2/2)$ for $c_2 = \frac{2\exp(1/2)}{\sqrt{2\pi}}$. Take $c = \max\{c_1, c_2\}$ finishes the proof. $\square$

**Corollary A.5.** *Let $B > 0$ and $R_w, R_b \leq \min\{1/B, 1\}$. Assume that $\|w_r(t) - w_r(0)\|_2 \leq R_w$ and $|b_r(t) - b_r(0)| \leq R_b$ for all $t \in [T]$. For $i \in [n]$, the flipping set $\overline{S}_i$ satisfies that*

$$\mathbb{P}[r \in \overline{S}_i] \leq c(R_w + R_b)\exp(-B^2/2)$$

*for some constant c, which implies*

$$\mathbb{P}[\forall i \in [n] : |\overline{S}_i| \leq 2mc(R_w + R_b)\exp(-B^2/2)] \geq 1 - n \cdot \exp\left(-\frac{2}{3}mc(R_w + R_b)\exp(-B^2/2)\right).$$

*Proof.* The proof is by observing that $\mathbb{P}[r \in \overline{S}_i] \leq \mathbb{P}[A_{i,r}]$. Then, by Bernstein's inequality,

$$\mathbb{P}[|\overline{S}_i| > t] \leq \exp\left(-\frac{t^2/2}{mc(R_w + R_b)\exp(-B^2/2) + t/3}\right).$$

Take $t = 2mc(R_w + R_b)\exp(-B^2/2)$ and a union bound over $[n]$, we have

$$\mathbb{P}[\forall i \in [n] : |\overline{S}_i| \leq 2mc(R_w + R_b)\exp(-B^2/2)] \geq 1 - n \cdot \exp\left(-\frac{2}{3}mc(R_w + R_b)\exp(-B^2/2)\right).$$

$\square$

A.3 BOUNDING NTK IF PERTURBING WEIGHTS AND BIASES

**Lemma A.6.** *Assume $\lambda > 0$. Let $B > 0$ and $R_b, R_w \leq \min\{1/B, 1\}$. Let $\tilde{W} = (\tilde{w}_1, \ldots, \tilde{w}_m)$ be vectors generated i.i.d. from $\mathcal{N}(0, I)$ and $\tilde{b} = (\tilde{b}_1, \ldots, \tilde{b}_m) = (B, \ldots, B)$. For any set of weights $W = (w_1, \ldots, w_m)$ and biases $b = (b_1, \ldots, b_m)$ that satisfy for any $r \in [m]$, $\|\tilde{w}_r - w_r\|_2 \leq R_w$ and $|\tilde{b}_r - b_r| \leq R_b$, we define the matrix $H(W, b) \in \mathbb{R}^{n \times n}$ by*

$$H_{ij}(W, b) = \frac{1}{m} \sum_{r=1}^{m} (\langle x_i, x_j \rangle + 1)\mathbb{I}(w_r^\top x_i \geq b_r, w_r^\top x_j \geq b_r).$$

*It satisfies that for some small positive constant c,*

1. *With probability at least $1 - n^2 \exp\left(-\frac{2}{3}cm(R_w + R_b)\exp(-B^2/2)\right)$, we have*

$$\left\|H(\tilde{W}, \tilde{b}) - H(W, b)\right\|_F \leq n \cdot 8c(R_w + R_b)\exp(-B^2/2),$$
$$\left\|Z(\tilde{W}, \tilde{b}) - Z(W, b)\right\|_F \leq \sqrt{n \cdot 8c(R_w + R_b)\exp(-B^2/2)}.$$

2. *With probability at least $1 - \delta - n^2 \exp\left(-\frac{2}{3}cm(R_w + R_b)\exp(-B^2/2)\right)$,*

$$\lambda_{\min}(H(W, b)) > 0.75\lambda - n \cdot 8c(R_w + R_b)\exp(-B^2/2).$$

*Proof.* We have

$$\left\|Z(W, b) - Z(\tilde{W}, \tilde{b})\right\|_F^2 = \sum_{i \in [n]} \left(\frac{2}{m} \sum_{r \in [m]} \left(\mathbb{I}(w_r^\top x_i \geq b_r) - \mathbb{I}(\tilde{w}_r^\top x_i \geq \tilde{b}_r)\right)^2\right)$$

$$= \sum_{i \in [n]} \left(\frac{2}{m} \sum_{r \in [m]} t_{r,i}\right)$$

and

$$\left\|H(W, b) - H(\tilde{W}, \tilde{b})\right\|_F^2$$
$$= \sum_{i \in [n],\, j \in [n]} (H_{ij}(W, b) - H_{ij}(\tilde{W}, \tilde{b}))^2$$
$$\leq \frac{4}{m^2} \sum_{i \in [n],\, j \in [n]} \left(\sum_{r \in [m]} |\mathbb{I}(w_r^\top x_i \geq b_r, w_r^\top x_j \geq b_r) - \mathbb{I}(\tilde{w}_r^\top x_i \geq \tilde{b}_r, \tilde{w}_r^\top x_j \geq \tilde{b}_r)|\right)^2$$
$$= \frac{4}{m^2} \sum_{i,j \in [n]} \left(\sum_{r \in [m]} s_{r,i,j}\right)^2,$$

where we define

$$s_{r,i,j} := |\mathbb{I}(w_r^\top x_i \geq b_r, w_r^\top x_j \geq b_r) - \mathbb{I}(\tilde{w}_r^\top x_i \geq \tilde{b}_r, \tilde{w}_r^\top x_j \geq \tilde{b}_r)|,$$
$$t_{r,i} := (\mathbb{I}(w_r^\top x_i \geq b_r) - \mathbb{I}(\tilde{w}_r^\top x_i \geq \tilde{b}_r))^2.$$

Notice that $t_{r,i} = 1$ only if the event $A_{i,r}$ happens (recall the definition of $A_{i,r}$ in Lemma A.4) and $s_{r,i,j} = 1$ only if the event $A_{i,r}$ or $A_{j,r}$ happens. Thus,

$$\sum_{r \in [m]} t_{r,i} \leq \sum_{r \in [m]} \mathbb{I}(A_{i,r}), \quad \sum_{r \in [m]} s_{r,i,j} \leq \sum_{r \in [m]} \mathbb{I}(A_{i,r}) + \mathbb{I}(A_{j,r}).$$

By Lemma A.4, we have

$$\mathbb{E}_{\tilde{w}_r}[s_{r,i,j}] \leq \mathbb{E}_{\tilde{w}_r}[s_{r,i,j}^2] \leq \mathop{\mathbb{P}}_{\tilde{w}_r}[A_{i,r}] + \mathop{\mathbb{P}}_{\tilde{w}_r}[A_{j,r}] \leq 2c(R_w + R_b)\exp(-B^2/2).$$

Define $s_{i,j} = \sum_{r=1}^m \mathbb{I}(A_{i,r}) + \mathbb{I}(A_{j,r})$. By Bernstein's inequality in Lemma D.1,

$$\mathbb{P}\left[s_{i,j} \geq m \cdot 2c(R_w + R_b)\exp(-B^2/2) + mt\right]$$
$$\leq \exp\left(-\frac{m^2 t^2/2}{m \cdot 2c(R_w + R_b)\exp(-B^2/2) + mt/3}\right), \quad \forall t \geq 0.$$

Let $t = 2c(R_w + R_b)\exp(-B^2/2)$. We get

$$\mathbb{P}[s_{i,j} \geq m \cdot 4c(R_w + R_b)\exp(-B^2/2)] \leq \exp\left(-\frac{2}{3}cm(R_w + R_b)\exp(-B^2/2)\right).$$

Thus, we obtain with probability at least $1 - n^2 \exp\left(-\frac{2}{3}cm(R_w + R_b)\exp(-B^2/2)\right)$,

$$\left\|H(\tilde{W}, \tilde{b}) - H(W, b)\right\|_F \leq n \cdot 8c(R_w + R_b)\exp(-B^2/2),$$

$$\left\|Z(\tilde{W}, \tilde{b}) - Z(W, b)\right\|_F \leq \sqrt{n \cdot 8c(R_w + R_b)\exp(-B^2/2)}.$$

For the second result, by Lemma A.1, $\mathbb{P}[\lambda_{\min}(H(\tilde{W}, \tilde{b})) \geq 0.75\lambda] \geq 1 - \delta$. Hence, with probability at least $1 - \delta - n^2 \exp\left(-\frac{2}{3}cm(R_w + R_b)\exp(-B^2/2)\right)$,

$$\lambda_{\min}(H(W, b)) \geq \lambda_{\min}(H(\tilde{W}, \tilde{b})) - \left\|H(W, b) - H(\tilde{W}, \tilde{b})\right\|$$

$$\geq \lambda_{\min}(H(\tilde{W}, \tilde{b})) - \left\|H(W, b) - H(\tilde{W}, \tilde{b})\right\|_F$$

$$\geq 0.75\lambda - n \cdot 8c(R_w + R_b)\exp(-B^2/2).$$

$\square$

### A.4 TOTAL MOVEMENT OF WEIGHTS AND BIASES

**Definition A.7** (NTK at time $t$). *For $t \geq 0$, let $H(t)$ be an $n \times n$ matrix with $(i, j)$-th entry*

$$H_{ij}(t) := \left\langle \frac{\partial f(x_i; \theta(t))}{\partial \theta(t)}, \frac{\partial f(x_j; \theta(t))}{\partial \theta(t)} \right\rangle = \frac{1}{m} \sum_{r=1}^{m} (\langle x_i, x_j \rangle + 1)\mathbb{I}(w_r(t)^\top x_i \geq b_r(t), w_r(t)^\top x_j \geq b_r(t)).$$

We follow the proof strategy from (Du et al., 2018). Now we derive the total movement of weights and biases. Let $f(t) = f(X; \theta(t))$ where $f_i(t) = f(x_i; \theta(t))$. The dynamics of each prediction is given by

$$\frac{d}{dt}f_i(t) = \left\langle \frac{\partial f(x_i; \theta(t))}{\partial \theta(t)}, \frac{d\theta(t)}{dt} \right\rangle = \sum_{j=1}^{n}(y_j - f_j(t))\left\langle \frac{\partial f(x_i; \theta(t))}{\partial \theta(t)}, \frac{\partial f(x_j; \theta(t))}{\partial \theta(t)} \right\rangle = \sum_{j=1}^{n}(y_j - f_j(t))H_{ij}(t),$$

which implies

$$\frac{d}{dt}f(t) = H(t)(y - f(t)). \tag{4}$$

**Lemma A.8** (Gradient Bounds). *For any $0 \leq s \leq t$, we have*

$$\left\|\frac{\partial L(W(s), b(s))}{\partial w_r(s)}\right\|_2 \leq \sqrt{\frac{n}{m}}\|f(s) - y\|_2,$$

$$\left\|\frac{\partial L(W(s), b(s))}{\partial b_r(s)}\right\|_2 \leq \sqrt{\frac{n}{m}}\|f(s) - y\|_2.$$

*Proof.* We have:

$$\left\|\frac{\partial L(W(s), b(s))}{\partial w_r(s)}\right\|_2 = \left\|\frac{1}{\sqrt{m}}\sum_{i=1}^{n}(f(x_i; W(s), b(s)) - y_i)a_r x_i \mathbb{I}(w_r(s)^\top x_i \geq b_r)\right\|_2$$

$$\leq \frac{1}{\sqrt{m}}\sum_{i=1}^{n}|f(x_i; W(s), b(s)) - y_i|$$

$$\leq \sqrt{\frac{n}{m}}\|f(s) - y\|_2,$$

where the first inequality follows from triangle inequality, and the second inequality follows from Cauchy-Schwarz inequality.

Similarly, we also have:

$$\left\|\frac{\partial L(W(s), b(s))}{\partial b_r(s)}\right\|_2 = \left\|\frac{1}{\sqrt{m}}\sum_{i=1}^{n}(f(x_i; W(s), b(s)) - y_i)a_r\mathbb{I}(w_r(s)^\top x_i \geq b_r)\right\|_2$$

$$\leq \frac{1}{\sqrt{m}} \sum_{i=1}^{n} |f(x_i; W(s), b(s)) - y_i|$$

$$\leq \sqrt{\frac{n}{m}} \|f(s) - y\|_2.$$

$\square$

### A.4.1 GRADIENT DESCENT

**Lemma A.9.** *Assume $\lambda > 0$. Assume $\|y - f(k)\|_2^2 \leq (1 - \eta\lambda/4)^k \|y - f(0)\|_2^2$ holds for all $k' \leq k$. Then for every $r \in [m]$,*

$$\|w_r(k+1) - w_r(0)\|_2 \leq \frac{8\sqrt{n}\,\|y - f(0)\|_2}{\sqrt{m}\lambda} := D_w,$$

$$|b_r(k+1) - b_r(0)| \leq \frac{8\sqrt{n}\,\|y - f(0)\|_2}{\sqrt{m}\lambda} := D_b.$$

*Proof.*

$$\|w_r(k+1) - w_r(0)\|_2 \leq \eta \sum_{k'=0}^{k} \left\| \frac{\partial L(W(k'))}{\partial w_r(k')} \right\|_2$$

$$\leq \eta \sum_{k'=0}^{k} \sqrt{\frac{n}{m}} \|y - f(k')\|_2$$

$$\leq \eta \sum_{k'=0}^{k} \sqrt{\frac{n}{m}} (1 - \eta\lambda/4)^{k'/2} \|y - f(0)\|_2$$

$$\leq \eta \sum_{k'=0}^{k} \sqrt{\frac{n}{m}} (1 - \eta\lambda/8)^{k'} \|y - f(0)\|_2$$

$$\leq \eta \sum_{k'=0}^{\infty} \sqrt{\frac{n}{m}} (1 - \eta\lambda/8)^{k'} \|y - f(0)\|_2$$

$$\leq \frac{8\sqrt{n}}{\sqrt{m}\lambda} \|y - f(0)\|_2,$$

where the first inequality is by Triangle inequality, the second inequality is by Lemma A.8, the third inequality is by our assumption and the fourth inequality is by $(1 - x)^{1/2} \leq 1 - x/2$ for $x \geq 0$.

The proof for $b$ is similar. $\square$

### A.4.2 GRADIENT FLOW

**Lemma A.10.** *Suppose for $0 \leq s \leq t$, $\lambda_{\min}(H(s)) \geq \frac{\lambda_0}{2} > 0$. Then we have $\|y - f(t)\|_2^2 \leq \exp(-\lambda_0 t) \|y - f(0)\|_2^2$ and for any $r \in [m]$, $\|w_r(t) - w_r(0)\|_2 \leq \frac{\sqrt{n}\|y - f(0)\|_2}{\sqrt{m}\lambda_0}$ and $|b_r(t) - b_r(0)| \leq \frac{\sqrt{n}\|y - f(0)\|_2}{\sqrt{m}\lambda_0}$.*

*Proof.* By the dynamics of prediction in Equation (4), we have

$$\frac{d}{dt} \|y - f(t)\|_2^2 = -2(y - f(t))^\top H(t)(y - f(t))$$

$$\leq -\lambda_0 \|y - f(t)\|_2^2,$$

which implies

$$\|y - f(t)\|_2^2 \leq \exp(-\lambda_0 t) \|y - f(t)\|_2^2.$$

Now we bound the gradient norm of the weights

$$\left\| \frac{d}{ds} w_r(s) \right\|_2 = \left\| \sum_{i=1}^n (y_i - f_i(s)) \frac{1}{\sqrt{m}} a_r x_i \mathbb{I}(w_r(s)^\top x_i \geq b(s)) \right\|_2$$

$$\leq \frac{1}{\sqrt{m}} \sum_{i=1}^n |y_i f_i(s)| \leq \frac{\sqrt{n}}{\sqrt{m}} \|y - f(s)\|_2 \leq \frac{\sqrt{n}}{\sqrt{m}} \exp(-\lambda_0 s) \|y - f(0)\|_2.$$

Integrating the gradient, the change of weight can be bounded as

$$\|w_r(t) - w_r(0)\|_2 \leq \int_0^t \left\| \frac{d}{ds} w_r(s) \right\|_2 ds \leq \frac{\sqrt{n} \|y - f(0)\|_2}{\sqrt{m} \lambda_0}.$$

For bias, we have

$$\left\| \frac{d}{ds} b_r(s) \right\|_2 = \left\| \sum_{i=1}^n (y_i - f_i(s)) \frac{1}{\sqrt{m}} a_r \mathbb{I}(w_r(s)^\top x_i \geq b(s)) \right\|_2$$

$$\leq \frac{1}{\sqrt{m}} \sum_{i=1}^n |y_i - f_i(s)| \leq \frac{\sqrt{n}}{\sqrt{m}} \|y - f(s)\|_2 \leq \frac{\sqrt{n}}{\sqrt{m}} \exp(-\lambda_0 s) \|y - f(0)\|_2.$$

Now, the change of bias can be bounded as

$$\|b_r(t) - b_r(0)\|_2 \leq \int_0^t \left\| \frac{d}{ds} w_r(s) \right\|_2 ds \leq \frac{\sqrt{n} \|y - f(0)\|_2}{\sqrt{m} \lambda_0}.$$

$\square$

## A.5 GRADIENT DESCENT CONVERGENCE ANALYSIS

### A.5.1 UPPER BOUND OF THE INITIAL ERROR

**Lemma A.11** (Initial error upper bound). *Let $B > 0$ be the initialization value of the biases and all the weights be initialized from standard Gaussian. Let $\delta \in (0, 1)$ be the failure probability. Then, with probability at least $1 - \delta$, we have*

$$\|f(0)\|_2^2 = O(n(\exp(-B^2/2) + 1/m) \log^3(mn/\delta)),$$
$$\|f(0) - y\|_2^2 = O\left(n + n\left(\exp(-B^2/2) + 1/m\right) \log^3(2mn/\delta)\right).$$

*Proof.* Since we are only analyzing the initialization stage, for notation ease, we omit the dependence on time without any confusion. We compute

$$\|y - f\|_2^2 = \sum_{i=1}^n (y_i - f(x_i))^2$$

$$= \sum_{i=1}^n \left( y_i - \frac{1}{\sqrt{m}} \sum_{r=1}^m a_r \sigma(w_r^\top x_i - B) \right)^2$$

$$= \sum_{i=1}^n \left( y_i^2 - 2 \frac{y_i}{\sqrt{m}} \sum_{r=1}^m a_r \sigma(w_r^\top x_i - B) + \frac{1}{m} \left( \sum_{r=1}^m a_r \sigma(w_r^\top x_i - B) \right)^2 \right).$$

Since $w_r^\top x_i \sim \mathcal{N}(0, 1)$ for all $r \in [m]$ and $i \in [n]$, by Gaussian tail bound and a union bound over $r, i$, we have

$$\mathbb{P}[\forall i \in [n], \ j \in [m] : w_r^\top x_i \leq \sqrt{2 \log(2mn/\delta)}] \geq 1 - \delta/2.$$

Let $E_1$ denote this event. Conditioning on the event $E_1$, let

$$z_{i,r} := \frac{1}{\sqrt{m}} \cdot a_r \cdot \min \left\{ \sigma(w_r^\top x_i - B), \sqrt{2 \log(2mn/\delta)} \right\}.$$

Notice that $z_{i,r} \neq 0$ with probability at most $\exp(-B^2/2)$. Thus,

$$\mathbb{E}_{a_r, w_r}[z_{i,r}^2] \leq \exp(-B^2/2)\frac{1}{m}2\log(2mn/\delta).$$

By randomness in $a_r$, we know $\mathbb{E}[z_{i,r}] = 0$. Now apply Bernstein's inequality in Lemma D.1, we have for all $t > 0$,

$$\mathbb{P}\left[\left|\sum_{r=1}^{m} z_{i,r}\right| > t\right] \leq \exp\left(-\min\left(\frac{t^2/2}{4\exp(-B^2/2)\log(2mn/\delta)}, \frac{\sqrt{m}t/2}{2\sqrt{2\log(2mn/\delta)}}\right)\right).$$

Thus, by a union bound, with probability at least $1 - \delta/2$, for all $i \in [n]$,

$$\left|\sum_{r=1}^{m} z_{i,r}\right| \leq \sqrt{2\log(2mn/\delta)\exp(-B^2/2)2\log(2n/\delta)} + 2\sqrt{\frac{2\log(2mn/\delta)}{m}}\log(2n/\delta)$$

$$\leq \left(2\exp(-B^2/4) + 2\sqrt{2/m}\right)\log^{3/2}(2mn/\delta).$$

Let $E_2$ denote this event. Thus, conditioning on the events $E_1, E_2$, with probability $1 - \delta$,

$$\|f(0)\|_2^2 = \sum_{i=1}^{n}\left(\sum_{r=1}^{m} z_{i,r}\right)^2 = O(n(\exp(-B^2/2) + 1/m)\log^3(mn/\delta))$$

and

$$\|y - f(0)\|_2^2$$

$$= \sum_{i=1}^{n} y_i^2 - 2\sum_{i=1}^{n} y_i \sum_{r=1}^{m} z_{i,r} + \sum_{i=1}^{n}\left(\sum_{r=1}^{m} z_{i,r}\right)^2$$

$$\leq \sum_{i=1}^{n} y_i^2 + 2\sum_{i=1}^{n}|y_i|\left(2\exp(-B^2/4) + 2\sqrt{2/m}\right)\log^{3/2}(2mn/\delta)$$

$$+ \sum_{i=1}^{n}\left(\left(2\exp(-B^2/4) + 2\sqrt{2/m}\right)\log^{3/2}(2mn/\delta)\right)^2$$

$$= O\left(n + n\left(\exp(-B^2/2) + 1/m\right)\log^3(2mn/\delta)\right),$$

where we assume $y_i = O(1)$ for all $i \in [n]$. $\qquad\square$

### A.5.2 Error Decomposition

We follow the proof outline in (Song & Yang, 2019; Song et al., 2021a) and we generalize it to networks with trainable $b$. Let us define matrix $H^{\perp}$ similar to $H$ except only considering flipped neurons by

$$H_{ij}^{\perp}(k) := \frac{1}{m}\sum_{r \in \overline{S}_i}(\langle x_i, x_j\rangle + 1)\mathbb{I}(w_r(k)^\top x_i \geq b_r(k), w_r(k)^\top x_j \geq b_r(k))$$

and vector $v_1, v_2$ by

$$v_{1,i} := \frac{1}{\sqrt{m}}\sum_{r \in S_i} a_r(\sigma(\langle w_r(k+1), x_i\rangle - b_r(k+1)) - \sigma(\langle w_r(k), x_i\rangle - b_r(k))),$$

$$v_{2,i} := \frac{1}{\sqrt{m}}\sum_{r \in \overline{S}_i} a_r(\sigma(\langle w_r(k+1), x_i\rangle - b_r(k+1)) - \sigma(\langle w_r(k), x_i\rangle - b_r(k))).$$

Now we give out our error update.

**Claim A.12.**

$$\|y - f(k+1)\|_2^2 = \|y - f(k)\|_2^2 + B_1 + B_2 + B_3 + B_4,$$

*where*

$$B_1 := -2\eta(y - f(k))^\top H(k)(y - f(k)),$$
$$B_2 := 2\eta(y - f(k))^\top H^\perp(k)(y - f(k)),$$
$$B_3 := -2(y - f(k))^\top v_2,$$
$$B_4 := \|f(k+1) - f(k)\|_2^2.$$

*Proof.* First we can write

$$v_{1,i} = \frac{1}{\sqrt{m}} \sum_{r \in S_i} a_r \left( \sigma \left( \left\langle w_r(k) - \eta \frac{\partial L}{\partial w_r}, x_i \right\rangle - \left( b_r(k) - \eta \frac{\partial L}{\partial b_r} \right) \right) - \sigma(\langle w_r(k), x_i \rangle - b_r(k)) \right)$$

$$= \frac{1}{\sqrt{m}} \sum_{r \in S_i} a_r \left( \left\langle -\eta \frac{\partial L}{\partial w_r}, x_i \right\rangle + \eta \frac{\partial L}{\partial b_r} \right) \mathbb{I}(\langle w_r(k), x_i \rangle - b_r(k) \geq 0)$$

$$= \frac{1}{\sqrt{m}} \sum_{r \in S_i} a_r \left( \eta \frac{1}{\sqrt{m}} \sum_{j=1}^n (y_j - f_j(k)) a_r (\langle x_j, x_i \rangle + 1) \mathbb{I}(w_r(k)^\top x_j \geq b_r(k)) \right) \mathbb{I}(\langle w_r(k), x_i \rangle - b_r(k) \geq 0)$$

$$= \eta \sum_{j=1}^n (y_j - f_j(k))(H_{ij}(k) - H_{ij}^\perp(k))$$

which means

$$v_1 = \eta(H(k) - H^\perp(k))(y - f(k)).$$

Now we compute

$$\|y - f(k+1)\|_2^2 = \|y - f(k) - (f(k+1) - f(k))\|_2^2$$
$$= \|y - f(k)\|_2^2 - 2(y - f(k))^\top (f(k+1) - f(k)) + \|f(k+1) - f(k)\|_2^2.$$

Since $f(k+1) - f(k) = v_1 + v_2$, we can write the cross product term as

$$(y - f(k))^\top (f(k+1) - f(k))$$
$$= (y - f(k))^\top (v_1 + v_2)$$
$$= (y - f(k))^\top v_1 + (y - f(k))^\top v_2$$
$$= \eta(y - f(k))^\top H(k)(y - f(k))$$
$$\quad - \eta(y - f(k))^\top H^\perp(k)(y - f(k)) + (y - f(k))^\top v_2.$$

$\square$

### A.5.3 BOUNDING THE DECREASE OF THE ERROR

**Lemma A.13.** *Assume $\lambda > 0$. Assume we choose $R_w, R_b, B$ where $R_w, R_b \leq \min\{1/B, 1\}$ such that $8cn(R_w + R_b) \exp(-B^2/2) \leq \lambda/8$. Denote $\delta_0 = \delta + n^2 \exp(-\frac{2}{3} cm(R_w + R_b) \exp(-B^2/2))$. Then,*

$$\mathbb{P}[B_1 \leq -\eta 5\lambda \|y - f(k)\|_2^2 /8] \geq 1 - \delta_0.$$

*Proof.* By Lemma A.6 and our assumption,

$$\lambda_{\min}(H(W)) > 0.75\lambda - n \cdot 8c(R_w + R_b) \exp(-B^2/2) \geq 5\lambda/8$$

with probability at least $1 - \delta_0$. Thus,

$$(y - f(k))^\top H(k)(y - f(k)) \geq \|y - f(k)\|_2^2 \, 5\lambda/8.$$

$\square$

### A.5.4 BOUNDING THE EFFECT OF FLIPPED NEURONS

Here we bound the term $B_2, B_3$. First, we introduce a fact.

**Fact A.14.**

$$\left\| H^{\perp}(k) \right\|_F^2 \leq \frac{4n}{m^2} \sum_{i=1}^n |\overline{S}_i|^2.$$

*Proof.*

$$\left\| H^{\perp}(k) \right\|_F^2 = \sum_{i,j\in[n]} \left( \frac{1}{m} \sum_{r\in\overline{S}_i} (x_i^\top x_j + 1) \mathbb{I}(w_r(k)^\top x_i \geq b_r(k), \ w_r(k)^\top x_j \geq b_r(k)) \right)^2$$

$$\leq \sum_{i,j\in[n]} \left( \frac{1}{m} 2|\overline{S}_i| \right)^2 \leq \frac{4n}{m^2} \sum_{i=1}^n |\overline{S}_i|^2.$$

$\square$

**Lemma A.15.** *Denote* $\delta_0 = n \exp(-\frac{2}{3} cm(R_w + R_b) \exp(-B^2/2))$. *Then,*

$$\mathbb{P}[B_2 \leq 8\eta nc(R_w + R_b) \exp(-B^2/2) \cdot \|y - f(k)\|_2^2] \geq 1 - \delta_0.$$

*Proof.* First, we have

$$B_2 \leq 2\eta \|y - f(k)\|_2^2 \left\| H^{\perp}(k) \right\|_2.$$

Then, by Fact A.14,

$$\left\| H^{\perp}(k) \right\|_2^2 \leq \left\| H^{\perp}(k) \right\|_F^2 \leq \frac{4n}{m^2} \sum_{i=1}^n |\overline{S}_i|^2.$$

By Corollary A.5, we have

$$\mathbb{P}[\forall i \in [n] : \ |\overline{S}_i| \leq 2mc(R_w + R_b) \exp(-B^2/2)] \geq 1 - \delta_0.$$

Thus, with probability at least $1 - \delta_0$,

$$\left\| H^{\perp}(k) \right\|_2 \leq 4nc(R_w + R_b) \exp(-B^2/2).$$

$\square$

**Lemma A.16.** *Denote* $\delta_0 = n \exp(-\frac{2}{3} cm(R_w + R_b) \exp(-B^2/2))$. *Then,*

$$\mathbb{P}[B_3 \leq 4c\eta n(R_w + R_b) \exp(-B^2/2) \|y - f(k)\|_2^2] \geq 1 - \delta_0.$$

*Proof.* By Cauchy-Schwarz inequality, we have $B_3 \leq 2 \|y - f(k)\|_2 \|v_2\|_2$. We have

$$\|v_2\|_2^2 \leq \sum_{i=1}^n \left( \frac{\eta}{\sqrt{m}} \sum_{r\in\overline{S}_i} \left| \left\langle \frac{\partial L}{\partial w_r}, x_i \right\rangle \right| + \left| \frac{\partial L}{\partial b_r} \right| \right)^2$$

$$\leq \sum_{i=1}^n \frac{\eta^2}{m} \max_{i\in[n]} \left( \left| \left\langle \frac{\partial L}{\partial w_r}, x_i \right\rangle \right| + \left| \frac{\partial L}{\partial b_r} \right| \right)^2 |\overline{S}_i|^2$$

$$\leq n\frac{\eta^2}{m} \left( 2\sqrt{\frac{n}{m}} \|f(k) - y\|_2 \, 2mc(R_w + R_b) \exp(-B^2/2) \right)^2$$

$$= 16c^2\eta^2 n^2 \|y - f(k)\|_2^2 (R_w + R_b)^2 \exp(-B^2),$$

where the last inequality is by Lemma A.8 and Corollary A.5 which holds with probability at least $1 - \delta_0$. $\square$

### A.5.5 BOUNDING THE NETWORK UPDATE

**Lemma A.17.**

$$B_4 \leq 4\eta^2 n^2 \|y - f(k)\|_2^2.$$

*new result:*

$$B_4 \leq C_2^2 \eta^2 n^2 \|y - f(k)\|_2^2 \exp(-B^2).$$

*for some constant $C_2$.*

*Proof.*

$$\|f(k+1) - f(k)\|_2^2 \leq \sum_{i=1}^{n} \left( \frac{\eta}{\sqrt{m}} \sum_{r=1}^{m} \left| \left\langle \frac{\partial L}{\partial w_r}, x_i \right\rangle \right| + \left| \frac{\partial L}{\partial b_r} \right| \right)^2$$
$$\leq 4\eta^2 n^2 \|y - f(k)\|_2^2.$$

$\square$

*New Proof.* Recall that the definition that $\mathcal{S}_{\mathrm{on}}(i,t) = \{r \in [m] : w_r(t)^\top x_i \geq b_r(t)\}$, i.e., the set of neurons that activates for input $x_i$ at the $t$-th step of gradient descent.

$$\|f(k+1) - f(k)\|_2^2 \leq \sum_{i=1}^{n} \left( \frac{\eta}{\sqrt{m}} \sum_{r:r \in \mathcal{S}_{\mathrm{on}}(i,k+1) \cup \mathcal{S}_{\mathrm{on}}(i,k)} \left| \left\langle \frac{\partial L}{\partial w_r}, x_i \right\rangle \right| + \left| \frac{\partial L}{\partial b_r} \right| \right)^2$$
$$\leq n \frac{\eta^2}{m} (|\mathcal{S}_{\mathrm{on}}(i,k+1)| + |\mathcal{S}_{\mathrm{on}}(i,k)|)^2 \max_{i \in [n]} \left( \left| \left\langle \frac{\partial L}{\partial w_r}, x_i \right\rangle \right| + \left| \frac{\partial L}{\partial b_r} \right| \right)^2$$
$$\leq n \frac{\eta^2}{m} \left( C_2 m \exp(-B^2/2) \cdot \sqrt{\frac{n}{m}} \|y - f(k)\|_2 \right)^2$$
$$\leq C_2^2 \eta^2 n^2 \|y - f(k)\|_2^2 \exp(-B^2).$$

where the third inequality is by Lemma A.19 for some $C_2$. $\square$

### A.5.6 PUTTING IT ALL TOGETHER

**Theorem A.18** (Convergence). *Assume $\lambda > 0$. Let $\eta \leq \lambda/(64n^2)$ $\eta \leq \frac{\lambda \exp(B^2)}{5C_2^2 n^2}$, $B \in [0, \sqrt{0.5 \log m}]$ and*

$$m \geq \widetilde{\Omega} \left( \lambda^{-4} n^4 \left( 1 + \left( \exp(-B^2/2) + 1/m \right) \log^3(2mn/\delta) \right) \exp(-B^2) \right).$$

*Assume $\lambda = \lambda_0 \exp(-B^2/2)$ for some constant $\lambda_0$. Then,*

$$\mathbb{P}\left[ \forall t : \|y - f(t)\|_2^2 \leq (1 - \eta\lambda/4)^t \|y - f(0)\|_2^2 \right] \geq 1 - \delta - e^{-\Omega(n)}.$$

*Proof.* From Lemma A.13, Lemma A.15, Lemma A.16 and Lemma A.17, we know with probability at least $1 - 2n^2 \exp(-\frac{2}{3}cm(R_w + R_b) \exp(-B^2/2)) - \delta$, we have

$$\|y - f(k+1)\|_2^2 \leq \|y - f(k)\|_2^2 (1 - 5\eta\lambda/8 + 12\eta nc(R_w + R_b) \exp(-B^2/2) + 4\eta^2 n^2).$$

$$\|y - f(k+1)\|_2^2 \leq \|y - f(k)\|_2^2 (1 - 5\eta\lambda/8 + 12\eta nc(R_w + R_b) \exp(-B^2/2) + C_2^2 \eta^2 n^2 \|y - f(k)\|_2^2 \exp(-B^2)).$$

By Lemma A.9, we need

$$D_w = \frac{8\sqrt{n} \|y - f(0)\|_2}{\sqrt{m}\lambda} \leq R_w,$$
$$D_b = \frac{8\sqrt{n} \|y - f(0)\|_2}{\sqrt{m}\lambda} \leq R_b.$$

By Lemma A.11, we have

$$\mathbb{P}[\|f(0) - y\|_2^2 = O\left(n + n\left(\exp(-B^2/2) + 1/m\right)\log^3(2mn/\delta)\right)] \geq 1 - \delta.$$

Let $R = \min\{R_w, R_b\}$, $D = \max\{D_w, D_b\}$. Combine the results we have

$$R > \Omega(\lambda^{-1}m^{-1/2}n\sqrt{1 + (\exp(-B^2/2) + 1/m)\log^3(2mn/\delta)}).$$

Lemma A.13 requires

$$8cn(R_w + R_b)\exp(-B^2/2) \leq \lambda/8$$
$$\Rightarrow R \leq \frac{\lambda \exp(B^2/2)}{128cn}.$$

which implies a lower bound on $m$

$$m \geq \Omega\left(\lambda^{-4}n^4\left(1 + \left(\exp(-B^2/2) + 1/m\right)\log^3(2mn/\delta)\right)\exp(-B^2)\right).$$

Lemma A.1 further requires a lower bound of $m = \Omega(\lambda^{-1}n \cdot \log(n/\delta))$ which can be ignored.

Lemma A.6 further requires $R < \min\{1/B, 1\}$ which implies

$$B < \frac{128cn}{\lambda \exp(B^2/2)},$$
$$m \geq \widetilde{\Omega}\left(\lambda^{-4}n^4\left(1 + \left(\exp(-B^2/2) + 1/m\right)\log^3(2mn/\delta)\right)\exp(-B^2)\right).$$

From Theorem F.1 in (Song et al., 2021a) we know that $\lambda = \lambda_0 \exp(-B^2/2)$ for some $\lambda_0$ with no dependence on $B$ and $\lambda \exp(B^2/2) \leq 1$. Thus, by our constraint on $m$ and $B$, this is always satisfied.

Finally, to require

$$12\eta nc(R_w + R_b)\exp(-B^2/2) + 4\eta^2 n^2 \leq \eta\lambda/4,$$

$$\textcolor{blue}{12\eta nc(R_w + R_b)\exp(-B^2/2) + C_2^2 \eta^2 n^2 \exp(-B^2) \leq \eta\lambda/4,}$$

we need $\eta \leq \lambda/(64n^2)$ $\textcolor{blue}{\eta \leq \frac{\lambda \exp(B^2)}{5C_2^2 n^2}}$. By our choice of $m, B$, we have

$$2n^2 \exp(-\frac{2}{3}cm(R_w + R_b)\exp(-B^2/2)) = e^{-\Omega(n)}.$$

$\square$

## A.6   BOUNDING THE NUMBER OF ACTIVATED NEURONS PER ITERATION

First we define the set of activated neurons at iteration $t$ for training point $x_i$ to be

$$\mathcal{S}_{\text{on}}(i, t) = \{r \in [m]: \ w_r(t)^\top x_i \geq b_r(t)\}.$$

**Lemma A.19** (Number of Activated Neurons at Initialization). *Assume the choice of $m$ in Theorem A.18. With probability at least $1 - e^{-\Omega(n)}$ over the random initialization, we have*

$$|\mathcal{S}_{\text{on}}(i, t)| = O(m \cdot \exp(-B^2/2)),$$

*for all $0 \leq t \leq T$ and $i \in [n]$. And As a by-product,*

$$\|Z(0)\|_F^2 \leq 8n \exp(-B^2/2).$$

*Proof.* First we bound the number of activated neuron at the initialization. We have $\mathbb{P}[w_r^\top x_i \geq B] \leq \exp(-B^2/2)$. By Bernstein's inequality,

$$\mathbb{P}[|\mathcal{S}_{\text{on}}(i, 0)| \geq m \exp(-B^2/2) + t] \leq \exp\left(-\frac{t^2}{m \exp(-B^2/2) + t/3}\right).$$

Take $t = m \exp(-B^2/2)$ we have

$$\mathbb{P}[|\mathcal{S}_{\text{on}}(i,0)| \geq 2m\exp(-B^2/2)] \leq \exp\left(-m\exp(-B^2/2)/4\right).$$

By a union bound over $i \in [n]$, we have

$$\mathbb{P}[\forall i \in [n] : |\mathcal{S}_{\text{on}}(i,0)| \leq 2m\exp(-B^2/2)] \geq 1 - n\exp\left(-m\exp(-B^2/2)/4\right).$$

Notice that

$$\|Z(0)\|_F^2 \leq \frac{4}{m}\sum_{r=1}^{m}\sum_{i=1}^{n}\mathbb{I}_{r,i}(0) \leq 8n\exp(-B^2/2).$$

$\square$

**Lemma A.20** (Number of Activated Neurons per Iteration). *Assume the parameter settings in Theorem A.18. With probability at least $1 - e^{-\Omega(n)}$ over the random initialization, we have*

$$|\mathcal{S}_{\text{on}}(i,t)| = O(m \cdot \exp(-B^2/2))$$

*for all $0 \leq t \leq T$ and $i \in [n]$.*

*Proof.* By Corollary A.5 and Theorem A.18, we have

$$\mathbb{P}[\forall i \in [n] : |\overline{S}_i| \leq 4mc\exp(-B^2/2)] \geq 1 - e^{-\Omega(n)}.$$

Recall $\overline{S}_i$ is the set of flipped neurons during the entire training process. Notice that $|\mathcal{S}_{\text{on}}(i,t)| \leq |\mathcal{S}_{\text{on}}(i,0)| + |\overline{S}_i|$. Thus, by Lemma A.19

$$\mathbb{P}[\forall i \in [n] : |\mathcal{S}_{\text{on}}(i,t)| = O(m\exp(-B^2/2))] \geq 1 - e^{-\Omega(n)}.$$

$\square$

# B   BOUNDING THE RESTRICTED SMALLEST EIGENVALUE WITH DATA SEPARATION

**Theorem B.1.** *Let $X = (x_1, \ldots, x_n)$ be points in $\mathbb{R}^d$ with $\|x_i\|_2 = 1$ for all $i \in [n]$ and $w \sim \mathcal{N}(0, I_d)$. Suppose that there exists $\delta \in [0, \sqrt{2}]$ such that*

$$\min_{i \neq j \in [n]}(\|x_i - x_j\|_2, \|x_i + x_j\|_2) \geq \delta.$$

*Let $B \geq 0$. Recall the limit NTK matrix $H^\infty$ defined as*

$$H_{ij}^\infty := \mathbb{E}_{w \sim \mathcal{N}(0,I)}\left[(\langle x_i, x_j \rangle + 1)\mathbb{I}(w^\top x_i \geq B, w^\top x_j \geq B)\right].$$

*Define $p_0 = \mathbb{P}[w^\top x_1 \geq B]$ and $p_{ij} = \mathbb{P}[w^\top x_i \geq B, w^\top x_j \geq B]$ for $i \neq j$. Define the (data-dependent) region $\mathcal{R} = \{a \in \mathbb{R}^n : \sum_{i \neq j} a_i a_j p_{ij} \geq \min_{i' \neq j'} p_{i'j'} \sum_{i \neq j} a_i a_j\}$ and let $\lambda := \min_{\|a\|_2 = 1, a \in \mathcal{R}} a^\top H^\infty a$. Then, $\lambda \geq \max(0, \lambda')$ where*

$$\lambda' \geq p_0 - \min_{i \neq j} p_{ij}$$

$$\geq \max\left(\frac{1}{2} - \frac{B}{\sqrt{2\pi}}, \left(\frac{1}{B} - \frac{1}{B^3}\right)\frac{e^{-B^2/2}}{\sqrt{2\pi}}\right) - e^{-B^2/(2-\delta^2/2)}\frac{\pi - \arctan\left(\frac{\delta\sqrt{1-\delta^2/4}}{1-\delta^2/2}\right)}{2\pi}.$$

*Proof.* Define $\Delta := \max_{i \neq j}|\langle x_i, x_j \rangle|$. Then by our assumption,

$$1 - \Delta = 1 - \max_{i \neq j}|\langle x_i, x_j \rangle| = \frac{\min_{i \neq j}(\|x_i - x_j\|_2^2, \|x_i + x_j\|_2^2)}{2} \geq \delta^2/2$$

$$\Rightarrow \Delta \leq 1 - \delta^2/2.$$

Further, we define

$$Z(w) := [x_1 \mathbb{I}(w^\top x_1 \geq B), x_2 \mathbb{I}(w^\top x_2 \geq B), \ldots, x_n \mathbb{I}(w^\top x_n \geq B)] \in \mathbb{R}^{d \times n}.$$

Notice that $H^\infty = \mathbb{E}_{w \sim \mathcal{N}(0,I)} \left[ Z(w)^\top Z(w) + \mathbb{I}(Xw \geq B) \mathbb{I}(Xw \geq B)^\top \right]$. We need to lower bound

$$\begin{aligned}
\min_{\|a\|_2 = 1, a \in \mathcal{R}} a^\top H^\infty a &= \min_{\|a\|_2 = 1, a \in \mathcal{R}} a^\top \mathbb{E}_{w \sim \mathcal{N}(0,I)} \left[ Z(w)^\top Z(w) \right] a \\
&\quad + a^\top \mathbb{E}_{w \sim \mathcal{N}(0,I)} \left[ \mathbb{I}(Xw \geq B) \mathbb{I}(Xw \geq B)^\top \right] a \\
&\geq \min_{\|a\|_2 = 1, a \in \mathcal{R}} a^\top \mathbb{E}_{w \sim \mathcal{N}(0,I)} \left[ \mathbb{I}(Xw \geq B) \mathbb{I}(Xw \geq B)^\top \right] a.
\end{aligned}$$

Now, for a fixed $a$,

$$\begin{aligned}
a^\top \mathbb{E}_{w \sim \mathcal{N}(0,I)} \left[ \mathbb{I}(Xw \geq B) \mathbb{I}(Xw \geq B)^\top \right] a &= \sum_{i=1}^n a_i^2 \, \mathbb{P}[w^\top x_i \geq B] + \sum_{i \neq j} a_i a_j \, \mathbb{P}[w^\top x_i \geq B, \, w^\top x_j \geq B] \\
&= p_0 \|a\|_2^2 + \sum_{i \neq j} a_i a_j p_{ij},
\end{aligned}$$

where the last equality is by $\mathbb{P}[w^\top x_1 \geq B] = \ldots = \mathbb{P}[w^\top x_n \geq B] = p_0$ which is due to spherical symmetry of standard Gaussian. Notice that $\max_{i \neq j} p_{ij} \leq p_0$. Since $a \in \mathcal{R}$,

$$\begin{aligned}
\mathbb{E}_{w \sim \mathcal{N}(0,I)} \left[ (a^\top \mathbb{I}(Xw \geq B))^2 \right] &\geq (p_0 - \min_{i \neq j} p_{ij}) \|a\|_2^2 + (\min_{i \neq j} p_{ij}) \|a\|_2^2 + (\min_{i \neq j} p_{ij}) \sum_{i \neq j} a_i a_j \\
&= (p_0 - \min_{i \neq j} p_{ij}) \|a\|_2^2 + (\min_{i \neq j} p_{ij}) \left( \sum_i a_i \right)^2.
\end{aligned}$$

Thus,

$$\begin{aligned}
\lambda &\geq \min_{\|a\|_2 = 1, a \in \mathcal{R}} \mathbb{E}_{w \sim \mathcal{N}(0,I)} \left[ (a^\top \mathbb{I}(Xw \geq B))^2 \right] \\
&\geq \min_{\|a\|_2 = 1, a \in \mathcal{R}} (p_0 - \min_{i \neq j} p_{ij}) \|a\|_2^2 + \min_{\|a\|_2 = 1, a \in \mathcal{R}} (\min_{i \neq j} p_{ij}) \left( \sum_i a_i \right)^2 \\
&\geq p_0 - \min_{i \neq j} p_{ij}.
\end{aligned}$$

Now we need to upper bound

$$\min_{i \neq j} p_{ij} \leq \max_{i \neq j} p_{ij}.$$

We divide into two cases: $B = 0$ and $B > 0$. Consider two fixed examples $x_1, x_2$. Then, let $v = (I - x_1 x_1^\top) x_2 / \left\| (I - x_1 x_1^\top) x_2 \right\|$ and $c = |\langle x_1, x_2 \rangle|$ [1].

**Case 1: $B = 0$.** First, let us define the region $\mathcal{A}_0$ as

$$\mathcal{A}_0 = \left\{ (g_1, g_2) \in \mathbb{R}^2 : \, g_1 \geq 0, \, g_1 \geq -\frac{\sqrt{1-c^2}}{c} g_2 \right\}.$$

Then,

$$\begin{aligned}
\mathbb{P}[w^\top x_1 \geq 0, \, w^\top x_2 \geq 0] &= \mathbb{P}[w^\top x_1 \geq 0, \, w^\top (cx_1 + \sqrt{1-c^2} v) \geq 0] \\
&= \mathbb{P}[g_1 \geq 0, \, cg_1 + \sqrt{1-c^2} g_2 \geq 0] \\
&= \mathbb{P}[\mathcal{A}_0] \\
&= \frac{\pi - \arctan\left( \frac{\sqrt{1-c^2}}{|c|} \right)}{2\pi}
\end{aligned}$$

---

[1] Here we force $c$ to be positive. Since we are dealing with standard Gaussian, the probability is exactly the same if $c < 0$ by symmetry and therefore, we force $c > 0$.

$$\leq \frac{\pi - \arctan\left(\frac{\sqrt{1-\Delta^2}}{|\Delta|}\right)}{2\pi},$$

where we define $g_1 := w^\top x_1$ and $g_2 := w^\top v$ and the second equality is by the fact that since $x_1$ and $v$ are orthonormal, $g_1$ and $g_2$ are two independent standard Gaussian random variables; the last inequality is by $\arctan$ is a monotonically increasing function and $\frac{\sqrt{1-c^2}}{|c|}$ is a decreasing function in $|c|$ and $|c| \leq \Delta$. Thus,

$$\min_{i \neq j} p_{ij} \leq \max_{i \neq j} p_{ij} \leq \frac{\pi - \arctan\left(\frac{\sqrt{1-\Delta^2}}{|\Delta|}\right)}{2\pi}.$$

**Case 2:** $B > 0$. First, let us define the region

$$\mathcal{A} = \left\{ (g_1, g_2) \in \mathbb{R}^2 : g_1 \geq B, \ g_1 \geq \frac{B}{c} - \frac{\sqrt{1-c^2}}{c} g_2 \right\}.$$

Then, following the same steps as in case 1, we have

$$\mathbb{P}[w^\top x_1 \geq B, \ w^\top x_2 \geq B] = \mathbb{P}[g_1 \geq B, \ cg_1 + \sqrt{1-c^2}g_2 \geq B] = \mathbb{P}[\mathcal{A}].$$

Let $B_1 = B$ and $B_2 = B\sqrt{\frac{1-c}{1+c}}$. Further, notice that $\mathcal{A} = \mathcal{A}_0 + (B_1, B_2)$. Then,

$$
\begin{aligned}
\mathbb{P}[\mathcal{A}] &= \iint_{(g_1, g_2) \in \mathcal{A}} \frac{1}{2\pi} \exp\left\{ -\frac{g_1^2 + g_2^2}{2} \right\} dg_1 \, dg_2 \\
&= \iint_{(g_1, g_2) \in \mathcal{A}_0} \frac{1}{2\pi} \exp\left\{ -\frac{(g_1 + B_1)^2 + (g_2 + B_2)^2}{2} \right\} dg_1 \, dg_2 \\
&= e^{-(B_1^2 + B_2^2)/2} \iint_{(g_1, g_2) \in \mathcal{A}_0} \frac{1}{2\pi} \exp\left\{ -B_1 g_1 - B_2 g_2 \right\} \exp\left\{ -\frac{g_1^2 + g_2^2}{2} \right\} dg_1 \, dg_2.
\end{aligned}
$$

Now, $B_1 g_1 + B_2 g_2 = B g_1 + B\sqrt{\frac{1-c}{1+c}} g_2 \geq 0$ always holds if and only if $g_1 \geq -\sqrt{\frac{1-c}{1+c}} g_2$. Define the region $\mathcal{A}_+$ to be

$$\mathcal{A}_+ = \left\{ (g_1, g_2) \in \mathbb{R}^2 : g_1 \geq 0, \ g_1 \geq -\sqrt{\frac{1-c}{1+c}} g_2 \right\}.$$

Observe that

$$\sqrt{\frac{1-c}{1+c}} \leq \frac{\sqrt{1-c^2}}{c} = \frac{\sqrt{(1-c)(1+c)}}{c} \Leftrightarrow c \leq 1 + c.$$

Thus, $\mathcal{A}_0 \subset \mathcal{A}_+$. Therefore,

$$
\begin{aligned}
\mathbb{P}[\mathcal{A}] &\leq e^{-(B_1^2 + B_2^2)/2} \iint_{(g_1, g_2) \in \mathcal{A}_0} \frac{1}{2\pi} \exp\left\{ -\frac{g_1^2 + g_2^2}{2} \right\} dg_1 \, dg_2 \\
&= e^{-(B_1^2 + B_2^2)/2} \, \mathbb{P}[\mathcal{A}_0] \\
&= e^{-(B_1^2 + B_2^2)/2} \frac{\pi - \arctan\left(\frac{\sqrt{1-c^2}}{|c|}\right)}{2\pi} \\
&\leq e^{-B^2/(1+\Delta)} \frac{\pi - \arctan\left(\frac{\sqrt{1-\Delta^2}}{|\Delta|}\right)}{2\pi}.
\end{aligned}
$$

Finally, we need to lower bound $p_0$. This can be done in two ways: when $B$ is small, we apply Gaussian anti-concentration bound and when $B$ is large, we apply Gaussian tail bounds. Thus,

$$p_0 = \mathbb{P}[w^\top x_1 \geq B] \geq \max\left( \frac{1}{2} - \frac{B}{\sqrt{2\pi}}, \ \left(\frac{1}{B} - \frac{1}{B^3}\right) \frac{e^{-B^2/2}}{\sqrt{2\pi}} \right).$$

Combining the lower bound of $p_0$ and upper bound on $\max_{i \neq j} p_{ij}$ we have

$$\lambda \geq p_0 - \min_{i \neq j} p_{ij} \geq \max\left( \frac{1}{2} - \frac{B}{\sqrt{2\pi}}, \ \left(\frac{1}{B} - \frac{1}{B^3}\right) \frac{e^{-B^2/2}}{\sqrt{2\pi}} \right) - e^{-B^2/(1+\Delta)} \frac{\pi - \arctan\left(\frac{\sqrt{1-\Delta^2}}{|\Delta|}\right)}{2\pi}.$$

Applying $\Delta \leq 1 - \delta^2/2$ and noticing that $H^\infty$ is positive semi-definite gives our final result. $\qquad \square$

## C  GENERALIZATION

### C.1  RADEMACHER COMPLEXITY

In this section, we would like to compute the Rademacher Complexity of our network. Rademacher complexity is often used to bound the deviation from empirical risk and true risk (see, e.g. (Shalev-Shwartz & Ben-David, 2014).)

**Definition C.1** (Empirical Rademacher Complexity). *Given $n$ samples $S$, the* **empirical Rademacher complexity** *of a function class $\mathcal{F}$, where $f : \mathbb{R}^d \to \mathbb{R}$ for $f \in \mathcal{F}$, is defined as*

$$\mathcal{R}_S(\mathcal{F}) = \frac{1}{n}\mathbb{E}_\epsilon \left[ \sup_{f \in \mathcal{F}} \sum_{i=1}^n \epsilon_i f(x_i) \right]$$

*where $\epsilon = (\epsilon_1, \ldots, \epsilon_n)^\top$ and $\epsilon_i$ is an i.i.d Rademacher random variable.*

**Theorem C.2** ((Shalev-Shwartz & Ben-David, 2014)). *Suppose the loss function $\ell(\cdot, \cdot)$ is bounded in $[0, c]$ and is $\rho$-Lipschitz in the first argument. Then with probability at least $1 - \delta$ over sample $S$ of size $n$:*

$$\sup_{f \in \mathcal{F}} L_\mathcal{D}(f) - L_S(f) \leq 2\rho\mathcal{R}_S(\mathcal{F}) + 3c\sqrt{\frac{\log(2/\delta)}{2n}}.$$

In order to get meaningful generalization bound via Rademacher complexity, previous results, such as (Arora et al., 2019; Song & Yang, 2019), multiply the neural network by a scaling factor $\kappa$ to make sure the neural network output something small at the initialization, which requires at least modifying all the previous lemmas we already established. We avoid repeating our arguments by utilizing symmetric initialization to force the neural network to output exactly zero for any inputs at the initialization. [2]

**Definition C.3** (Symmetric Initialization). *For a one-hidden layer neural network with $2m$ neurons, the network is initialized as the following*

1. *For $r \in [m]$, initialize $w_r \sim \mathcal{N}(0, I)$ and $a_r \sim \text{Uniform}(\{-1, 1\})$.*

2. *For $r \in \{m + 1, \ldots, 2m\}$, let $w_r = w_{r-m}$ and $a_r = -a_{r-m}$.*

It is not hard to see that all of our previously established lemmas hold including expectation and concentration. The only effect this symmetric initialization brings is to worse the concentration by a constant factor of 2 which can be easily addressed. For detailed analysis, see (Munteanu et al., 2022).

In order to state our final theorem, we need to use Definition 3.7. Now we can state our theorem for generalization.

**Theorem C.4.** *Fix a failure probability $\delta \in (0, 1)$ and an accuracy parameter $\epsilon \in (0, 1)$. Suppose the training data $S = \{(x_i, y_i)\}_{i=1}^n$ are i.i.d. samples from a $(\lambda, \delta, n)$-non-degenerate distribution $\mathcal{D}$. Assume the settings in Theorem A.18 except now we let*

$$m \geq \widetilde{\Omega}\left(\lambda^{-4}n^6\left(1 + \left(\exp(-B^2/2) + 1/m\right)\log^3(2mn/\delta)\right)\exp(-B^2)\right).$$

*Consider any loss function $\ell : \mathbb{R} \times \mathbb{R} \to [0, 1]$ that is 1-Lipschitz in its first argument. Then with probability at least $1 - 2\delta - e^{-\Omega(n)}$ over the symmetric initialization of $W(0) \in \mathbb{R}^{m \times d}$ and $a \in \mathbb{R}^m$ and the training samples, the two layer neural network $f(W(k), b(k), a)$ trained by gradient descent for $k \geq \Omega(\frac{1}{\eta\lambda}\log\frac{n\log(1/\delta)}{\epsilon})$ iterations has population loss $L_\mathcal{D}(f) = \mathbb{E}_{(x,y)\sim\mathcal{D}}[\ell(f(x), y)]$ upper bounded as*

$$L_\mathcal{D}(f(W(k), b(k), a)) \leq \sqrt{\frac{y^\top(H^\infty)^{-1}y \cdot 32\exp(-B^2/2)}{n}} + \tilde{O}\left(\frac{1}{n^{1/2}}\right).$$

---

[2]While preparing the manuscript, the authors notice that this can be alternatively solved by reparameterized the neural network by $f(x; W) - f(x; W_0)$ and thus minimizing the following objective $L = \frac{1}{2}\sum_{i=1}^n(f(x_i; W) - f(x_i; W_0) - y_i)^2$. The corresponding generalization is the same since Rademacher complexity is invariant to translation. However, since the symmetric initialization is widely adopted in theory literature, we go with symmetric initialization here.

*Proof.* First, we need to bound $L_S$. After training, we have $\|f(k) - y\|_2 \le \epsilon < 1$, and thus

$$
\begin{aligned}
L_S(f(W(k), b(k), a)) &= \frac{1}{n} \sum_{i=1}^{n} [\ell(f_i(k), y_i) - \ell(y_i, y_i)] \\
&\le \frac{1}{n} \sum_{i=1}^{n} |f_i(k) - y_i| \\
&\le \frac{1}{\sqrt{n}} \|f(k) - y\|_2 \\
&\le \frac{1}{\sqrt{n}}.
\end{aligned}
$$

By Theorem C.2, we know that

$$
\begin{aligned}
L_{\mathcal{D}}(f(W(k), b(k), a)) &\le L_S(f(W(k), b(k), a)) + 2\mathcal{R}_S(\mathcal{F}) + \tilde{O}(n^{-1/2}) \\
&\le 2\mathcal{R}_S(\mathcal{F}) + \tilde{O}(n^{-1/2}).
\end{aligned}
$$

Then, by Theorem C.5, we get that for sufficiently large $m$,

$$
\begin{aligned}
\mathcal{R}_S(\mathcal{F}) &\le \sqrt{\frac{y^\top (H^\infty)^{-1} y \cdot 8 \exp(-B^2/2)}{n}} + \tilde{O}\left(\frac{\exp(-B^2/4)}{n^{1/2}}\right) \\
&\le \sqrt{\frac{y^\top (H^\infty)^{-1} y \cdot 8 \exp(-B^2/2)}{n}} + \tilde{O}\left(\frac{1}{n^{1/2}}\right),
\end{aligned}
$$

where the last step follows from $B > 0$.

Therefore, we conclude that:

$$
L_{\mathcal{D}}(f(W(k), b(k), a)) \le \sqrt{\frac{y^\top (H^\infty)^{-1} y \cdot 32 \exp(-B^2/2)}{n}} + \tilde{O}\left(\frac{1}{n^{1/2}}\right).
$$

$\square$

**Theorem C.5.** *Fix a failure probability $\delta \in (0, 1)$. Suppose the training data $S = \{(x_i, y_i)\}_{i=1}^{n}$ are i.i.d. samples from a $(\lambda, \delta, n)$-non-degenerate distribution $\mathcal{D}$. Assume the settings in Theorem A.18 except now we let*

$$
m \ge \tilde{\Omega}\left(\lambda^{-6} n^6 \left(1 + \left(\exp(-B^2/2) + 1/m\right) \log^3(2mn/\delta)\right) \exp(-B^2)\right).
$$

*Denote the set of one-hidden-layer neural networks trained by gradient descent as $\mathcal{F}$. Then with probability at least $1 - 2\delta - e^{-\Omega(n)}$ over the randomness in the symmetric initialization and the training data, the set $\mathcal{F}$ has empirical Rademacher complexity bounded as*

$$
\mathcal{R}_S(\mathcal{F}) \le \sqrt{\frac{y^\top (H^\infty)^{-1} y \cdot 8 \exp(-B^2/2)}{n}} + \tilde{O}\left(\frac{\exp(-B^2/4)}{n^{1/2}}\right).
$$

Note that the only extra requirement we make on $m$ is the $(n/\lambda)^6$ dependence instead of $(n/\lambda)^4$ which is needed for convergence. The dependence of $m$ on $n$ is significantly better than previous work (Song & Yang, 2019) where the dependence is $n^{14}$. We take advantage of our initialization and new analysis to improve the dependence on $n$.

*Proof.* Let $R_w$ ($R_b$) denotes the maximum distance moved any any neuron weight (bias), the same role as $D_w$ ($D_b$) in Lemma A.9. From Lemma A.9 and Lemma A.11, and we have

$$
\max(R_w, R_b) \le O\left(\frac{n\sqrt{1 + (\exp(-B^2/2) + 1/m) \log^3(2mn/\delta)}}{\sqrt{m}\lambda}\right).
$$

The rest of the proof depends on the results from Lemma C.6 and Lemma C.8. Let $R :=$ $\|[W, b](k) - [W, b](0)\|_F$. By Lemma C.6 we have

$$\mathcal{R}_S(\mathcal{F}_{R_w, R_b, R}) \le R\sqrt{\frac{8\exp(-B^2/2)}{n}} + 4c(R_w + R_b)^2\sqrt{m}\exp(-B^2/2)$$

$$\le R\sqrt{\frac{8\exp(-B^2/2)}{n}} + O\left(\frac{n^2(1 + (\exp(-B^2/2) + 1/m)\log^3(2mn/\delta))\exp(-B^2/2)}{\sqrt{m}\lambda^2}\right).$$

Lemma C.8 gives that

$$R \le \sqrt{y^\top(H^\infty)^{-1}y} + O\left(\frac{n}{\lambda}\left(\frac{\exp(-B^2/2)\log(n/\delta)}{m}\right)^{1/4}\right) + O\left(\frac{n\sqrt{(R_w + R_b)\exp(-B^2/2)}}{\lambda}\right)$$

$$+ \frac{n}{\lambda^2} \cdot O\left(\exp(-B^2/4)\sqrt{\frac{\log(n^2/\delta)}{m}} + (R_w + R_b)\exp(-B^2/2)\right).$$

Combining the above results and using the choice of $m, R, B$ in Theorem A.18 gives us

$$\mathcal{R}(\mathcal{F}) \le \sqrt{\frac{y^\top(H^\infty)^{-1}y \cdot 8\exp(-B^2/2)}{n}} + O\left(\frac{\sqrt{n}\exp(-B^2/2)}{\lambda}\left(\frac{\exp(-B^2/2)\log(n/\delta)}{m}\right)^{1/4}\right)$$

$$+ O\left(\frac{\sqrt{n}(R_w + R_b)}{\lambda\exp(B^2/2)}\right) + \frac{\sqrt{n}}{\lambda^2} \cdot O\left(\exp(-B^2/2)\sqrt{\frac{\log(n^2/\delta)}{m}} + (R_w + R_b)\exp(-3B^2/4)\right)$$

$$+ O\left(\frac{n^2(1 + (\exp(-B^2/2) + 1/m)\log^3(2mn/\delta))\exp(-B^2/2)}{\sqrt{m}\lambda^2}\right).$$

Now, we analyze the terms one by one by plugging in the bound of $m$ and $R_w, R_b$ and show that they can be bounded by $\tilde{O}(\exp(-B^2/4)/n^{1/2})$. For the second term, we have

$$O\left(\frac{\sqrt{n}\exp(-B^2/2)}{\lambda}\left(\frac{\exp(-B^2/2)\log(n/\delta)}{m}\right)^{1/4}\right) = O\left(\frac{\sqrt{\lambda}\exp(-B^2/8)\log^{1/4}(n/\delta)}{n}\right).$$

For the third term, we have

$$O\left(\frac{\sqrt{n}(R_w + R_b)}{\lambda\exp(B^2/2)}\right) = O\left(\frac{\sqrt{n}}{\lambda\exp(B^2/2)}\frac{\sqrt{n}(1 + (\exp(-B^2/2) + 1/m)\log^3(2mn/\delta))^{1/4}}{m^{1/4}\lambda^{1/2}}\right)$$

$$= O\left(\frac{n}{\exp(B^2/2)n^{6/4}\exp(-B^2/4)}\right)$$

$$= O\left(\frac{\exp(-B^2/4)}{n^{1/2}}\right).$$

For the fourth term, we have

$$\frac{\sqrt{n}}{\lambda^2} \cdot O\left(\exp(-B^2/2)\sqrt{\frac{\log(n^2/\delta)}{m}} + (R_w + R_b)\exp(-3B^2/4)\right)$$

$$= O\left(\frac{\lambda\sqrt{\log(n/\delta)}}{n^{2.5}}\right) + O\left(\frac{\exp(-B^2/4)}{n^{1.5}}\right).$$

For the last term, we have

$$O\left(\frac{n^2(1 + (\exp(-B^2/2) + 1/m)\log^3(2mn/\delta))\exp(-B^2/2)}{\sqrt{m}\lambda^2}\right)$$

$$= O\left(\frac{\lambda\sqrt{1 + (\exp(-B^2/2) + 1/m)\log^3(2mn/\delta)}}{n}\right).$$

Recall our discussion on $\lambda$ in Section 3.4 that $\lambda = \lambda_0\exp(-B^2/2) \le 1$ for some $\lambda_0$ independent of $B$. Putting them together, we get the desired upper bound for $\mathcal{R}(\mathcal{F})$, and the theorem is then proved. □

**Lemma C.6.** *Assume the choice of $R_w, R_b, m$ in Theorem A.18. Given $R > 0$, with probability at least $1 - e^{-\Omega(n)}$ over the random initialization of $W(0), a$, the following function class*

$$\mathcal{F}_{R_w, R_b, R} = \{f(W, a, b) : \|W - W(0)\|_{2,\infty} \le R_w, \ \|b - b(0)\|_{\infty} \le R_b,$$
$$\|\text{vec}([W, b] - [W(0), b(0)])\| \le R\}$$

*has empirical Rademacher complexity bounded as*

$$\mathcal{R}_S(\mathcal{F}_{R_w, R_b, R}) \le R\sqrt{\frac{8\exp(-B^2/2)}{n}} + 4c(R_w + R_b)^2\sqrt{m}\exp(-B^2/2).$$

*Proof.* We need to upper bound $\mathcal{R}_S(\mathcal{F}_{R_w, R_b, R})$. Define the events

$$A_{r,i} = \{|w_r(0)^\top x_i - b_r(0)| \le R_w + R_b\}, \ i \in [n], \ r \in [m]$$

and a shorthand $\mathbb{I}(w_r(0)^\top x_i - B \ge 0) = \mathbb{I}_{r,i}(0)$. Then,

$$\sum_{i=1}^{n}\epsilon_i\sum_{r=1}^{m}a_r\sigma(w_r^\top x_i - b_r) - \sum_{i=1}^{n}\epsilon_i\sum_{r=1}^{m}a_r\mathbb{I}_{r,i}(0)(w_r^\top x_i - b_r)$$

$$= \sum_{i=1}^{n}\sum_{r=1}^{m}\epsilon_i a_r\left(\sigma(w_r^\top x_i - b_r) - \mathbb{I}_{r,i}(0)(w_r^\top x_i - b_r)\right)$$

$$= \sum_{i=1}^{n}\sum_{r=1}^{m}\mathbb{I}(A_{r,i})\epsilon_i a_r\left(\sigma(w_r^\top x_i - b_r) - \mathbb{I}_{r,i}(0)(w_r^\top x_i - b_r)\right)$$

$$= \sum_{i=1}^{n}\sum_{r=1}^{m}\mathbb{I}(A_{r,i})\epsilon_i a_r\left(\sigma(w_r^\top x_i - b_r) - \mathbb{I}_{r,i}(0)(w_r(0)^\top x_i - b_r(0)) - \mathbb{I}_{r,i}(0)((w_r - w_r(0))^\top x_i - (b_r - b_r(0)))\right)$$

$$= \sum_{i=1}^{n}\sum_{r=1}^{m}\mathbb{I}(A_{r,i})\epsilon_i a_r\left(\sigma(w_r^\top x_i - b_r) - \sigma(w_r(0)^\top x_i - b_r(0)) - \mathbb{I}_{r,i}(0)((w_r - w_r(0))^\top x_i - (b_r - b_r(0)))\right)$$

$$\le \sum_{i=1}^{n}\sum_{r=1}^{m}\mathbb{I}(A_{r,i})2(R_w + R_b),$$

where the second equality is due to the fact that $\sigma(w_r^\top x_i - b_r) = \mathbb{I}_{r,i}(0)(w_r^\top x_i - b_r)$ if $r \notin A_{r,i}$. Thus, the Rademacher complexity can be bounded as

$\mathcal{R}_S(\mathcal{F}_{R_w, R_b, R})$

$$= \frac{1}{n}\mathbb{E}_\epsilon\left[\sup_{\substack{\|W-W(0)\|_{2,\infty}\le R_w, \ \|b-b(0)\|_{\infty}\le R_b, \\ \|\text{vec}([W,b]-[W(0),b(0)])\|\le R}}\sum_{i=1}^{n}\epsilon_i\sum_{r=1}^{m}\frac{a_r}{\sqrt{m}}\sigma(w_r^\top x_i - b_r)\right]$$

$$\le \frac{1}{n}\mathbb{E}_\epsilon\left[\sup_{\substack{\|W-W(0)\|_{2,\infty}\le R_w, \ \|b-b(0)\|_{\infty}\le R_b, \\ \|\text{vec}([W,b]-[W(0),b(0)])\|\le R}}\sum_{i=1}^{n}\epsilon_i\sum_{r=1}^{m}\frac{a_r}{\sqrt{m}}\mathbb{I}_{r,i}(0)(w_r^\top x_i - b_r)\right] + \frac{2(R_w + R_b)}{n\sqrt{m}}\sum_{i=1}^{n}\sum_{r=1}^{m}\mathbb{I}(A_{r,i})$$

$$= \frac{1}{n}\mathbb{E}_\epsilon\left[\sup_{\|\text{vec}([W,b]-[W(0),b(0)])\|\le R}\text{vec}([W, b])^\top Z(0)\epsilon\right] + \frac{2(R_w + R_b)}{n\sqrt{m}}\sum_{i=1}^{n}\sum_{r=1}^{m}\mathbb{I}(A_{r,i})$$

$$= \frac{1}{n}\mathbb{E}_\epsilon\left[\sup_{\|\text{vec}([W,b]-[W(0),b(0)])\|\le R}\text{vec}([W, b] - [W(0), b(0)])^\top Z(0)\epsilon\right] + \frac{2(R_w + R_b)}{n\sqrt{m}}\sum_{i=1}^{n}\sum_{r=1}^{m}\mathbb{I}(A_{r,i})$$

$$\le \frac{1}{n}\mathbb{E}_\epsilon[R\|Z(0)\epsilon\|_2] + \frac{2(R_w + R_b)}{n\sqrt{m}}\sum_{i=1}^{n}\sum_{r=1}^{m}\mathbb{I}(A_{r,i})$$

$$\le \frac{R}{n}\sqrt{\mathbb{E}_\epsilon[\|Z(0)\epsilon\|_2^2]} + \frac{2(R_w + R_b)}{n\sqrt{m}}\sum_{i=1}^{n}\sum_{r=1}^{m}\mathbb{I}(A_{r,i})$$

$$= \frac{R}{n} \|Z(0)\|_F + \frac{2(R_w + R_b)}{n\sqrt{m}} \sum_{i=1}^{n} \sum_{r=1}^{m} \mathbb{I}(A_{r,i}),$$

where we recall the definition of the matrix

$$Z(0) = \frac{1}{\sqrt{m}} \begin{bmatrix} \mathbb{I}_{1,1}(0)a_1[x_1^\top, -1]^\top & \cdots & \mathbb{I}_{1,n}(0)a_1[x_n^\top, -1]^\top \\ \vdots & & \vdots \\ \mathbb{I}_{m,1}(0)a_m[x_1^\top, -1]^\top & \cdots & \mathbb{I}_{m,n}(0)a_m[x_n^\top, -1]^\top \end{bmatrix} \in \mathbb{R}^{m(d+1) \times n}.$$

By Lemma A.19, we have $\|Z(0)\|_F \le \sqrt{8n \exp(-B^2/2)}$ and by Corollary A.5, we have

$$\mathbb{P}\left[\forall i \in [n]: \sum_{r=1}^{m} \mathbb{I}(A_{r,i}) \le 2mc(R_w + R_b)\exp(-B^2/2)\right] \ge 1 - e^{-\Omega(n)}.$$

Thus, with probability at least $1 - e^{-\Omega(n)}$, we have

$$\mathcal{R}_S(\mathcal{F}_{R_w, R_b, R}) \le R\sqrt{\frac{8\exp(-B^2/2)}{n}} + 4c(R_w + R_b)^2\sqrt{m}\exp(-B^2/2).$$

$\square$

## C.2 ANALYSIS OF RADIUS

**Theorem C.7.** *Assume the parameter settings in Theorem A.18. With probability at least $1 - \delta - e^{-\Omega(n)}$ over the initialization we have*

$$f(k) - y = -(I - \eta H^\infty)^k y \pm e(k),$$

*where*

$$\|e(k)\|_2 = k(1 - \eta\lambda/4)^{(k-1)/2}\eta n^{3/2} \cdot O\left(\exp(-B^2/4)\sqrt{\frac{\log(n^2/\delta)}{m}} + (R_w + R_b)\exp(-B^2/2)\right).$$

*Proof.* Before we start, we assume all the events needed in Theorem A.18 succeed, which happens with probability at least $1 - \delta - e^{-\Omega(n)}$.

Recall the no-flipping set $S_i$ in Definition A.3. We have

$$f_i(k+1) - f_i(k) = \frac{1}{\sqrt{m}} \sum_{r=1}^{m} a_r[\sigma(w_r(k+1)^\top x_i - b_r(k+1)) - \sigma(w_r(k)^\top x_i - b_r(k))]$$

$$= \frac{1}{\sqrt{m}} \sum_{r \in S_i} a_r[\sigma(w_r(k+1)^\top x_i - b_r(k+1)) - \sigma(w_r(k)^\top x_i - b_r(k))]$$

$$+ \underbrace{\frac{1}{\sqrt{m}} \sum_{r \in \overline{S}_i} a_r[\sigma(w_r(k+1)^\top x_i - b_r(k+1)) - \sigma(w_r(k)^\top x_i - b_r(k))]}_{\epsilon_i(k)}.$$

(5)

Now, to upper bound the second term $\epsilon_i(k)$,

$$|\epsilon_i(k)| = \left|\frac{1}{\sqrt{m}} \sum_{r \in \overline{S}_i} a_r[\sigma(w_r(k+1)^\top x_i - b_r(k+1)) - \sigma(w_r(k)^\top x_i - b_r(k))]\right|$$

$$\le \frac{1}{\sqrt{m}} \sum_{r \in \overline{S}_i} |w_r(k+1)^\top x_i - b_r(k+1) - (w_r(k)^\top x_i - b_r(k))|$$

$$\le \frac{1}{\sqrt{m}} \sum_{r \in \overline{S}_i} \|w_r(k+1) - w_r(k)\|_2 + |b_r(k+1) - b_r(k)|$$

$$
= \frac{1}{\sqrt{m}} \sum_{r \in \overline{S}_i} \left\| \frac{\eta}{\sqrt{m}} a_r \sum_{j=1}^{n} (f_j(k) - y_j) \mathbb{I}_{r,j}(k) x_j \right\|_2 + \left| \frac{\eta}{\sqrt{m}} a_r \sum_{j=1}^{n} (f_j(k) - y_j) \mathbb{I}_{r,j}(k) \right|
$$

$$
\leq \frac{2\eta}{m} \sum_{r \in \overline{S}_i} \sum_{j=1}^{n} |f_j(k) - y_j|
$$

$$
\leq \frac{2\eta \sqrt{n} |\overline{S}_i|}{m} \|f(k) - y\|_2
$$

$$
\Rightarrow \|\epsilon\|_2 = \sqrt{\sum_{i=1}^{n} \frac{4\eta^2 n |\overline{S}_i|^2}{m^2} \|f(k) - y\|_2^2} \leq \eta n O((R_w + R_b) \exp(-B^2/2)) \|f(k) - y\|_2 \quad (6)
$$

where we apply Corollary A.5 in the last inequality. To bound the first term,

$$
\frac{1}{\sqrt{m}} \sum_{r \in S_i} a_r [\sigma(w_r(k+1)^\top x_i - b_r(k+1)) - \sigma(w_r(k)^\top x_i - b_r(k))]
$$

$$
= \frac{1}{\sqrt{m}} \sum_{r \in S_i} a_r \mathbb{I}_{r,i}(k) \left( (w_r(k+1) - w_r(k))^\top x_i - (b_r(k+1) - b_r(k)) \right)
$$

$$
= \frac{1}{\sqrt{m}} \sum_{r \in S_i} a_r \mathbb{I}_{r,i}(k) \left( \left( -\frac{\eta}{\sqrt{m}} a_r \sum_{j=1}^{n} (f_j(k) - y_j) \mathbb{I}_{r,j}(k) x_j \right)^\top x_i - \frac{\eta}{\sqrt{m}} a_r \sum_{j=1}^{n} (f_j(k) - y_j) \mathbb{I}_{r,j}(k) \right)
$$

$$
= \frac{1}{\sqrt{m}} \sum_{r \in S_i} a_r \mathbb{I}_{r,i}(k) \left( -\frac{\eta}{\sqrt{m}} a_r \sum_{j=1}^{n} (f_j(k) - y_j) \mathbb{I}_{r,j}(k) (x_j^\top x_i + 1) \right)
$$

$$
= -\eta \sum_{j=1}^{n} (f_j(k) - y_j) \frac{1}{m} \sum_{r \in S_i} \mathbb{I}_{r,i}(k) \mathbb{I}_{r,j}(k) (x_j^\top x_i + 1)
$$

$$
= -\eta \sum_{j=1}^{n} (f_j(k) - y_j) H_{ij}(k) + \underbrace{\eta \sum_{j=1}^{n} (f_j(k) - y_j) \frac{1}{m} \sum_{r \in \overline{S}_i} \mathbb{I}_{r,i}(k) \mathbb{I}_{r,j}(k) (x_j^\top x_i + 1)}_{\epsilon'_i(k)} \quad (7)
$$

where we can upper bound $|\epsilon'_i(k)|$ as

$$
|\epsilon'_i(k)| \leq \frac{2\eta}{m} |\overline{S}_i| \sum_{j=1}^{n} |f_j(k) - y_j| \leq \frac{2\eta \sqrt{n} |\overline{S}_i|}{m} \|f(k) - y\|_2
$$

$$
\Rightarrow \|\epsilon'\|_2 = \sqrt{\sum_{i=1}^{n} \frac{4\eta^2 n |\overline{S}_i|^2}{m^2} \|f(k) - y\|_2^2} \leq \eta n O((R_w + R_b) \exp(-B^2/2)) \|f(k) - y\|_2 .
$$

$$(8)$$

Combining Equation (5), Equation (6), Equation (7) and Equation (8), we have

$$
f_i(k+1) - f_i(k) = -\eta \sum_{j=1}^{n} (f_j(k) - y_j) H_{ij}(k) + \epsilon_i(k) + \epsilon'_i(k)
$$

$$
\Rightarrow f(k+1) - f(k) = -\eta H(k)(f(k) - y) + \epsilon(k) + \epsilon'(k)
$$

$$
= -\eta H^\infty (f(k) - y) + \underbrace{\eta (H^\infty - H(k))(f(k) - y) + \epsilon(k) + \epsilon'(k)}_{\zeta(k)}
$$

$$
\Rightarrow f(k) - y = (I - \eta H^\infty)^k (f(0) - y) + \sum_{t=0}^{k-1} (I - \eta H^\infty)^t \zeta(k - 1 - t)
$$

$$
= -(I - \eta H^\infty)^k y + \underbrace{(I - \eta H^\infty)^k f(0) + \sum_{t=0}^{k-1} (I - \eta H^\infty)^t \zeta(k - 1 - t)}_{e(k)} .
$$

Now the rest of the proof bounds the magnitude of $e(k)$. From Lemma A.2 and Lemma A.6, we have

$$\|H^\infty - H(k)\|_2 \leq \|H(0) - H^\infty\|_2 + \|H(0) - H(k)\|_2$$

$$= O\left(n\exp(-B^2/4)\sqrt{\frac{\log(n^2/\delta)}{m}}\right) + O(n(R_w + R_b)\exp(-B^2/2)).$$

Thus, we can bound $\zeta(k)$ as

$$\|\zeta(k)\|_2 \leq \eta\|H^\infty - H(k)\|_2\|f(k) - y\|_2 + \|\epsilon(k)\|_2 + \|\epsilon'(k)\|_2$$

$$= O\left(\eta n\left(\exp(-B^2/4)\sqrt{\frac{\log(n^2/\delta)}{m}} + (R_w + R_b)\exp(-B^2/2)\right)\right)\|f(k) - y\|_2.$$

Notice that $\|H^\infty\|_2 \leq \mathrm{Tr}(H^\infty) \leq n$ since $H^\infty$ is symmetric. By Theorem A.18, we pick $\eta = O(\lambda/n^2) \ll 1/\|H^\infty\|_2$ and, with probability at least $1 - \delta - e^{-\Omega(n)}$ over the random initialization, we have $\|f(k) - y\|_2 \leq (1 - \eta\lambda/4)^{k/2}\|f(0) - y\|_2$.

Since we are using symmetric initialization, we have $(I - \eta H^\infty)^k f(0) = 0$.

Thus,

$$\|e(k)\|_2 = \left\|\sum_{t=0}^{k-1}(I - \eta H^\infty)^t \zeta(k-1-t)\right\|_2$$

$$\leq \sum_{t=0}^{k-1}\|I - \eta H^\infty\|_2^t\|\zeta(k-1-t)\|_2$$

$$\leq \sum_{t=0}^{k-1}(1 - \eta\lambda)^t \eta n O\left(\exp(-B^2/4)\sqrt{\frac{\log(n^2/\delta)}{m}} + (R_w + R_b)\exp(-B^2/2)\right)\|f(k-1-t) - y\|_2$$

$$\leq \sum_{t=0}^{k-1}(1 - \eta\lambda)^t \eta n O\left(\exp(-B^2/4)\sqrt{\frac{\log(n^2/\delta)}{m}} + (R_w + R_b)\exp(-B^2/2)\right)$$

$$\cdot (1 - \eta\lambda/4)^{(k-1-t)/2}\|f(0) - y\|_2$$

$$\leq k(1 - \eta\lambda/4)^{(k-1)/2}\eta n O\left(\exp(-B^2/4)\sqrt{\frac{\log(n^2/\delta)}{m}} + (R_w + R_b)\exp(-B^2/2)\right)\|f(0) - y\|_2$$

$$\leq k(1 - \eta\lambda/4)^{(k-1)/2}\eta n^{3/2} O\left(\left(\exp(-B^2/4)\sqrt{\frac{\log(n^2/\delta)}{m}} + (R_w + R_b)\exp(-B^2/2)\right)\right.$$

$$\left.\cdot\left(\sqrt{1 + (\exp(-B^2/2) + 1/m)\log^3(2mn/\delta)}\right)\right)$$

$$= k(1 - \eta\lambda/8)^{k-1}\eta n^{3/2} O\left(\exp(-B^2/4)\sqrt{\frac{\log(n^2/\delta)}{m}} + (R_w + R_b)\exp(-B^2/2)\right).$$

$\square$

**Lemma C.8.** *Assume the parameter settings in Theorem A.18. Then with probability at least $1 - \delta - e^{-\Omega(n)}$ over the random initialization, we have for all $k \geq 0$,*

$$\|[W, b](k) - [W, b](0)\|_F \leq \sqrt{y^\top (H^\infty)^{-1} y} + O\left(\frac{n}{\lambda}\left(\frac{\exp(-B^2/2)\log(n/\delta)}{m}\right)^{1/4}\right)$$

$$+ O\left(\frac{n\sqrt{R\exp(-B^2/2)}}{\lambda}\right)$$

$$+ \frac{n}{\lambda^2}\cdot O\left(\exp(-B^2/4)\sqrt{\frac{\log(n^2/\delta)}{m}} + R\exp(-B^2/2)\right)$$

*where $R = R_w + R_b$.*

*Proof.* Before we start, we assume all the events needed in Theorem A.18 succeed, which happens with probability at least $1 - \delta - e^{-\Omega(n)}$.

$$
\begin{aligned}
&\text{vec}([W, b](K)) - \text{vec}([W, b](0)) \\
&= \sum_{k=0}^{K-1} \text{vec}([W, b](k+1)) - \text{vec}([W, b](k)) \\
&= - \sum_{k=0}^{K-1} Z(k)(u(k) - y) \\
&= \sum_{k=0}^{K-1} \eta Z(k)((I - \eta H^\infty)^k y - e(k)) \\
&= \sum_{k=0}^{K-1} \eta Z(k)(I - \eta H^\infty)^k y - \sum_{k=0}^{K-1} \eta Z(k)e(k) \\
&= \underbrace{\sum_{k=0}^{K-1} \eta Z(0)(I - \eta H^\infty)^k y}_{T_1} + \underbrace{\sum_{k=0}^{K-1} \eta (Z(k) - Z(0))(I - \eta H^\infty)^k y}_{T_2} - \underbrace{\sum_{k=0}^{K-1} \eta Z(k)e(k)}_{T_3}. \quad (9)
\end{aligned}
$$

Now, by Lemma A.6, we have $\|Z(k) - Z(0)\|_F \le O(\sqrt{nR\exp(-B^2/2)})$ which implies

$$
\begin{aligned}
\|T_2\|_2 &= \left\| \sum_{k=0}^{K-1} \eta (Z(k) - Z(0))(I - \eta H^\infty)^k y \right\|_2 \\
&\le \sum_{k=0}^{K-1} \eta \cdot O(\sqrt{nR\exp(-B^2/2)}) \|I - \eta H^\infty\|_2^k \|y\|_2 \\
&\le \eta \cdot O(\sqrt{nR\exp(-B^2/2)}) \sum_{k=0}^{K-1} (1 - \eta\lambda)^k \sqrt{n} \\
&= O\left( \frac{n\sqrt{R\exp(-B^2/2)}}{\lambda} \right). \quad (10)
\end{aligned}
$$

By $\|Z(k)\|_2 \le \|Z(k)\|_F \le \sqrt{2n}$, we get

$$
\begin{aligned}
\|T_3\|_2 &= \left\| \sum_{k=0}^{K-1} \eta Z(k)e(k) \right\|_2 \\
&\le \sum_{k=0}^{K-1} \eta\sqrt{2n} \left( k(1 - \eta\lambda/8)^{k-1}\eta n^{3/2} O\left( \exp(-B^2/4)\sqrt{\frac{\log(n^2/\delta)}{m}} + R\exp(-B^2/2) \right) \right) \\
&= \frac{n}{\lambda^2} \cdot O\left( \exp(-B^2/4)\sqrt{\frac{\log(n^2/\delta)}{m}} + R\exp(-B^2/2) \right). \quad (11)
\end{aligned}
$$

Define $T = \eta \sum_{k=0}^{K-1} (I - \eta H^\infty)^k$. By Lemma A.2, we know $\|H(0) - H^\infty\|_2 \le O(n\exp(-B^2/4)\sqrt{\frac{\log(n/\delta)}{m}})$ and this implies

$$
\begin{aligned}
\|T_1\|_2^2 &= \left\| \sum_{k=0}^{K-1} \eta Z(0)(I - \eta H^\infty)^k y \right\|_2^2 \\
&= \|Z(0)Ty\|_2^2
\end{aligned}
$$

$$\begin{aligned}
&= y^\top T Z(0)^\top Z(0) T y \\
&= y^\top T H(0) T y \\
&\le y^\top T H^\infty T y + \|H(0) - H^\infty\|_2 \|T\|_2^2 \|y\|_2^2 \\
&\le y^\top T H^\infty T y + O\left(n \exp(-B^2/4)\sqrt{\frac{\log(n/\delta)}{m}}\right)\left(\eta \sum_{k=0}^{K-1}(1-\eta\lambda)^k\right)^2 n \\
&= y^\top T H^\infty T y + O\left(\frac{n^2 \exp(-B^2/4)}{\lambda^2}\sqrt{\frac{\log(n/\delta)}{m}}\right).
\end{aligned}$$

Let $H^\infty = U\Sigma U^\top$ be the eigendecomposition. Then

$$T = U\left(\eta \sum_{k=0}^{K-1}(I-\eta\Sigma)^k\right)U^\top = U((I-(I-\eta\Sigma)^K)\Sigma^{-1})U^\top$$

$$\Rightarrow T H^\infty T = U((I-(I-\eta\Sigma)^K)\Sigma^{-1})^2 \Sigma U^\top = U(I-(I-\eta\Sigma)^K)^2 \Sigma^{-1} U^\top \preceq U\Sigma^{-1}U^\top = (H^\infty)^{-1}.$$

Thus,

$$\begin{aligned}
\|T_1\|_2^2 &= \left\|\sum_{k=0}^{K-1}\eta Z(0)(I-\eta H^\infty)^k y\right\|_2 \\
&\le \sqrt{y^\top (H^\infty)^{-1}y + O\left(\frac{n^2 \exp(-B^2/4)}{\lambda^2}\sqrt{\frac{\log(n/\delta)}{m}}\right)} \\
&\le \sqrt{y^\top (H^\infty)^{-1}y} + O\left(\frac{n}{\lambda}\left(\frac{\exp(-B^2/2)\log(n/\delta)}{m}\right)^{1/4}\right).
\end{aligned} \tag{12}$$

Finally, plugging in the bounds in Equation (9), Equation (12), Equation (10), and Equation (11), we have

$$\begin{aligned}
&\|[W,b](K) - [W,b](0)\|_F \\
&= \|\mathrm{vec}([W,b](K)) - \mathrm{vec}([W,b](0))\|_2 \\
&\le \sqrt{y^\top (H^\infty)^{-1}y} + O\left(\frac{n}{\lambda}\left(\frac{\exp(-B^2/2)\log(n/\delta)}{m}\right)^{1/4}\right) \\
&\quad + O\left(\frac{n\sqrt{R\exp(-B^2/2)}}{\lambda}\right) \\
&\quad + \frac{n}{\lambda^2}\cdot O\left(\exp(-B^2/4)\sqrt{\frac{\log(n^2/\delta)}{m}} + R\exp(-B^2/2)\right).
\end{aligned}$$

$\square$

# D  PROBABILITY

**Lemma D.1** (Bernstein's Inequality). *Assume $Z_1,\ldots,Z_n$ are $n$ i.i.d. random variables with $\mathbb{E}[Z_i] = 0$ and $|Z_i| \le M$ for all $i \in [n]$ almost surely. Let $Z = \sum_{i=1}^n Z_i$. Then, for all $t > 0$,*

$$\mathbb{P}[Z > t] \le \exp\left(-\frac{t^2/2}{\sum_{j=1}^n \mathbb{E}[Z_j^2] + Mt/3}\right) \le \exp\left(-\min\left\{\frac{t^2}{2\sum_{j=1}^n \mathbb{E}[Z_j^2]}, \frac{t}{2M}\right\}\right)$$

*which implies with probability at least $1 - \delta$,*

$$Z \le \sqrt{2\sum_{j=1}^n \mathbb{E}[Z_j^2]\log\frac{1}{\delta}} + 2M\log\frac{1}{\delta}.$$

**Lemma D.2** (Matrix Chernoff Bound, (Tropp et al., 2015)). *Let $X_1, \ldots, X_m \in \mathbb{R}^{n \times n}$ be $m$ independent random Hermitian matrices. Assume that $0 \preceq X_i \preceq L \cdot I$ for some $L > 0$ and for all $i \in [m]$. Let $X := \sum_{i=1}^m X_i$. Then, for $\epsilon \in (0, 1]$, we have*

$$\mathbb{P}\left[\lambda_{\min}(X) \leq \epsilon \lambda_{\min}(\mathbb{E}[X])\right] \leq n \cdot \exp(-(1 - \epsilon)^2 \lambda_{\min}(\mathbb{E}[X])/(2L)).$$

**Lemma D.3** ((Li & Shao, 2001, Theorem 3.1) with Improved Upper Bound for Gaussian). *Let $b > 0$ and $r > 0$. Then,*

$$\exp(-b^2/2) \mathop{\mathbb{P}}_{w \sim \mathcal{N}(0,1)}[|w| \leq r] \leq \mathop{\mathbb{P}}_{w \sim \mathcal{N}(0,1)}[|x - b| \leq r] \leq 2r \cdot \frac{1}{\sqrt{2\pi}} \exp(-(\max\{b - r, 0\})^2/2).$$

*Proof.* To prove the upper bound, we have

$$\mathop{\mathbb{P}}_{w \sim \mathcal{N}(0,1)}[|x - b| \leq r] = \int_{b-r}^{b+r} \frac{1}{\sqrt{2\pi}} \exp(-x^2/2) \, dx \leq 2r \cdot \frac{1}{\sqrt{2\pi}} \exp(-(\max\{b - r, 0\})^2/2).$$

$\square$

**Lemma D.4** (Anti-concentration of Gaussian). *Let $Z \sim \mathcal{N}(0, \sigma^2)$. Then for $t > 0$,*

$$\mathbb{P}[|Z| \leq t] \leq \frac{2t}{\sqrt{2\pi}\sigma}.$$

## E  THE BENEFIT OF CONSTANT INITIALIZATION OF BIASES

In short, the benefit of constant initialization of biases lies in inducing sparsity in activation and thus reducing the per step training cost. This is the main motivation of our work on studying sparsity from a deep learning theory perspective. Since our convergence shows that sparsity doesn't change convergence rate, the total training cost is also reduced.

To address the width's dependence on $B$, our argument goes like follows. In practice, people set up neural network models by first picking a neural network of some pre-chosen size and then choose other hyper-parameters such as learning rate, initialization scale, etc. In our case, the hyper-parameter is the bias initialization. Thus, *the network width is picked before $B$.* Let's say we want to apply our theoretical result to guide our practice. Since we usually don't know the exact data separation and the minimum eigenvalue of the NTK, we don't have a good estimate on the exact width needed for the network to converge and generalize. We may pick a network with width that is much larger than needed (e.g. we pick a network of width $\Omega(n^{12})$ whereas only $\Omega(n^4)$ is needed; this is possible because the smallest eigenvalue of NTK can range from $[\Omega(1/n^2), O(1)]$). Also, it is an empirical observation that the neural networks used in practice are very overparameterized and there is always room for sparsification. If the network width is very large, then per step gradient descent is very costly since the cost scales linearly with width and can be improved to scale linearly with the number of active neurons if done smartly. If the bias is initialized to zero (as people usually do in practice), then the number of active neurons is $O(m)$. However, since we can sparsify the neural network activation by non-zero bias initialization, the number of active neurons can scale *sub-linearly* in $m$. Thus, if the neural network width we choose at the beginning is much larger than needed, then we are indeed able to obtain total training cost reduction by this initialization. The above is an informal description of the result proven in (Song et al., 2021a) and the message is *sparsity can help reduce the per step training cost.* If the network width is pre-chosen, then the lower bound on network width $m \geq \tilde{\Omega}(\lambda_0^{-4} n^4 \exp(B^2))$ in Theorem 3.1 can be translated into an upper bound on bias initialization: $B \leq \tilde{O}(\sqrt{\log \frac{\lambda_0^4 m}{n^4}})$ if $m \geq \tilde{\Omega}(\lambda_0^{-4} n^4)$. This would be a more appropriate interpretation of our result. Note that this is different from how Theorem 3.1 is presented: first pick $B$ and then choose $m$; since $m$ is picked later, $m$ can always satisfy $B \leq \sqrt{0.5 \log m}$ and $m \geq \tilde{\Omega}(\lambda_0^{-4} n^4 \exp(B^2))$. Of course, we don't know the best (largest) possible $B$ that works but as long as we can get some $B$ to work, we can get computational gain from sparsity.

In summary, sparsity can reduce the per step training cost since we don't know the exact width needed for the network to converge and generalize. Our result should be interpreted as an upper bound on $B$ since the width is always chosen before $B$ in practice.

