# OpenReview forum: "Sharper Analysis of Sparsely Activated Wide Neural Networks with Trainable Biases"
_ICLR.cc/2023/Conference — Submitted to ICLR 2023_

### Official Review · Reviewer_neAa · 2022-10-21

**Confidence:** 4
**Correctness:** 4
**Technical Novelty And Significance:** 2
**Empirical Novelty And Significance:** Not applicable
**Recommendation:** 3

**Clarity, Quality, Novelty And Reproducibility:**

**Clarity**

I do have some small questions/comments about the clarity of the writing.

In the abstract:
* perhaps make the following edit “towards zero in linear rate” —> “towards zero at a linear rate”

* You write “and same-as-previous (ignoring logarithmic factors) generalization bound”.  I think by "same-as-previous" you mean that you recover prior bounds.  Please perhaps rephrase as this sentence is a bit confusing to read.
* Consider rephrasing “Up to our knowledge” as “To our knowledge”

page 1: Consider making the following edit “The literature of sparse neural network” —> “The literature of sparse neural networks”

page 2: “and same as-previous (up to logarithmic factors) generalization bound.” Maybe this sentence can also be rewritten.

page 4: “our study handles the trainable bias in the convergence analysis (which is the first of such a type).”  Can you specify exactly what you mean by this sentence?  There are existing works that train the bias's in convergence analysis.  Which aspect of your work is the first in this regard?

page 4: You state "To elaborate, our bound only requires $m \geq \tilde{\Omega}(\lambda_0^{-4} n^4 \exp(B^2))$, as opposed to the bound $m \geq \tilde{\Omega}(\lambda_0^{-4} n^4 B^2 \exp(2B^2)$ in (Song et al.,
2021a, Lemma D.9).  If we take $B = \sqrt{0.25 \log m}$ (as allowed by the theorem), then our lower bound yields a polynomial improvement by a factor of $\Theta((n / \lambda_0)^{8/3})$ , which implies that the neural network width can be much smaller to achieve the same linear convergence”.  Can you explain how you reach this conlusion?  The ratio of the two bounds is $B^2 \exp(B^2)$ which after plugging in $B = \sqrt{0.25 \log m}$ becomes $\frac{1}{4} m^{1/4} \log(m)$.

**Novelty**

My concerns about novelty were outlined in the "Strengths and Weaknesses" section.

**Reproducibility**

Since this is primarily a theoretical work with proofs I do not find reproducibility applicable here.



**Strength And Weaknesses:**

**Strengths:**
This paper attempts to relate sparsity of activation patterns to the optimization and generalization of ReLU networks.  I do believe that sparsity of activation patterns in relation to the optimization dynamics and generalization is indeed an interesting topic that deserves study.

**Weaknesses:**
While I do think sparsity of activations is an interesting topic in relation to optimization and generalization, I think the sparsity studied in this present work is unnatural. Initializing the bias’s to some large constant value is not done in practice, and this work does not make an effort justify this form of initialization to induce sparsity.  While they cite [1] which considers this setting, I think this work should try to justify this to the reader on its own.  The number of neurons required by Theorem 3.1 is super-exponential in the bias initialization value $B$, and the convergence rate and step size decay super-exponentially with $B$.  Thus inducing sparsity by setting $B$ to be large comes at a high cost, and this must at least be explained.

In Theorem 3.1 the requirement on the width $m$ has quadratic dependence on $n$ similar to Theorem 2.3 in [2] (the arxiv version is the one I’m referencing).  The main difference I can see is that in [2] the network does not have bias’s.  My question to the authors is aside from training the bias’s what is the novelty in Theorem 3.1?

Theorem 3.8 appears to be an adaption of Theorem 5.1 in [3].  I do not see what is different here aside from the modification to the NTK that comes from the specific bias initialization.  Also it does not seem that [3] is properly referenced in Theorem 3.8 (also note that the result in [3] is an instantiation of Lemma 22 in [4]).

Theorem 3.11 is essentially outlining a sufficient condition on the labels so that the quantity $y^T (H^\infty)^{-1} y$ is well bounded.  However this data dependent region they define seems quite arbitrary.  $y^T (H^\infty)^{-1} y$ can be viewed as the discrete RKHS norm of the target function.  Improved optimization and generalization for target functions that have small RKHS norm has been investigated in [5] and [6].

In sum, I am not convinced about the contribution of the results of this work in relation to previous works.  The authors emphasize repeatedly that they improve over the results of [1], however I am not convinced that they significantly improve over the works I outline above.  If the authors can provide strong responses to my objections specifying the improvements of their works in relation to these works I will consider changing my recommendation.  However, for now I am inclined to issue a reject recommendation for this submission.

**References:**

[1] Zhao Song, Shuo Yang, and Ruizhe Zhang. Does preprocessing help training over-parameterized neural networks? Advances in Neural Information Processing Systems, 34:22890–22904, 2021a.

[2] Samet Oymak and Mahdi Soltanolkotabi.  Towards moderate overparameterization: global convergence guarantees for training shallow neural networks.  Published in: IEEE Journal on Selected Areas in Information Theory, 1(1):84–105, 2020.
Preprint: https://arxiv.org/pdf/1902.04674.pdf

[3] Sanjeev Arora, Simon Du, Wei Hu, Zhiyuan Li, and Ruosong Wang. Fine-grained analysis of optimization and generalization for overparameterized two-layer neural networks. In Kamalika Chaudhuri and Ruslan Salakhutdinov (eds.), Proceedings of the 36th International Conference on Machine Learning, volume 97 of Proceedings of Machine Learning Research, pp. 322–332. PMLR, 09–15 Jun 2019a. URL https://proceedings.mlr.press/v97/arora19a. html.

[4] Peter L. Bartlett and Shahar Mendelson. Rademacher and Gaussian complexities: Risk bounds and structural results. J. Mach. Learn. Res., 3(null):463–482, mar 2003. ISSN 1532-4435.

[5] Ziwei Ji and Matus Telgarsky. Polylogarithmic width suffices for gradient descent to achieve arbitrarily small test error with shallow ReLU networks. In International Conference on Learning Representations, 2020. URL https://openreview.net/forum?id=HygegyrYwH.

[6] Lili Su and Pengkun Yang. On learning over-parameterized neural networks: A functional approximation perspective. In H. Wallach, H. Larochelle, A. Beygelzimer, F. d'Alche-Buc, E. Fox, and ´ R. Garnett (eds.), Advances in Neural Information Processing Systems, volume 32. Curran Associates, Inc., 2019. URL https://proceedings.neurips.cc/paper/2019/file/ 253f7b5d921338af34da817c00f42753-Paper.pdf.


**Summary Of The Paper:**

 This paper studies the convergence and generalization of shallow ReLU networks with trainable bias’s initialized to a constant value.  They provide a global convergence guarantee and a local Rademacher complexity based generalization bound.


**Summary Of The Review:**

This work attempts to address an interesting topic (the relation between sparse activations and optimization/generalization).  However the type of sparsity they study (initializing bias's to a large contant) feels contrived and is not justified.  Furthermore, I have doubts about the contribution of their work in relation to prior works.  If the authors can provide strong responses to my reservations which show that I missed essential components of their work I will consider raising my recommendation.  However, for now I am issuing a reject recommendation.

---

> ### Author Response · Authors · 2022-11-14
> **Response to Reviewer neAa (1/3)**
>
> We thank Reviewer neAa for the careful review and writing suggestion. We address the reviewer's comments below.
>
> **Improved convergence analysis**
>
> We first want to let the reviewer know that our revision improves our analysis of the convergence rate, and show that sparsity activated network achieves \textit{the same convergence rate} for all sparsity (i.e., all bias initialization value $B$), and thus sparsity doesn't hurt convergence. Especially, the learning rate is now allowed to be chosen from a larger range.
>
> This result is obtained as the following. By utilizing the fact that the network has sparse activation, we improve Lemma A.17 with a sharper bound. This leads to a more relaxed condition on learning rate $\eta$ in Theorem A.18 (same as Theorem 3.1): the upper bound on $\eta$ now allows an additional $\exp(B^2)$ factor which compensates the decrease of the smallest eigenvalue. This leads to better convergence.
>
> **Q:**
> I think the sparsity studied in this present work is unnatural. Initializing the bias’s to some large constant value is not done in practice, and this work does not make an effort justify this form of initialization to induce sparsity.
> The number of neurons required by Theorem 3.1 is super-exponential in the bias initialization value $B$, and the convergence rate and step size decay super-exponentially with $B$. Thus inducing sparsity by setting
> $B$ to be large comes at a high cost, and this must at least be explained.
>
> **A:**
> Many thanks for the question. Given the tools and techniques available in deep learning theory, initializing bias as a constant is a simplified, yet still sensible way to capture the sparse-to-sparse training practice such as [5,6]. Our results based on this simple model can already capture the benefits of sparse training. As we proved in Lemma 3.5, such an initialization yields sparse activation during the entire training. This shows that the computational complexity per step training can be significantly reduced as in [1,2,3,4], capturing the main benefit of inducing sparsity.
> Since we have shown that sparsity doesn't slow down convergence rate (as we describe above), the overall computational cost over the entire training is reduced.
>
> Regarding the network width, we want to point out that the motivation is not that we require a large network to make the sparse training work, which then incurs a large cost. Our motivation follows a different logic. Our starting point is that the network width is typically picked to be large in practice due to various considerations such as expressive capacity, optimization and generalization.
> This thus motivates the sparse training to reduce the computation cost and speedup the training. Thus, our study here implies that as long as we initialize the bias parameter $B$ accordingly, then the computational cost of the training can be significantly reduced while preserving convergence and generalization of the non-sparsified large model. We further added a new section Appendix E to explain the benefit of this initialization in details.
>
>
> **Q:**
> In Theorem 3.1 the requirement on the width $m$ has quadratic dependence on
> $n$ similar to Theorem 2.3 in [7] (the arxiv version is the one I’m referencing). The main difference I can see is that in [7] the network does not have biases. My question to the authors is aside from training the bias’s what is the novelty in Theorem 3.1?
>
> **A:** Without sparse initialization, yes, our result is similar to that in [7]. However, we use a different approach to prove it from what is used in [7]. Our approach is along the same line of [1] with sharper bounds than those in [1], which can be of independent interest for the basic NTK theory. It turns out that our approach can easily accommodate the case when $B\neq 0$ and the bias needs to be trainable.
> We also want to point out to the reviewer that our main contribution in this paper is to provider a tighter analysis for the sparse activation regime.

---

> > ### Author Response · Authors · 2022-11-14
> > **Response to Reviewer neAa (2/3)**
> >
> > **Q:**
> > Theorem 3.8 appears to be an adaption of Theorem 5.1 in [8]. I do not see what is different here aside from the modification to the NTK that comes from the specific bias initialization.
> >
> > **A:** We respectfully disagree here.
> > Theorem 3.8 does not only adapt/generalize Theorem 5.1 in [8] to the sparse activation case,
> > our lower bound on $m$ in Theorem 3.8 (with $B = 0$) also improves the corresponding condition in Theorem 5.1 in [8].
> > Specifically, Theorem 5.1 in [8] presents a network width lower bound of $m \geq \kappa^{-2} \textnormal{poly}(n, \lambda_0^{-1}, \delta^{-1})$ with $\kappa = O(\frac{\lambda_0 \delta}{n})$, where the dependence on $n$ can be estimated to be a very large polynomial, much larger than our result in Theorem 3.8, where the dependence on $n$ is $n^6$. Further note that Theorem G.7 in [9] also improves Theorem 5.1 in [8] and their dependence on $n$ is $n^{16}$.
> > Technically, Theorem 5.1 in [8] uses a rebalancing factor $\kappa$ to balance between the network's initial output and other terms in the Rademacher Complexity. In contrast, we solve this problem by utilizing symmetric initialization to force the network output to be zero at the initialization, which leads to a much cleaner and tighter analysis (a detailed description is in Section 3.3).
> >
> > **Q:**
> > Theorem 3.11 is essentially outlining a sufficient condition on the labels so that the quantity $y^T(H^\infty)^{-1}y$ is well bounded. However this data dependent region they define seems quite arbitrary.
> >
> > **A:** We respectfully disagree here. First of all, the region is not something artificial, but can be achieved by simply shifting the label vector $y$ into the positive orthant (see our comments in Remark 3.12). This is important since the previous bound in the literature will yield a generalization bound of $O(n)$ which is clearly not tight as the loss is \textbf{bounded}, whereas our bound will yield $O(1)$ generalization bound for fixed data separation. We also want to highlight the advantage of our result that if the label vector $y$ is in this region, then the lower bound of the restricted least eigenvalue on this region is better than the lower bound of the least eigenvalue of the entire domain by a factor of $n^2$.
> >
> > We thank the reviewer for pointing out [11,12] and the relation that $y^\top (H^\infty)^{-1} y$ can be viewed as a discretized version of RKHS norm of the target function. However, we don't find results in [11,12] directly connect to our results here.
> >
> > The work [11] uses NTK margin $1/\gamma$ to bound the optimization and generalization performance. However, it is unclear how and whether good NTK margin in [11] implies lower bound of the minimum eigenvalue of the infinite NTK in our paper. Note that their Proposition 5.1 only implies that if $\gamma_1$ is positive, then it implies $\gamma_1$ NTK margin, but there is no result on the other direction, i.e., how minimum eigenvalue can benefit from good NTK margin. Further, [11] considers a totally different setting and assumption from ours: although both works considers one-hidden-layer neural network in the NTK regime, [11] considers binary classification using cross-entropy loss and assume the infinite-width NTK can achieve good margin, whereas our work considers regression setting using mean square loss and only assume the data makes the least eigenvalue of the NTK matrix to be positive (and this can be achieved by data separation). We don't see any direct connection between the results in two studies.
> >
> > The work [12] is closer to our setting where they also study training one-hidden-layer neural network in the NTK regime using the square loss.
> > However, [12] makes an assumption that $f^\star$ can be approximated by some low rank eigenspace of NTK and thus circumvent the smallest eigenvalue of NTK and is essentially restricted to the least eigenvalue within such a low-rank space. But our work doesn't make any distributional assumption on the data (other than the data makes the infinite-width NTK positive definite) nor any structural assumption on $f^\star$. Our work studies the least eigenvalue of the limiting NTK from a geometric perspective through the lens of data separation, which is a much harder problem. Thus, there is not a direct connection between the two studies.
> >
> >
> > **Q:** Can you specify exactly what you mean by this sentence?
> >  “our study handles the trainable bias in the convergence analysis (which is the first of such a type).”
> >
> > **A:** We want to say "our study handles the trainable bias with constant initialization in the convergence analysis (which is the first of such a type)."

---

> > > ### Author Response · Authors · 2022-11-14
> > > **Response to Reviewer neAa (3/3)**
> > >
> > > **Q:** How do you get $\Theta((n / \lambda_0)^{8/3})$ improvement?
> > >
> > > **A:** Take $B = \sqrt{0.25 \log m}$, the bound $m \geq \tilde{\Omega}(\lambda_0^{-4} n^{4} \exp(B^2))$ becomes $m \geq \tilde{\Omega}(\lambda_0^{-4} n^{4} m^{1/4})$ which implies $m \geq \tilde{\Omega}((n/\lambda_0)^{16/3})$.
> > > Taking $B = \sqrt{0.25 \log m}$, the other bound $m \geq \tilde{\Omega}(\lambda_0^{-4} n^{4} B^2 \exp(2 B^2))$ becomes $m \geq \tilde{\Omega}(\lambda_0^{-4} n^{4} m^{1/2})$ which implies $m \geq \tilde{\Omega}((n/\lambda_0)^8)$.
> > > This implies a $\tilde{\Theta}((n / \lambda_0)^{8/3})$ improvement.
> > >
> > > [1] Song, Zhao, Shuo Yang, and Ruizhe Zhang. "Does Preprocessing Help Training Over-parameterized Neural Networks?." Advances in Neural Information Processing Systems 34 (2021): 22890-22904.
> > >
> > > [2] Song, Zhao, Lichen Zhang, and Ruizhe Zhang. "Training Multi-Layer Over-Parametrized Neural Network in Subquadratic Time." arXiv preprint arXiv:2112.07628 (2021).
> > >
> > > [3] Hu, Hang, et al. "Training overparametrized neural networks in sublinear time." arXiv preprint arXiv:2208.04508 (2022).
> > >
> > > [4] Gao, Yeqi, et al. "A Sublinear Adversarial Training Algorithm." arXiv preprint arXiv:2208.05395 (2022).
> > >
> > > [5] Liu, Shiwei, et al. "Sparse training via boosting pruning plasticity with neuroregeneration." Advances in Neural Information Processing Systems 34 (2021): 9908-9922.
> > >
> > > [6] Liu, Shiwei, et al. "Do we actually need dense over-parameterization? in-time over-parameterization in sparse training." International Conference on Machine Learning. PMLR, 2021.
> > >
> > > [7] Oymak, Samet, and Mahdi Soltanolkotabi. "Toward moderate overparameterization: Global convergence guarantees for training shallow neural networks." IEEE Journal on Selected Areas in Information Theory 1.1 (2020): 84-105. Preprint: https://arxiv.org/pdf/1902.04674.pdf
> > >
> > > [8] Arora, Sanjeev, et al. "Fine-grained analysis of optimization and generalization for overparameterized two-layer neural networks." International Conference on Machine Learning. PMLR, 2019. URL https://proceedings.mlr.press/v97/arora19a. html.
> > >
> > > [9] Song, Zhao, and Xin Yang. "Quadratic suffices for over-parametrization via matrix chernoff bound." arXiv preprint arXiv:1906.03593 (2019).
> > >
> > > [10] Ji, Ziwei, and Matus Telgarsky. "Polylogarithmic width suffices for gradient descent to achieve arbitrarily small test error with shallow ReLU networks." International Conference on Learning Representations. 2019. URL https://openreview.net/forum?id=HygegyrYwH.
> > >
> > > [11] Su, Lili, and Pengkun Yang. "On learning over-parameterized neural networks: A functional approximation perspective." Advances in Neural Information Processing Systems 32 (2019). URL https://proceedings.neurips.cc/paper/2019/file/ 253f7b5d921338af34da817c00f42753-Paper.pdf.

---

> > > > ### Comment · Reviewer_neAa · 2022-11-17
> > > > **Response to authors**
> > > >
> > > > I would like to thank the authors for responding to my questions.  The main improvement to the paper in the revision is the improved convergence rate, which is an improvement that I will take into consideration.  The authors primary contribution as they state it is to provide tighter analysis for the sparse activation regime.  In Theorem 3.1 they improve upon the convergence results in [1].  For the Rademacher complexity result they provide a version of Theorem 5.1 in [2] with improved bounds for the sparse activation setting.  While there is value to technical improvements, I am still not convinced about the significance of the contributions in this manuscript relative to prior works.    Furthermore I still have skepticism about the relevance of the form of sparsity addressed in the manuscript, while I recognize that other works consider this setting.  However, I will keep an open mind during discussion with other reviewers.  Nevertheless at the moment I am keeping my original recommendation.
> > > >
> > > > [1] Zhao Song, Shuo Yang, and Ruizhe Zhang. Does preprocessing help training over-parameterized
> > > > neural networks? Advances in Neural Information Processing Systems, 34:22890–22904,
> > > > 2021a.
> > > >
> > > > [2] Sanjeev Arora, Simon Du, Wei Hu, Zhiyuan Li, and Ruosong Wang. Fine-grained analysis of optimization and generalization for overparameterized two-layer neural networks. In Kamalika Chaudhuri and Ruslan Salakhutdinov (eds.), Proceedings of the 36th International Conference on Machine Learning, volume 97 of Proceedings of Machine Learning Research, pp. 322–332. PMLR, 09–15 Jun 2019a. URL https://proceedings.mlr.press/v97/arora19a. html.

---

> > > > > ### Author Response · Authors · 2022-11-17
> > > > > **Thanks, and more concrete questions will be appreciated**
> > > > >
> > > > > Dear Reviewer neAa,
> > > > >
> > > > > We truly appreciate that you already agree we have improved our theoretical clarity and novelty during revision. We are also thankful for your time spent. We are happy/eager to address your remaining concerns, but wouldn't be able to do so if they were not concretely asked.
> > > > >
> > > > > Specifically, we would be very thankful if you could share more detailed thoughts on two brief "concern" comments:
> > > > >
> > > > > - "I am still not convinced about the significance of the contributions in this manuscript relative to prior works": given that you have seen our improvements over important prior arts, e.g., [1,2], we sincerely want to know what questions remain regarding our significance.
> > > > >
> > > > > - "I still have skepticism about the relevance of the form of sparsity addressed in the manuscript, while I recognize that other works consider this setting": this comment is also confusing us.
> > > > > Please kindly refer to the second part of our general response: https://openreview.net/forum?id=G6-oxjbc_mK&noteId=cshE4oxRWeE where we thoroughly discussed (i) the current four combination categories of sparse training - none of which has theoretical work done yet - hence a sparse training theory is new and relevant for the entire field; (ii) recent SOTA empirical works that adopt our specific category of sparse training (dynamic sparse activation training).
> > > > >
> > > > > We thus humbly request more clarifications on your concerns and would be actively addressing them to our best ability. Thank you again!

---

> > > > > > ### Comment · Reviewer_neAa · 2022-11-17
> > > > > > **Response to authors**
> > > > > >
> > > > > > * I meant that versions of these theorems exist in prior work, and the contribution is to improve the bounds in a specific setting.  However, I am still not convinced of the relevance of large bias initialization setting as I mentioned in my original review.
> > > > > > * "recent SOTA empirical works that adopt our specific category of sparse training (dynamic sparse activation training)."  Dynamic sparse activation training https://arxiv.org/pdf/2005.06870.pdf is not sparsity induced by large bias initialization, it's sparsity induced by a dynamic trainable mask.  How is this the same setting?  I do not deny sparse training in general is interesting, such as that induced by iterative magnitude pruning (IMP).  I said sparse training has relevance in my original review.  What I question is sparse training *induced by large bias initialization* is of relevance.  Please show empirical works that use large bias initialization to get improved generalization and I may reconsider.  However I am as of now still unconvinced.

---

> > > > > > > ### Author Response · Authors · 2022-11-18
> > > > > > > **Further response**
> > > > > > >
> > > > > > > We thank Reviewer neAa for the prompt feedback.
> > > > > > >
> > > > > > > **Q1**: I meant that versions of these theorems exist in prior work, and the contribution is to improve the bounds in a specific setting.
> > > > > > >
> > > > > > > **A1**: We respectfully disagree with reviewer's comments as we elaborate below:
> > > > > > >
> > > > > > > * Our theorems should be viewed as a more general version of the previous work in the following sense.
> > > > > > >
> > > > > > > - (a) Our results instantiate to (and in fact **improve**) the previous result in [1] if we set bias to be fixed, but not trainable. As an important implication, our improved bound on the convergence means that the cost is reduced over entire training, whereas the result in [1] shows that such improvement holds only per gradient descent step. Thus, our result is not *merely* an improvement of the old result.
> > > > > > > It has important new implication. Further, we have provided the generalization result for this sparse-activated training setting that [1] doesn't have, which fills an important gap.
> > > > > > >
> > > > > > > - (b) Our results instantiate to (and in fact improve) the previous result in [2] if we set bias to be zero. Further, we have provided a new lower bound on the minimum eigenvalue of the limiting NTK on a restricted region, which is free of explicit dependence on number of training sample $n$.
> > > > > > > This further establishes the vanishing behavior of the least eigenvalue, but such a result is not in [2].
> > > > > > >
> > > > > > > - (c) We also would like to point out that we choose our setting because it makes the least assumption on the data that we know of.
> > > > > > > The only assumption we make on the training data is data separation, instead of assuming the underlying distribution has some desirable properties like sub-Gaussian concentration, moment bounds, etc, or that the underlying true function is low rank, etc.
> > > > > > >
> > > > > > >
> > > > > > > **Q2**: Dynamic sparse activation training is not the sparsity induced by large bias initialization, it's sparsity induced by a dynamic trainable mask. How is this the same setting?
> > > > > > > What I question is sparse training induced by large bias initialization is of relevance. Please show empirical works that use large bias initialization to get improved generalization and I may reconsider. However I am as of now still unconvinced.
> > > > > > >
> > > > > > > **A2**: In our last message, we meant that the trainable bias shares the same category with dynamic sparse activation training. We agree that the practical algorithms used for sparse training is different and typically very complicated. Here given the current tools and techniques in deep learning theory, we adopt a simplified model and consider sparsity induced by a nonzero bias initialization. The same model has been used in [1], which appears to be well received by the ML community.
> > > > > > >
> > > > > > > Nevertheless, our theory does indicate that sparse activation induced by large bias initialization provides computational cost reduction while preserving convergence and generalization. It will be an important future work to generalize this simplified model to more complicated sparse setting.
> > > > > > >
> > > > > > > [1] Song, Zhao, Shuo Yang, and Ruizhe Zhang. "Does Preprocessing Help Training Over-parameterized Neural Networks?." Advances in Neural Information Processing Systems 34 (2021): 22890-22904.
> > > > > > >
> > > > > > > [2] Arora, Sanjeev, et al. "Fine-grained analysis of optimization and generalization for overparameterized two-layer neural networks." International Conference on Machine Learning. PMLR, 2019. URL https://proceedings.mlr.press/v97/arora19a. html.

---

> > > > > > > > ### Comment · Reviewer_neAa · 2022-11-18
> > > > > > > > **Response to authors**
> > > > > > > >
> > > > > > > > A1
> > > > > > > >
> > > > > > > > (a) I acknowledge this improvement over [1], still my reservations are the same as I question the soundness of sparsity from large bias initialization in general.
> > > > > > > >
> > > > > > > > (b) "Further, we have provided a new lower bound on the minimum eigenvalue of the limiting NTK on a restricted region, which is free of explicit dependence on number of training sample . This further establishes the vanishing behavior of the least eigenvalue, but such a result is not in [2]".  I should have mentioned this in my original review but Theorem 3.1.1 is not a statement about the smallest eigenvalue of the NTK, as the smallest eigenvalue of the NTK is given by the minimum of $a^T H^\infty a$ where $a$ is taken to be over the entire sphere $S^{n - 1}$.  This is a sufficient condition on the vector $a$ so that the quadratic $a^T H^\infty a$ is $\Omega(1)$.  This is misleading to call this the restricted smallest eigenvalue of the NTK.  What this translates to is a sufficient condition on the labels so that the term $y^T (H^\infty)^{-1} y$ is bounded.  *This is making an additional assumption on the distribution of the labels and is not a statement about the smallest eigenvalue of the NTK*.  The prior works [1] and [2] could have made this assumption as well to get improved results.
> > > > > > > >
> > > > > > > > (c) "We also would like to point out that we choose our setting because it makes the least assumption on the data that we know of. The only assumption we make on the training data is data separation, instead of assuming the underlying distribution has some desirable properties like sub-Gaussian concentration, moment bounds, etc, or that the underlying true function is low rank, etc."  In Theorem 3.11 and Remark 3.12 *you are making an assumption on the data*.  Namely you are assuming that the labels lie in the favorable region outlined in Theorem 3.11.  Note assuming $y^T (H^\infty)^{-1} y >> \lambda_{min}(H^\infty)^{-1} \lVert y \rVert_2^2$ is equivalent to assuming that the labels mostly lie in the top eigenspaces of $H^\infty$, which is precisely what is studied in [3].
> > > > > > > >
> > > > > > > >
> > > > > > > > [1] Song, Zhao, Shuo Yang, and Ruizhe Zhang. "Does Preprocessing Help Training Over-parameterized Neural Networks?." Advances in Neural Information Processing Systems 34 (2021): 22890-22904.
> > > > > > > >
> > > > > > > > [2] Arora, Sanjeev, et al. "Fine-grained analysis of optimization and generalization for overparameterized two-layer neural networks." International Conference on Machine Learning. PMLR, 2019. URL https://proceedings.mlr.press/v97/arora19a. html.
> > > > > > > >
> > > > > > > > [3] Lili Su and Pengkun Yang. On learning over-parameterized neural networks: A functional approximation perspective. In H. Wallach, H. Larochelle, A. Beygelzimer, F. d'Alché-Buc, E. Fox,
> > > > > > > > and R. Garnett (eds.), Advances in Neural Information Processing Systems, volume 32. Curran Associates, Inc., 2019. URL https://proceedings.neurips.cc/paper/2019/file/
> > > > > > > > 253f7b5d921338af34da817c00f42753-Paper.pdf.

---

> > > > > > > > > ### Author Response · Authors · 2022-11-22
> > > > > > > > > **Response to the Reviewer**
> > > > > > > > >
> > > > > > > > > We thank the reviewer for spending time on discussing with us. Although we respectfully disagree with some of the reviewer's comments, we appreciate the opportunities of interacting with the expert on this topic, which we believe significantly improves our paper.
> > > > > > > > >
> > > > > > > > > **Q** I question the soundness of sparsity from large bias initialization in general.
> > > > > > > > >
> > > > > > > > > **A** This form of sparsity has been well accepted in the theory community as we mentioned in our previous message.
> > > > > > > > >
> > > > > > > > > **Q** This is misleading to call this the restricted smallest eigenvalue of the NTK.
> > > > > > > > >
> > > > > > > > > **A** The name "restricted smallest eigenvalue" is inspired by the restricted eigenvalue condition in the compressed sensing literature, e.g., [4].
> > > > > > > > >
> > > > > > > > > **Q** The prior works [1] and [2] could have made this assumption as well to get improved results.
> > > > > > > > >
> > > > > > > > > **A**
> > > > > > > > > We respectfully disagree with the above point. As a matter of fact, **even with this assumption, [1] and [2] may not have an improvement, because [1] and [2] don't have trainable bias.** It is the trainable bias that yields an additional term in the infinite-width NTK: $\mathbb{E}\_{w \sim \mathcal{N}(0, I)} [\mathbb{I}(X w \geq B) \mathbb{I}(X w \geq B)^\top]$, and Theorem 3.11 essentially provides a lower bound on the restricted eigenvalue of this matrix. Without the trainable bias, the infinite-width NTK is given by $\mathbb{E}_{w \sim \mathcal{N}(0, I)} [Z(w)^\top Z(w)] $ where $Z(w) = [x_1 \mathbb{I}(w^\top x_1 \geq B), x_2 \mathbb{I}(w^\top x_2 \geq B), \ldots, x_n \mathbb{I}(w^\top x_n \geq B)]$. It is unclear whether the restricted smallest eigenvalue of such a matrix have any improvement; at least our analysis tools won't hold here.
> > > > > > > > >
> > > > > > > > > We emphasize that obtaining the improved result even under new assumptions is interesting, and is absolutely the contribution of this paper. We fully disagree that the reviewer chooses to ignore this paper's contribution by hypothetically assuming that "[1] and [2] could have made the assumption to get the improved result". We believe no research community supports such a logic.
> > > > > > > > >
> > > > > > > > >
> > > > > > > > >
> > > > > > > > > **Q**
> > > > > > > > > Note assuming $ y^\top (H^\infty)^{-1} y \gg \||y\||\_2^2 / \lambda\_{\textnormal{min}}(H^\infty)$ is equivalent to assuming that the labels mostly lie in the top eigenspaces of $H^\infty$, which is precisely what is studied in [3].
> > > > > > > > >
> > > > > > > > > **A** We respectfully disagree with the above point. As a matter of fact, the two studies are not equivalent at all. There are two clear differences.
> > > > > > > > >
> > > > > > > > > * Let the labels are generated by the true function $f^\star$, i.e., $y = f^\star(x)$. The true function $f^\star$ in our setting can lie on the **entire** eigenspace of the spectrum of the NTK, not the **top** eigenspace. In contrast, [3] assumes that the underlying true function $f^\star$ has low rank structure of the underlying function, which then lies on the **top** eigenspace.
> > > > > > > > >
> > > > > > > > > * More importantly, our Theorem 3.11 captures an important behavior of the smallest eigenvalue: as the number of sample goes to infinity, the smallest eigenvalue goes to zero. In our Theorem 3.11, as the number of training samples goes to infinity, the data separation will go to zero and thus the lower bound of the restricted smallest eigenvalue will go to zero.
> > > > > > > > > Our contribution lies in improving the rate this lower bound goes to zero (now without the explicit dependence on $n$). Such phenomena are significantly different from what is captured [3], where the diminishing behavior of the smallest eigenvalue of the infinite-width NTK is avoided under the low rank assumption on $f^\star$. Thus, convergence will depend on the smallest eigenvalue on the restricted low rank eigen subspace which is *non-diminishing* as the number of training sample goes to infinity.
> > > > > > > > >
> > > > > > > > >
> > > > > > > > > [1] Song, Zhao, Shuo Yang, and Ruizhe Zhang. "Does Preprocessing Help Training Over-parameterized Neural Networks?." Advances in Neural Information Processing Systems 34 (2021): 22890-22904.
> > > > > > > > >
> > > > > > > > > [2] Arora, Sanjeev, et al. "Fine-grained analysis of optimization and generalization for overparameterized two-layer neural networks." International Conference on Machine Learning. PMLR, 2019. URL https://proceedings.mlr.press/v97/arora19a. html.
> > > > > > > > >
> > > > > > > > > [3] Lili Su and Pengkun Yang. On learning over-parameterized neural networks: A functional approximation perspective. In H. Wallach, H. Larochelle, A. Beygelzimer, F. d'Alché-Buc, E. Fox, and R. Garnett (eds.), Advances in Neural Information Processing Systems, volume 32. Curran Associates, Inc., 2019. URL https://proceedings.neurips.cc/paper/2019/file/ 253f7b5d921338af34da817c00f42753-Paper.pdf.
> > > > > > > > >
> > > > > > > > > [4] Raskutti, Garvesh, Martin J. Wainwright, and Bin Yu. "Restricted eigenvalue properties for correlated Gaussian designs." The Journal of Machine Learning Research 11 (2010): 2241-2259.

---

### Official Review · Reviewer_crhr · 2022-10-26

**Confidence:** 4
**Clarity, Quality, Novelty And Reproducibility:** Well written, but somewhat incrementa…
**Correctness:** 4
**Technical Novelty And Significance:** 2
**Empirical Novelty And Significance:** Not applicable
**Recommendation:** 5

**Strength And Weaknesses:**

Strenghts: The tighter bounds on width and generalization seem to apply regardless of the sparsity level and could be a useful addition to the NTK literature.

Weaknesses:

I have trouble understanding the motivation behind this work: in the NTK regime, using a different initialization of the bias basically corresponds to using a different activation, namely a shifted ReLU $\sigma(\cdot - B)$ instead of $\sigma$. While this may lead to sparser activation patterns, the width required to achieve similar guarantees to the $B = 0$ case (in the kernel regime at least) needs to be much larger, basically compensating the possible sparsity benefits.

It is also unclear if the use of a different activation requires an entirely new analysis, as opposed to using previous results. For instance, the bounds on least singular values of the kernel matrix have been tightly characterized in various settings with generic activations, e.g. [here](https://arxiv.org/abs/2007.12826) in certain high dimensional regimes. In particular, such statistical properties typically depend on Hermite coefficients of the activation, and for the shifted ReLU it should be easy to see that these are more or less scaled down by a factor $\sim e^{-B^2/2}$ compared to the ReLU coefficients.

That said, I might have missed the motivation of the authors, and encourage them to explain this further. If the key contribution is about improving the bounds, then I'm not sure activation sparsity should be part of the story?

**Summary Of The Paper:**

The paper considers optimization and generalization properties one-hidden-layer ReLU networks trained in the NTK regime, with an initialization of the bias that is fixed to a non-zero constant $B \geq 0$. This leads to networks with sparse activation patterns, which may be useful for sparsifying network parameters after training. The authors also provide improved bounds on width and generalization compared to previous works studying the NTK regime in similar settings, and complement the theory with simple experiments illustrating the sparsity of activations.

**Summary Of The Review:**

The technical contributions are potentially useful, but the motivation is unclear.

---

> ### Author Response · Authors · 2022-11-14
> **Response to Reviewer crhr**
>
> We thank Reviewer crhr for the review. We address the reviewer's concerns below.
>
> **Q:** I have trouble understanding the motivation behind this work: in the NTK regime, using a different initialization of the bias basically corresponds to using a different activation, namely a shifted ReLU instead of $\sigma(\cdot - B)$. While this may lead to sparser activation patterns, the width required to achieve similar guarantees to the case (in the kernel regime at least) needs to be much larger, basically compensating the possible sparsity benefits.
>
> **A:**
> Many thanks for the question. Given the tools and techniques available in deep learning theory, initializing bias as a constant is a simplified, yet still sensible way to capture the sparse-to-sparse training practice such as [6,7]. Our results based on this simple model can already capture the benefits of sparse training. As we proved in Lemma 3.5, such an initialization yields sparse activation during the entire training. This shows that the computational complexity per step training can be significantly reduced as in [1,3,4,5], capturing the main benefit of inducing sparsity.
> Since we have shown that sparsity doesn't slow down convergence rate (as we describe above), the overall computational cost over the entire training is reduced.
>
> Regarding the network width, we want to point out that the motivation is not that we require a large network to make the sparse training work, which then incurs a large cost. Our motivation follows a different logic. Our starting point is that the network width is typically picked to be large in practice due to various considerations such as expressive capacity, optimization and generalization.
> This thus motivates the sparse training to reduce the computation cost and speedup the training. Thus, our study here implies that as long as we initialize the bias parameter $B$ accordingly, then the computational cost of the training can be significantly reduced while preserving convergence and generalization of the non-sparsified large model. We further added a new section Appendix E to explain the benefit of this initialization in details.
>
> **Q:** It is also unclear if the use of a different activation requires an entirely new analysis, as opposed to using previous results. For instance, the bounds on least singular values of the kernel matrix have been tightly characterized in various settings with generic activations, e.g. [2] in certain high dimensional regimes. In particular, such statistical properties typically depend on Hermite coefficients of the activation, and for the shifted ReLU it should be easy to see that these are more or less scaled down by a $e^{-B^2/2}$ factor compared to the ReLU coefficients.
>
> **A:**
> While we agree with the reviewer's above comments such as it can be more or less seen that the least eigenvalue is scaled by $e^{-B^2/2}$ as in [2], those are not our contributions in this paper. Instead,
> the contribution of our work lies in a sharper analysis of the convergence in sparsely activated network training, a sparsity-dependent generalization result, and a novel restricted least eigenvalue lower bound in this sparse activation regime. These results cannot be derived from the least eigenvalue of the kernel matrix in previous studies. For example, we don't see how the result in [2] would imply the lower bound on network width $m$ in convergence (our Theorem 3.1) and generalization (our Theorem 3.8).
>
>
>
> **Q:** That said, I might have missed the motivation of the authors, and encourage them to explain this further. If the key contribution is about improving the bounds, then I'm not sure activation sparsity should be part of the story?
>
> **A:** In fact, our motivation lies in improving the bounds for analyzing the activation sparse training. Hence, **sparse training** is the context of the problem for which we improve the convergence and generalization bounds and is of central importance in the story.
>
> More specifically, we focus on improving the following bounds for sparse training:
> (a) Even though [1] considers bias with constant initialization to induce sparsity, the bias is not trainable, contrary to what people are doing in the practice; Thus, we would like to analyze the impact of trainable bias on the convergence and generalization bounds. (b) We suspect the convergence analysis in [1] is not tight, and our study sharpens the bounds on the convergence. (c) The generalization analysis is missing in the previous studies, although it is of central interest and importance in machine learning. Thus, we want to fill such a gap, and study the relationship between sparsity and generalization.
> Further, our study develops a novel result on restricted least eigenvalue on some data-dependent region that sharpens the dependence on the $n$.

---

> > ### Author Response · Authors · 2022-11-14
> > **Response to Reviewer crhr (continued)**
> >
> > [1] Song, Zhao, Shuo Yang, and Ruizhe Zhang. "Does Preprocessing Help Training Over-parameterized Neural Networks?." Advances in Neural Information Processing Systems 34 (2021): 22890-22904.
> >
> > [2] Montanari, Andrea, and Yiqiao Zhong. "The interpolation phase transition in neural networks: Memorization and generalization under lazy training." The Annals of Statistics 50.5 (2022): 2816-2847.
> >
> > [3] Song, Zhao, Lichen Zhang, and Ruizhe Zhang. "Training Multi-Layer Over-Parametrized Neural Network in Subquadratic Time." arXiv preprint arXiv:2112.07628 (2021).
> >
> > [4] Hu, Hang, et al. "Training overparametrized neural networks in sublinear time." arXiv preprint arXiv:2208.04508 (2022).
> >
> > [5] Gao, Yeqi, et al. "A Sublinear Adversarial Training Algorithm." arXiv preprint arXiv:2208.05395 (2022).
> >
> > [6] Liu, Shiwei, et al. "Sparse training via boosting pruning plasticity with neuroregeneration." Advances in Neural Information Processing Systems 34 (2021): 9908-9922.
> >
> > [7] Liu, Shiwei, et al. "Do we actually need dense over-parameterization? in-time over-parameterization in sparse training." International Conference on Machine Learning. PMLR, 2021.

---

### Official Review · Reviewer_dzEj · 2022-10-27

**Confidence:** 3
**Correctness:** 4
**Technical Novelty And Significance:** 3
**Empirical Novelty And Significance:** Not applicable
**Recommendation:** 6

**Clarity, Quality, Novelty And Reproducibility:**

Clarity: The paper is very well-written in general though some discussion seems missing.

Quality: The paper provides solid theoretical study. On a technical level, the significance seems a bit limited as the analysis are mostly based on adapting existing analysis (for the convergence) or making strong assumptions (for bounding the smallest eigenvalue).

Novelty: I believe the theoretical study of a two-layer NN with a trainable and nontrivially initialized bias is novel.

**Strength And Weaknesses:**

# Strength

Sparsity in activation is becoming increasingly important as the size of the models we are training are becoming very large and resource-consuming to use. Existing theoretical study on networks with sparse activations seem rare, and this work may motivate many subsequent studies along this direction, potentially providing justification or guidance for introducing sparse activation in large models in more principled ways.

The paper is very well written and is easy to follow despite that it contains a lot of theoretical formulations and claims. Though certain relevant discussions seem missing.

The theoretical results on generalizatoin, convergence, and smallest eigenvalue seems interesting. (Though I am not in a position to understand their significance / novelty on a technical level)

# Weakness

Two major questions I have:

- Section 3.1 is structured as first presenting the main result (i.e., Theorem 3.1), then explains the key ideas of proving this result and some lemmas along the way. Here, I am not understanding how these lemmas lead to Theorem 3.1.

Specifically, Thm 3.1 is about training loss decaying to zero. Lemma 3.3 is about bounding activation flipping probability when the change in network parameter is not too large. However, I don't see a discussion on 1) why we expect the change in parameter is not too large, and 2) what is the benefit of bounding the activation flipping probability. Then, Lemma 3.4 provides an error bound on initial loss L(0), but there is no discussion on why we need to bound it given that we only need to bound the ratio L(t) / L(0).

- The paper is about sparsity in activation maps, but there is in general a lack of discussion on 1) how / whether sparsity helps or provides benefits with proving the convergence / generalization results, and 2) how / whether sparsity facilitates convergence / improves generalization by their theoretical results, and whether such results aligns with what can be observed in practice (e.g. faster convergence and better generalization).

Other comments:

- Theorem 3.1: Maybe I missed it but I did not find where delta on the RHS of the inequality is defined.

- Remark 3.2: Is the improvement over Song et al. obtained because this paper considers trainable bias while Song et al. considers non-trainable bias?

- Definition 3.10: Data-dependent Region. This assumption is the key for the third contribution, but it seems quite arbitrary in the sense that we don't know if or how likely it is going to be satisfied in practice.

# Additional Comments

There are some references on sparse activation in existing network architectures. These are particularly relevant as their sparsity emerges automatically from regular training.

- Rhu, Minsoo, et al. "Compressing DMA engine: Leveraging activation sparsity for training deep neural networks." 2018 IEEE International Symposium on High Performance Computer Architecture (HPCA). IEEE, 2018.
- Andriushchenko, Maksym, et al. "SGD with large step sizes learns sparse features." arXiv preprint arXiv:2210.05337 (2022).
- Li, Zonglin, et al. "Large Models are Parsimonious Learners: Activation Sparsity in Trained Transformers." arXiv preprint arXiv:2210.06313 (2022).


**Summary Of The Paper:**

This is a theoretical paper on an NTK analysis of two-layer ReLU network with a trainable nontrivial initial bias. The motivation of study is that by using an appropriately initialized bias term, the network produces *sparse* activation at initialization; and if one considers the kernel regime then the sparsity is maintained throughout training, hence may be leveraged for efficiency purposes (but this application is not the focus of this paper).

The main contributions are:

- A convergence analysis to show that GD drives training error towards zero;
- A generalization analysis to show generalization bound via Rademacher complexity;
- (On a technical level) A new bound on the smallest eigenvalue of limiting NTK, by resorting to a "data-dependent region".

**Summary Of The Review:**

The paper studies the important problem of sparse activation using a testbed of two-layer neural network in the NTK regime. The presentation is mostly good but there may be some missing discussions. The technical novelty seems a bit weak (though I am not familiar with related studies).

---

> ### Author Response · Authors · 2022-11-14
> **Response to Reviewer dzEj**
>
> We thank Reviewer dzEj for the supportive review and providing relevant references.
> We address the reviewer's concerns here.
>
>
> **Q:** Section 3.1 is structured as first presenting the main result (i.e., Theorem 3.1), then explains the key ideas of proving this result and some lemmas along the way. Here, I am not understanding how these lemmas lead to Theorem 3.1.
>
> Specifically, Thm 3.1 is about training loss decaying to zero. Lemma 3.3 is about bounding activation flipping probability when the change in network parameter is not too large. However, I don't see a discussion on 1) why we expect the change in parameter is not too large, and 2) what is the benefit of bounding the activation flipping probability. Then, Lemma 3.4 provides an error bound on initial loss L(0), but there is no discussion on why we need to bound it given that we only need to bound the ratio L(t) / L(0).
>
> **A:** We provide to the reviewer's three questions below.
>
> - **Why do we expect the neuron weight change is not too large.** This is a standard result typically expected from the NTK type analysis. In the NTK regime, the difference between the neuron weights at the initialization and after training is diminishing with the network width (since we are considering a kernel with sparse activation, we show this result for our setting in Lemma A.9 in the Appendix). Thus, when the network width is large enough, the neuron weight will stay close to the initialization after training (which is why [1] calls this regime "lazy training").
>
> - **The benefit of bounding activation flipping probability.** This is a technical point. We need to show that the NTK during training is close to the NTK at the initialization and thus the smallest eigenvalue of the NTK is always positive. There are two sources that drive the change of NTK: (1) the weight change for activated neuron and (2) neurons going from inactive to active. Bounding the flipping probability can help bound the second source of the change of NTK, which can be further applied to showing that the training error goes down to zero in linear rate.
>
> - **Why do we need to bound $L(0)$ since we only care about $L(t) / L(0)$?** The proof of Theorem 3.1 indeed requires an upper bound on $L(0)$ to bound the neuron weight change. Further, this bound is used in proving the generalization in Theorem 3.8, which needs gradient descent to drive the training loss below a certain small value.
> Also, in practice, the goal of training is to have the training error below a threshold $\epsilon$, and thus bounding how big the initial error is can be of useful.
>
> **Q:** The paper is about sparsity in activation maps, but there is in general a lack of discussion on 1) how / whether sparsity helps or provides benefits with proving the convergence / generalization results, and 2) how / whether sparsity facilitates convergence / improves generalization by their theoretical results, and whether such results aligns with what can be observed in practice (e.g. faster convergence and better generalization).
>
> **A:**
> The major advantage of sparsity is that it can significantly reduce the computational cost while preserving convergence (Theorem 3.1) and generalization (Theorem 3.8).
> As we proved in Lemma 3.5, the initialization yields sparse activation and it can be preserved during the entire training. It can be shown that the computational complexity per step training can be significantly reduced as in [2,3,4,5], capturing the main benefit of inducing sparsity. Since we have shown that sparsity doesn't slow down the convergence rate (as we describe in our general response), the overall computational cost over the entire training is reduced.
>
> **Q:** Theorem 3.1: Maybe I missed it but I did not find where $\delta$ on the RHS of the inequality is defined.
>
> **A:** $\delta$ denotes the failure probability. Thanks for pointing it out. We will clarify in the revision.
>
> **Q:** Remark 3.2: Is the improvement over Song et al. obtained because this paper considers trainable bias while Song et al. considers non-trainable bias?
>
> **A:** Not exactly. The improvement is due to a sharper analysis of the convergence, as we describe in Section 3.1 (even without trainable bias). Specifically, we derived a sharper upper bound $O(R\exp(-B^2/2))$ on the flipping probability in Lemma 3.3 whereas Song, et al. derived a $O(\min\{R, \exp(-B^2/2)\})$ bound.
> We derived a sharper characterization of the initial error in Lemma 3.4 of $\tilde{O}(n)$ where as Song, et al. derived a bound of $\tilde{O}(n(1 + B^2))$.
> Also, in our revision, we relaxed the condition on the learning rate which leads to as fast convergence as the non-sparse network.

---

> > ### Author Response · Authors · 2022-11-14
> > **Response to Reviewer dzEj (Continued)**
> >
> > **Q:**
> > Definition 3.10: Data-dependent Region. This assumption is the key for the third contribution, but it seems quite arbitrary in the sense that we don't know if or how likely it is going to be satisfied in practice.
> >
> > **A:**
> > First of all, the region is not something artificial, but can be achieved by simply shifting the label vector $y$ into the positive orthant (see our comments in Remark 3.12). This is important since the previous bound in the literature will yield a generalization bound of $O(n)$ which is clearly not tight as the loss is **bounded**, whereas our bound will yield $O(1)$ generalization bound for fixed data separation. We also want to highlight the advantage of our result that if the label vector $y$ is in this region, then the lower bound of the restricted least eigenvalue on this region is better than the lower bound of the least eigenvalue of the entire domain by a factor of $n^2$.
> >
> > [1] Chizat, Lenaic, Edouard Oyallon, and Francis Bach. "On lazy training in differentiable programming." Advances in Neural Information Processing Systems 32 (2019).
> >
> > [2] Song, Zhao, Shuo Yang, and Ruizhe Zhang. "Does Preprocessing Help Training Over-parameterized Neural Networks?." Advances in Neural Information Processing Systems 34 (2021): 22890-22904.
> >
> > [3] Song, Zhao, Lichen Zhang, and Ruizhe Zhang. "Training Multi-Layer Over-Parametrized Neural Network in Subquadratic Time." arXiv preprint arXiv:2112.07628 (2021).
> >
> > [4] Hu, Hang, et al. "Training overparametrized neural networks in sublinear time." arXiv preprint arXiv:2208.04508 (2022).
> >
> > [5] Gao, Yeqi, et al. "A Sublinear Adversarial Training Algorithm." arXiv preprint arXiv:2208.05395 (2022).

---

### Official Review · Reviewer_D4Qh · 2022-10-30

**Confidence:** 4
**Clarity, Quality, Novelty And Reproducibility:** The paper is written clearly.
**Correctness:** 4
**Technical Novelty And Significance:** 3
**Empirical Novelty And Significance:** 2
**Recommendation:** 6

**Strength And Weaknesses:**

Strengths:

The paper generalises classic NTK results to the trainable bias setting which is the standard setting for several training protocols.

The paper gives short and simple proof sketches to get the gist of the proof.The proof sketches do a good job of capturing the main technical novelty/contribution in the paper.

Careful analysis and development of new proof techniques to get better bounds.


Weaknesses:

1. Trainable biases with non-constant/random init can easily be modelled with no-bias results using modified data with a concatenated constant to the feature vector. What advantage does constant bias init have over this init? This needs to be made clear.

2. The advantage of a non-zero B initialisation seems non-existent in all the theoretical results -- Theorem 3.1 requires a larger net, and achieves slower convergence with a non-zero B when compared to B=0. The first (and more dominant) term of Rademacher complexity in Theorem 3.8 is independent of B as mentioned in the para after the theorem. Similarly in Theorem 3.11 using a non-zero B seems to only reduce the smallest eigen value. So why should one bother with this initialisation at all?

3. The definition of "data-dependent region" is confusing. What is the Probabilitiy in definition of P_{ij} over?

4. The experimental results for different initial biases is underwhelming and incomplete. Why not give results for positive B that would actually increase the number of inactive neurons?  In the experiments here the non-zero biases seem to increase the active neurons, while actually reducing the accuracy.

5. It is also not clear which of the three main theorems the experimental result demonstrates. Perhaps it demonstrates a key property exploited by the Theorems, in which case this needs to be stated as a separate theorem.



**Summary Of The Paper:**

The paper analyses 1 layer ReLU nets with trainable biases under a fixed bias init using the NTK framework. It polishes the proof techniques used in various other papers and gives better bounds for the overparameterisation required to be in the NTK regime.

**Summary Of The Review:**

Despite my issues listed in the weakness section, I think the paper makes enough technical contributions and proof technique ideas to merit an acceptance.

The paper would do very well to motivate why any practitioner would want a constant non-zero bias init both empirically and theoretically.

---

> ### Author Response · Authors · 2022-11-14
> **Response to Reviewer D4Qh**
>
> We thank Reviewer D4Qh for the review and support of our work.
> We address the reviewer's concerns here.
>
> **Q:** Trainable biases with non-constant/random init can easily be modelled with no-bias results using modified data with a concatenated constant to the feature vector. What advantage does constant bias init have over this init?
>
> **A:** Random initialization doesn't induce sparsity in activation, whereas constant initialization can induce sparsity and hence reduce the computational complexity in the training.
>
> **Q:** The advantage of a non-zero B initialisation seems non-existent in all the theoretical results -- Theorem 3.1 requires a larger net, and achieves slower convergence with a non-zero B when compared to B=0. The first (and more dominant) term of Rademacher complexity in Theorem 3.8 is independent of B as mentioned in the para after the theorem. Similarly in Theorem 3.11 using a non-zero B seems to only reduce the smallest eigenvalue. So why should one bother with this initialisation at all?
>
> **A:** Regarding the slower convergence, excitingly, we were able to improve our analysis in the revision, and show that sparsity activated network achieves *the same convergence rate* for all sparsity (i.e., all bias initialization value $B$), and thus sparsity doesn't hurt convergence. Especially, the learning rate is now allowed to be chosen from a larger range. This result is obtained as the following. By utilizing the fact that the network has sparse activation, we improve Lemma A.17 with a sharper bound. This leads to a more relaxed condition on learning rate $\eta$ in Theorem A.18 (same as Theorem 3.1): the upper bound on $\eta$ now allows an additional $\exp(B^2)$ factor which compensates the decrease of the smallest eigenvalue. This leads to better convergence.
>
> Regarding the advantage of non-zero B initialization, as we proved in Lemma 3.5, such an initialization yields sparse activation during the entire training. This shows that the computational complexity per step training can be significantly reduced as in [1,2,3,4], capturing the main benefit of inducing sparsity. Since we have shown that sparsity doesn't slow down the convergence rate (as we describe above), the overall computational cost over the entire training is reduced.
>
> Regarding the network width, we want to point out that the motivation is not that we require a large network to make the sparse training work, which then incurs a large cost. Our motivation follows a different logic. Our starting point is that the network width is typically picked to be large in practice due to various considerations such as expressive capacity, optimization and generalization. This thus naturally motivates the sparse training to reduce the computation cost and speedup the training. Thus,
> our study here implies that as long as we initialize the bias parameter $B$ accordingly, then the computation cost of the training can be significantly reduced, while preserving convergence and generalization of the non-sparsified large model.
>
>
>
> **Q:** The definition of "data-dependent region" is confusing. What is the probabilitiy in definition of $P_{ij}$ over?
>
> **A:** The probability is over the randomness of $w$, which follows a standard multi-dimensional Gaussian distribution, i.e., $p_{ij} = \Pr_{w \sim \mathcal{N}(0, I)} [w^\top x_i \geq B, w^\top x_j \geq B]$. We will clarify this in our revision.
>
>
> **Q:** The experimental results for different initial biases is underwhelming and incomplete. Why not give results for positive B that would actually increase the number of inactive neurons? In the experiments here the non-zero biases seem to increase the active neurons, while actually reducing the accuracy.
>
> **A:** When we say we initialize the bias term as $0, -0.5,-1$ in the experiment section, these values incorporate the negative sign in front of bias (see our neural network definition on page 3, Section 2.1; when we write code in Pytorch, we do need to include the negative sign) and thus will increase the number of inactive neuron.
> The "sparsity" on the y-axis of Figure 1 denotes the ratio of the number of inactive neurons to the total number of neurons. The higher sparsity means the more fraction of the neurons stays inactive. As we increase the magnitude of the bias initialization from 0 to -1.0 (from Figure 1 (a) to 1 (c)), the fraction of the inactive neurons increases.

---

> > ### Author Response · Authors · 2022-11-14
> > **Response to Reviewer D4Qh (Continued)**
> >
> > **Q:** It is also not clear which of the three main theorems the experimental result demonstrates. Perhaps it demonstrates a key property exploited by the Theorems, in which case this needs to be stated as a separate theorem.
> >
> > **A:** The experiment results are meant to demonstrate Lemma 3.3 and Lemma 3.5, which are key steps in proving the convergence. In particular, Lemma 3.3 shows that the change of sparsity will be small during the entire training and Lemma 3.5 shows that the number of activated neurons will be smaller (or the sparsity will be larger) during the entire training as we increase the bias initialization magnitude $B$.
> > In Figure 1 (a) to (c), as the bias initialization goes from 0 to -1.0, the sparsity becomes higher which verifies Lemma 3.5; and the fluctuation of the sparsity curve is small during the training which verifies Lemma 3.3.
> >
> >
> > [1] Song, Zhao, Shuo Yang, and Ruizhe Zhang. "Does Preprocessing Help Training Over-parameterized Neural Networks?." Advances in Neural Information Processing Systems 34 (2021): 22890-22904.
> >
> > [2] Song, Zhao, Lichen Zhang, and Ruizhe Zhang. "Training Multi-Layer Over-Parametrized Neural Network in Subquadratic Time." arXiv preprint arXiv:2112.07628 (2021).
> >
> > [3] Hu, Hang, et al. "Training overparametrized neural networks in sublinear time." arXiv preprint arXiv:2208.04508 (2022).
> >
> > [4] Gao, Yeqi, et al. "A Sublinear Adversarial Training Algorithm." arXiv preprint arXiv:2208.05395 (2022).

---

### Author Response · Authors · 2022-11-14
**General Response**

**Improved Convergence in the Revision**

We thank Reviewer D4Qh and Reviewer neAa for pointing out that our early result shows that sparsity slows convergence which leads to higher cost.
We refined our analysis, and especially, the learning rate is now allowed to be chosen from a larger range.
This result delivers a new message: the sparse network can achieve *the same convergence rate* regardless of sparsity level (i.e., it holds all bias initialization value $B$) under the NTK regime.
We believe this is the **first** result shows that sparsity doesn't hurt convergence.

This result is obtained as the following: it turns out that our old Lemma A.17 proves a sub-optimal bound which didn't utilize the fact that the network has sparse activation and the new analysis corrects that.
This leads to a more relaxing condition on learning rate $\eta$ in Theorem A.18 (same as Theorem 3.1): the upper bound on $\eta$ now allows an additional $\exp(B^2)$ factor which compensates the decrease of the smallest eigenvalue.
This leads to better convergence.

**The relationship between width and sparsity**

Empirically, it is known that large and dense networks always outperform small and dense networks.
However, training large and dense networks requires a large amount of resources.
Introducing sparsity can help develop efficient algorithms.
Also, in practice, network width is chosen before other hyperparameters.
Our theory shows that if the network width is chosen much larger than needed, then we can sparsify the network with constant bias initialization and the network will still preserve the same convergence and generalization.
We further added Appendix E to explain the benefit of our bias initialization.

The practical background of our setting is sparse training which can be divided in to dynamics sparse training or static sparse training, and, at the same time can also be divided into weight sparsity or activation sparsity.
Our work belongs to dynamic sparse training with activation sparsity, which in practice help memory efficiency [1, 2, 3].
To our knowledge, any category (out of 4 possible combination) has few theoretical work, and dynamic sparse training has none to our best knowledge.
As the first work, we choose a simplified theory model for analysis, which can lead to dynamic sparse activation in training.

[1] Li, Zonglin, et al. "Large Models are Parsimonious Learners: Activation Sparsity in Trained Transformers." arXiv preprint arXiv:2210.06313 (2022).

[2] Jiang, Ziyu, et al. "Back Razor: Memory-Efficient Transfer Learning by Self-Sparsified Backpropogation" Advances in Neural Information Processing Systems (2022), https://openreview.net/forum?id=mTXQIpXPDbh

[3] Raihan, Md Aamir, and Tor Aamodt. "Sparse weight activation training." Advances in Neural Information Processing Systems 33 (2020): 15625-15638.

---

### Decision · Program_Chairs · 2023-01-20

**Decision:**

Reject

**Justification For Why Not Higher Score:**

While this paper presents new results related to generalization bounds and convergence results for single-layer NTK ReLU networks, the manner in which those results are utilized to draw conclusions about dynamic sparse training remains unconvincing.

**Justification For Why Not Lower Score:**

N/A.

**Metareview: Summary, Strengths And Weaknesses:**

This paper examines wide single-hidden-layer ReLU networks with trainable biases that are initialized to a non-zero value, thereby inducing sparsity in the post-activation layer. In this context, the convergence of gradient descent is analyzed and a generalization bound is obtained, improving prior results and extending them to this setting.

The reviewers found the paper clear and relatively easy to read and they appreciated the importance of studying dynamic sparse training. They also acknowledged the improvements over prior results that follow even in the case where B=0. There were a variety of issues raised in the discussion, and although the authors addressed many of them, concerns remain about the manner in which sparsity is induced and studied in this paper.

In particular, it was noted that the constant initial bias can be absorbed into a redefinition of the activation function, and since many prior works have studied generic activation functions, the setting here would seem to be a special case of prior work. The authors note that their improved analysis relative to those prior works yields quantitatively better results and qualitatively new conclusions. However, I share the reviewers' concerns that interpreting those conclusions in terms of the types of sparsity emerging in practice may be a bit of a stretch. While I agree that certain liberties can be afforded to theoretical works, especially as they break new ground, in this case I believe that the messaging and framing of the paper have gone a bit too far. I would recommend that the authors either (a) develop and improve their empirical analysis to better support the connection between their results and practice, or (b) reframe the main takeaways of the paper to highlight the technical contributions with less of an explicit focus on sparsity.